# Generalization Bounds for Rank-sparse Neural Networks

**Antoine Ledent**
School of Computing and Information Systems (SCIS)
Singapore Management University (SMU)
aledent@smu.edu.sg

**Rodrigo Alves**
Department of Applied Mathematics
Czech Technical University in Prague (CTU)
rodrigo.alves@fit.cvut.cz

**Yunwen Lei**
Department of Mathematics
The University of Hong Kong
leiyw@hku.hk

## Abstract

It has been recently observed in much of the literature that neural networks exhibit a bottleneck rank property: for larger depths, the activation and weights of neural networks trained with gradient-based methods tend to be of approximately low rank. In fact, the rank of the activations of each layer converges to a fixed value referred to as the "bottleneck rank", which is the minimum rank required to represent the training data. This perspective is in line with the observation that regularizing linear networks (without activations) with weight decay is equivalent to minimizing the Schatten $p$ quasi norm of the neural network. In this paper we investigate the implications of this phenomenon for generalization. More specifically, we prove generalization bounds for neural networks which exploit the approximate low rank structure of the weight matrices if present. The final results rely on the Schatten $p$ quasi norms of the weight matrices: for small $p$, the bounds exhibit a sample complexity $\widetilde{O}(WrL^2)$ where $W$ and $L$ are the width and depth of the neural network respectively and where $r$ is the rank of the weight matrices. As $p$ increases, the bound behaves more like a norm-based bound instead.

## 1 Introduction

The success of neural networks in many applications during the most recent artificial intelligence boom has driven substantial research efforts into understanding their generalization behavior [1, 2, 3, 4, 5, 6]. Whilst early work [7] focused on traditional approaches such as VC dimensions and parameter count, the increasingly evident ability of DNNs to generalize well (even in massively *overparametrized* regimes) has shifted the community's focus beyond purely architectural bounds [8, 9]. It is now accepted that a satisfactory explanation of Neural Networks' generalizing abilities must incorporate more refined information about the training process and the properties of the representation functions, which spontaneously arise from training DNNs on *natural data* [10, 11].

One of the simplest ways to characterize the 'simplicity' of functions learnt by neural networks through the geometry of weight space is by considering *norms* of the *weight matrices*. Indeed, several works have provided various bounds with reduced or absent explicit dependence on architectural quantities, preferring to express neural network capacity through various functions of the weight matrices. Such generalization error bounds, which primarily rely on the norms of the weight matrices to express function class capacity are generally referred to as *'norm-based bounds'*. The earliest attempts include many bounds obtained through various vector contraction inequalities and the

39th Conference on Neural Information Processing Systems (NeurIPS 2025).

'*peeling technique*' [12, 13]. In this subcategory of bounds, the Rademacher complexity is bounded iteratively by peeling off layers, resulting in a product of norms of the weight matrices. For instance, the Rademacher complexity bound in [13], which extends earlier work from [12], scales as [1]:

$$\left( \sqrt{\frac{\mathfrak{B}^2 L \prod_{\ell=1}^{L} \|A_\ell\|_{\mathrm{Fr}}^2}{N}} \right). \tag{1.1}$$

Here, $N$ is the number of samples, $\mathfrak{B}$ is an upper bound on the norms of the samples and $A_1, \dots, A_L$ are the network's weight matrices. A remarkable feat of this bound is the complete lack of explicit dependence on architectural parameters (i.e., depth, width, etc.). However, the implicit dependence introduced by the product of Frobenius norm is large: even assuming that each weight matrix has unit spectral norm, if the rank of each $A_\ell$ is $r$ and each matrix has homogeneous spectrum, then the bound will still correspond to a sample complexity of $r^L$, an exponential dependence on the depth $L$.

Later norm-based bounds have largely favored alternative approaches via covering numbers, which yield a less dramatic exponential depth dependence. The most recognizable examples of such results include the bounds of [3] and [1], which were independently discovered through two approaches:

$$\tilde{O} \left( \frac{L\sqrt{\overline{W}}}{\sqrt{N}} \left( \prod_{\ell=1}^{L} \|A_\ell\| \right) \left( \sum_{\ell=1}^{L} \frac{\|A_\ell - M_\ell\|_{\mathrm{Fr}}^2}{\|A_\ell\|_\sigma^2} \right)^{\frac{1}{2}} \right), \tag{1.2}$$

$$\tilde{O} \left( \frac{1}{\sqrt{N}} \prod_{\ell=1}^{L} \|A_\ell\| \left( \sum_{\ell=1}^{L} \frac{\|(A_\ell - M_\ell)^\top\|_{2,1}^{\frac{2}{3}}}{\|A_\ell\|^{\frac{2}{3}}} \right)^{\frac{3}{2}} \right). \tag{1.3}$$

Here the $M_\ell$s are arbitrary *reference matrices* chosen in advance [2]. The implicit exponential dependence in depth is reduced to a product of *spectral* norms which, at least in theory, could be expected to scale as $O(1)$ or forced to be a unit value [14]. With this interpretation, it becomes clearer that the bounds improve when the networks stay close to initialization, especially if the number of updated weights is sparse, which would make the norm $\|(A_\ell - M_\ell)^\top\|_{2,1}$ small.

Such observations can begin to explain some of DNNs' successes, in particular with respect to the lottery ticket hypothesis [15]. However, they do not paint a complete picture, and are largely vacuous by large margins. In fact, the most impressive recent bounds in the theory of neural networks tend to implicitly incorporate data dependence. For instance, in [5, 6, 16] various upper bounds on norms of activations of intermediary layers and Lipschitz constants of gradients are replaced by empirical equivalents through a powerful technique called loss function augmentation, greatly attenuating the implicit exponential dependency on depth. Pushing it to the extreme, [11] prove bounds which incorporate a notion of confidence for intermediary concepts learnt at each layer.

The discovery of the Neural Tangent Kernel [17, 18] is certainly one of the biggest breakthroughs in deep learning theory in recent years. Indeed, the underlying theory demonstrates that Neural Networks behave approximately like kernels at the overparametrized regime, and are able to characterize the model's generalization behavior directly in terms of the alignment between the labels and the NTK features of the data distribution. However, such bounds rely on very strict overparametrization assumptions. Furthermore, it has been shown that the NTK regime does not outperform the moderate overparametrization regime [19, 20].

Another tremendous advance towards understanding the generalization power of neural networks occurred with the gradual understanding that *depth induces low-rank representations* [21, 22, 23, 24, 25, 26]. In other words, the training dynamic of DNNs through gradient based methods naturally encourages low-rank representations, converging to a bottleneck intrinsic dimension which captures the data's features parsimoniously. In particular, the connection between weight decay and implicit Schatten quasi-norm regularizaion has been made in [26, 24]. There has been a lot of recent interest in this phenomenon, which has been demonstrated experimentally in various contexts spanning both DNNs, CNNs and even LLMs [27]. The phenomenon can most easily be understood by initially considering the case of *linear neural networks*. In this case, we have the following powerful result:

---

[1]For simplicity, we assume that the nonlinearities are 1-Lipschitz in this related works section.
[2]For instance, they can be set to zero or the initialized values of the weights

**Theorem 1.1** (cf. [24], Theorem 1). *Let $w \geqslant \min(\mathcal{C}, d)$ and let $B_1, \ldots, B_L$ be matrices such that $B_1 \in \mathbb{R}^{w \times d}, B_L \in \mathbb{R}^{\mathcal{C} \times w}, B_2 \ldots, B_{L-1} \in \mathbb{R}^{w \times w}$. For any matrix $A \in \mathbb{R}^{\mathcal{C} \times d}$ we have*

$$L\|A\|_{\mathrm{sc}, \frac{2}{L}}^{\frac{2}{L}} = \min \sum_{\ell=1}^{L} \|B_\ell\|^2 \quad subject~to \quad B_L B_{L-1} \ldots B_1 = A. \tag{1.4}$$

For any matrix $Z$ and any $0 < p \leqslant 2$, our notation $\|Z\|_{\mathrm{sc},p}$ refers to the Schatten $p$ quasi norm of $Z$ [3]:

$$\|Z\|_{\mathrm{sc},p}^p = \sum_i \sigma_i(Z)^p, \tag{1.5}$$

where $\sigma_i(Z)$ refers to the $i$th singular value of $Z$ (listed in decreasing order). Thus, as $p$ approaches zero, the quantity $\|Z\|_{\mathrm{sc},p}^p$ approaches the rank of the matrix $Z$. Slightly larger values of $p$ induce a softer form of rank sparsity. Thus, Theorem 1.1 implies that a simple weight decay constraint on factor matrices $B_1, \ldots, B_L$ is equivalent to a rank-sparsity inducing constraint on $\|A\|_p^p$ where $p = \frac{2}{L} \to 0$ when $L \to \infty$. Deep results [24, 22] in optimization theory demonstrate that under specific circumstances, the above described implicit rank-sparsity inducing effect of weight decay at large depth *occurs even in the presence of activation functions*: the ranks of each layer's *post activation representations*[4] converges to a single quantity referred to as the "bottleneck rank", which can be understood as the minimum dimension required to effectively represent the data.

In this work, we characterize the effects of rank-sparsity in the weight matrices in terms of *generalization*. We provide bounds which *interpolate between the norm-based and parametric regimes* at each layer with the *parametric interpolation* technique originally developed for matrix completion [28]. This allows us to control the $L^\infty$ (cf. Prop. F.2) or $L^2$ (cf. Prop. I.3) covering number of each layer in term of the Schatten $p$ quasi norms of the weight matrices. Treating Lipschitz and boundedness conditions on the loss function as constant, our contributions are summarized as follows.

- We prove new generalization bounds for **linear networks** which incorporate the implicit low-rank effects of depth by incorporating the Schatten $p$ quasi norm of the weight matrix. As $p \to 0$, the bound provides a sample complexity of $\widetilde{O}\left([\mathcal{C} + d]\mathrm{rank}(A)\right)$. To the best of our knowledge, this is a new characterization of the generalization behavior of multi-class linear classification with low-rank dependencies between the classes.

- For fully-connected neural networks, we prove (cf. Theorem 3.2) bounds of the order of
$$\widetilde{O}\left(\sqrt{\frac{L}{N}}\left[\sum_{\ell=1}^{L}\left[\mathfrak{B}\prod_{i=1}^{L}\rho_i\|A_i\|\right]^{\frac{2p_\ell}{p_\ell+2}} \times \left[\frac{\|A_\ell - M_\ell\|_{\mathrm{sc},p_\ell}^{p_\ell}}{\|A_\ell\|^{p_\ell}}\right]^{\frac{2}{p_\ell+2}}[w_\ell + w_{\ell-1}]^{1+\frac{p_\ell}{p_\ell+2}}\right]^{\frac{1}{2}}\right).$$

- Our results hold simultaneously over every sequence of Schatten indices $p_\ell$s, which we tune to balance the effects of the norm-based factor $\left[\mathfrak{B}\prod_{i=1}^{L}\rho_i\|A_i\|\right]^{2p_\ell/[p_\ell+2]}$ and the low rank structure in the term $\|A_\ell\|_{\mathrm{sc},p_\ell}^{p_\ell}/\|A_\ell\|^{p_\ell}$ (for $M_\ell = 0$).

- We also provide extensions of those results relying on Loss Function Augmentation to replace the norm-based factor $\mathfrak{B}\prod_{i=1}^{L}\rho_i\|A_i\|$ by $\mathfrak{B}_{\ell-1,A}^{\mathfrak{C}}\mathfrak{B}\prod_{i=\ell}^{L}\rho_i\|A_i\|$ where $\mathfrak{B}_{\ell-1,A}^{\mathfrak{C}}$ is the maximum norm of a convolutional patch at layer $\ell - 1$.

- To the best of our knowledge, the proof technique, *parametric interpolation*, has only been previously used in the context of matrix completion [28], a substantially different setting.

## 2 Related Works

Deep Learning Theory is a vast and active area of research spanning various topics from optimization [29, 30, 31] to generalization [1, 32] and the intersection between both [17, 33]. In this section, we focus on norm-based and parameter-counting bounds for neural networks satisfying certain norm

---

[3]For $p \geqslant 1$, $\|\cdot\|_{\mathrm{sc},p}$ is a norm, but we refer to it as a quasi-norm to cover the cases $p < 1$ and $p \geqslant 1$.

[4]Whilst the low-rank properties of the *activations* and *weight matrices* are not exactly equivalent, they are closely related. Indeed, consider $N$ input activations $x_1, \ldots, x_N$ lying on a subspace of dimension $r$, then for any weight matrix $W \in \mathbb{R}^{m \times n}$, there exists a matrix $\overline{W} \in \mathbb{R}^{m \times d}$ of rank $\leqslant r$ s.t. $\overline{W}x = Wx$ for all $i$.

constraints. Indeed, this is the branch of literature closest to our work. A more mathematically detailed description of the related works is available in Sections B and C.

**Norm-based Bounds by peeling techniques** were established for Deep Neural Networks in [12] and [13], resulting in bounds involving products of Frobenius norms of the weights (cf. equation (1.1)). In [34], further norm-based bounds were proved involving the products of the $L^1$ norms of the columns of the weight matrices. Recently, the technique has also been ingeniously extended to **Convolutional Neural Networks (CNNs)** in [32], which elegantly exploits the sparsity of connections to achieve in bounds of the order $\frac{\mathrm{L_l} \prod_{\ell=1}^{L} \|A_\ell\|_{\mathrm{Fr}} \sqrt{L} \prod_{\ell=1}^{L} \underline{w}_\ell \underline{B}}{\sqrt{N}}$, where $\underline{w}_\ell$ is the spatial dimension of the convolutional patches at layer $\ell$ and $\underline{B}$ is the maximum $L^2$ norm of any convolutional patch. The results in [32] are not directly comparable to ours, and are especially well adapted to the case where the spatial dimension is very small. Indeed, as long as the convolutional filters are not one dimensional, the bound depends exponentially on depth. [16] also proves generalization bounds for CNNs which take weight sharing into account, and the bounds form the basis for the treatment of the norm based components in our proofs. However, the bounds in those works cannot incorporate rank-sparsity into the analysis and scale unfavorably with depth. There is also a variety of works which applies generalization analyses to non-standard learning settings or loss functions, including pairwise [35, 36, 37] learning or triplet-based learning and contrastive learning [38, 39, 40, 41, 42, 43, 44, 45, 46, 47, 48]. However, such works usually focus on the effect of the loss function. In addition to the above fundamental works, many works provide non-vacuous bounds for practical architectures through various post processing techniques including data-dependent priors [49], **optimization of the bound** or network compression/discretization [50, 51, 52, 53]. A particularly notable contribution is made in [54], which provides PAC Bayesian generalization bounds for two-layer networks where the main complexity term involves Froebenius norms of the weight matrices, remarkably, without an additional factor of the width, achieving non vacuous bounds in various cases. However, the bounds are PAC Bayesian and not directly comparable to ours since the training error is replaced by an expectation over a posterior distribution (derandomized analogues such as [55] reintroduce a factor of width), only GELU and ERF activation functions are covered and the number of layers is limited to 2. The **gradient dynamic** of deep neural networks and deep linear networks was widely studied from an optimization perspective to demonstrate neural rank collapse, with some works also covering networks with activations [24, 56, 27, 57, 58, 59, 60, 61, 62, 63]. In terms of the implications of such results for **linear networks** on *generalization*, one can of course consider the results which apply to neural networks with activations and apply them to this case [1, 3] (see Corollary C.1 for a transfer of such results to the linear case), or construct Rademacher complexity bounds specifically for linear networks as in [64]. However, such approaches involve an exponential dependence on depth which make it more difficult to capture any low-rank structure in the classes, since such structure appears for larger values of $L$ (smaller values of $p$). One can also consider all existing bounds for the *multi-class linear classification problem*, where the state of the art is provided by [65] which proves a bound of $\widetilde{O}\left(\sqrt{\frac{\|A\|_{\mathrm{Fr}}^2 \mathfrak{B}^2}{N}}\right)$, improving earlier results from [66, 67, 68] in terms of implicit dependence on $\mathcal{C}$. Since then, the result has been extended to many variants of the learning problem, including multi-label learning, structured output prediction and ranking problems [69, 70, 71, 72, 73].

Parameter counting bounds such as those of [4] and [74] apply both to CNNs and DNNs (in fact, [74] proves both norm based and parameter counting bounds for DNNs, CNNs and ResNets). Taking into account implicit factors of $L$ coming through logarithmic factors, the tightest parameter counting bound in both cases scales like $\widetilde{O}\left(\sqrt{\frac{\mathcal{W}L}{N}}\right)$ where $\mathcal{W}$ is the number of parameters in the network. Our own result in Corollary G.6 additionally takes the low-rank structure into account. Norm-based [75] and parameter counting [76] bounds have also been recently provided for transformers [77].

To the best of our knowledge, there are only two works which have utilized the *low-rank structure* in neural networks: [13], relies on Schatten $p$ norms of weight matrices to improve on the bound (1.1) for matrices with low Schatten $p$ quasi-norm. The argument relies on Lipschitz continuity arguments and yields bounds with a decay in $N$ worse than $O\left(1/\sqrt{N}\right)$. The recent work [64] manages to achieve tighter bounds taking explicit low-rank structure into account. The resulting bound scales as $\mathrm{L_l} C_1^L \mathfrak{B} \left[\prod_{\ell=1}^{L} \|A_\ell\|\right] L\underline{r}\sqrt{\frac{\overline{W}}{N}}$. One of the consequences of the proof strategy is the presence of

explicit exponential dependence on $L$ through the term $C_1^L$ where $C_1$ is an intractable constant [5]. In addition, even ignoring the factor of $C_1^L$, the bound still involves a product of spectral norms (with an exponent of 1), whereas our bounds involve a product of spectral norms with a potentially smaller, tunable exponent of $\frac{2p_\ell}{p_\ell+2}$. However, the elegant and short proof in [64] does imply that there are far fewer polylogarithmic factors compared to our bounds. See Appendix C for detailed formulae.

Our proof technique draws some inspiration from [28], which introduced the *parametric interpolation technique* to estimate the complexity of **matrix completion** with Schatten $p$ quasi norm constraints. However, the derivations are radically different. Indeed, in the matrix completion literature on *approximate recovery* [79, 80, 81], the *observations* are *entries* of an unknown matrix $A$: the independent variable $x = (i, j)$ in the supervised learning formulation is drawn from an arbitrary distribution over the discrete set of entries $[m] \times [n]$. In contrast, in our work, the *observations* are modelled as *outputs of a neural network*, i.e., they take the form $F(x_i)$ where $F$ is a neural network as defined in equation (3.6). Even for a single linear layer, the learning setting is different and the outputs take the form $Ax_i$ instead of $A_{i,j}$. The analysis in [28] relies on bounding Rademacher complexities directly through elaborate chaining arguments, bypassing the need for covering numbers and interpolating between $p = 0$ and $p = 1$. In fact, the authors mention that any readily obtainable covering number would yield vacuous bounds in the matrix completion context. In contrast, our analysis interpolates between $p = 0$ and $p = 2$ and produces chainable covers for weight spaces with Schatten constraints. A more detailed discussion of related works is provided in Appendix C.

## 3   Main Results

We consider a learning problem using a loss function $l : \mathbb{R}^{\mathcal{C}} \times \mathcal{Y} \to \mathbb{R}^{+}$, where $\mathcal{C}$ is the number of classes (in classification) or 1 (in regression). In this work, we assume loss function is $L_l$-Lipschitz with respect to the $L^{\infty}$ norm and uniformly bounded by $\mathcal{B}$. In regression, the output space is $\mathcal{Y} = \mathbb{R}$ whilst in classification, it is $\mathcal{Y} = \{1, \ldots, \mathcal{C}\}$. In the case of regression, an example of such a loss function is the truncated square loss

$$l(\hat{y}, y) = \min(\mathcal{B}, |\hat{y} - y|^2). \tag{3.1}$$

Indeed, in this case, the loss function is uniformly bounded by $\mathcal{B}$ and uniformly $2\mathcal{B}$-Lipschitz. For classification, a typical example used in much of the literature is the following margin loss [1, 4]:

$$l(\hat{y}, y) = \begin{cases} 1, & \text{if } \arg\max_i \hat{y}_i \neq y, \\ 1 - \frac{\hat{y}_y - \max\limits_{i \neq y} \hat{y}_i}{\gamma}, & \text{if } 0 \leqslant \hat{y}_y - \max\limits_{i \neq y} \hat{y}_i \leqslant \gamma, \\ 0, & \text{if } \hat{y}_y \geqslant \max\limits_{i \neq y} \hat{y}_i + \gamma. \end{cases} \tag{3.2}$$

In this case, the loss function is uniformly bounded by $\mathcal{B} = 1$ and uniformly $2/\gamma$-Lipschitz with respect to the $L^{\infty}$ norm. We assume the learner is provided with $N$ i.i.d. samples $x_1, \ldots, x_N \in \mathcal{X} = \mathbb{R}^d$ and the associated labels $y_1, \ldots, y_N \in \mathcal{Y} = \{1, 2, \ldots, \mathcal{C}\}$, each of which is drawn from a joint distribution $\mathcal{D} \sim \mathcal{X} \times \mathcal{Y}$. We are interested in high probability bounds on the *generalization gap*:

$$\text{GAP} := \mathbb{E}\left[l(F_A(x), y)\right] - \widehat{\mathbb{E}}\left[l(F_A(x), y)\right] = \mathbb{E}\left[l(F_A(x), y)\right] - \frac{1}{N}\sum_{i=1}^{N} l(F_A(x_i), y_i), \tag{3.3}$$

where $\widehat{\mathbb{E}}$ denotes the empirical expectation over the sample $(x_1, y_1), \ldots, (x_N, y_N)$, and $F_A = F_{(A_1, \ldots, A_L)}$ is our trained network. For classification, an advantage of the margin loss described in Eq. (3.2) is that a bound on GAP immediately translates to a bound on the probability of misclassification: let $I_{\gamma, X} = \frac{|i : \hat{y}_y - \max_{i \neq y} \hat{y}_i \leqslant \gamma|}{N}$ denote the proportion of samples which are either classified incorrectly or correctly but with margin $\leqslant \gamma$. The misclassification probability [1, 4, 74] satisfies

$$\mathbb{P}\left(\arg\max_c [F_A(x_i)]_c \neq y\right) \leqslant \mathbb{E}\left[l(F_A(x), y)\right] \leqslant \text{GAP} + \widehat{\mathbb{E}}\left[l(F_A(x), y)\right] \leqslant \text{GAP} + I_{\gamma, X}. \tag{3.4}$$

---

[5]The constant is described as 'rather large' in [78], since it accumulates factors of Talagrand's majorizing measure theorem and generic chaining, suggesting at an absolute minimum $C_1 \geqslant 11$.

We provide results for three types of models: Linear Networks, Deep Neural Networks, and Convolutional Neural Networks (CNN). In all the results below, the matrices $M_\ell$ are reference matrices which can take any fixed value as long as it is chosen in advance. Thus, they play an analogous role to a prior in PAC-Bayesian bounds [49]. Typical choices for $M_\ell$ include $M_\ell = 0$ or setting $M_\ell$ to the initialization when training a neural networks with gradient based methods. Whilst traditional norm-based bounds such as (1.3) typically benefit from setting $M_\ell$ to the initialization, our bounds typically perform better by setting $M_\ell = 0$, since this is the configuration which allows the quantity $\frac{\|A_\ell - M_\ell\|_{\mathrm{sc},p}^p}{\|A_\ell\|^p} = \frac{\|A_\ell\|_{\mathrm{sc},p}^p}{\|A_\ell\|^p}$ to approach $\mathrm{rank}(A_\ell)$. We provide an experimental evaluation of the behavior of our bounds on MNIST and CIFAR-10 for both DNNs and CNNs in Appendix D.

## 3.1 Linear Networks

We first present our results for linear networks, which amount to generalization bounds for multi-class *linear classification* with a bounded loss in the presence of a low-rank structure over the classes. However, the true value of those results is to illustrate the potential gains in generalization ability which arise from the implicit low-rank structure which appears in neural networks trained with gradient-based methods. In this section, we consider classifiers defined as follows:

$$F : \mathbb{R}^d \to \mathbb{R}^{\mathcal{C}} : x \to Ax = B_L B_{L-1} \ldots B_1 x \qquad (3.5)$$

for some matrices $B_L \in \mathbb{R}^{\mathcal{C} \times w}, B_{L-1} \in \mathbb{R}^{w \times w}, \ldots B_2 \in \mathbb{R}^{w \times w}, B_1 \in \mathbb{R}^{w \times d}$.

We have the following result, which shows the implications of Theorem 1.1 for generalization:

**Theorem 3.1.** *W.p.* $\geqslant 1 - \delta$ *over the draw of the training set, every linear network as defined in equation* (3.5) *satisfies the following generalization bound:* $\mathrm{GAP} - O\left(\mathcal{B}\sqrt{\frac{\log(1/\delta)}{N}}\right) \leqslant$

$$\widetilde{O}\left(\sqrt{\frac{[\mathrm{L_l}\mathfrak{B}]^{\frac{2p}{2+p}}\|A\|_{\mathrm{sc},p}^{\frac{2p}{2+p}}\min(\mathcal{C},d)^{\frac{p}{p+2}}[\mathcal{C}+d]^{\frac{2}{p+2}}}{N}}\right) \leqslant \widetilde{O}\left(\sqrt{[\mathrm{L_l}\mathfrak{B}]^{\frac{2}{L+1}}\frac{[\sum\|B_\ell\|_{\mathrm{Fr}}^2]^{\frac{L}{L+1}}[\mathcal{C}+d]}{NL^{\frac{L}{L+1}}}}\right)$$

*where* $\mathfrak{B} = \sup_{i=1}^N \|x_i\|$, *and the* $\widetilde{O}$ *hides polylogarithmic factors of* $\|A\|, \|A\|_{\mathrm{sc},p}, \mathcal{C}, d, N, p = \frac{2}{L}$.

The idea of the proof is to separate the function class into two components by applying a tunable thresholding operator in the singular values of the matrix $A$. A proof sketch is provided in subsection 3.2.1 below and the full proof is included in the appendix. Our full results in the appendix (cf. Theorem E.2) incorporate a finer dependency on architectural quantities such as the number of classes through a more refined analysis over different norms at each activation space, though we omit such subtleties from the main paper to reduce confusion. Assuming $L \gg 1$ and $\mathrm{L_l}, \mathfrak{B} \in O(1)$, $\|B_\ell\| = 1 \, \forall \ell$, our Theorem 3.1 scales like $\widetilde{O}\left(\frac{\sum_\ell \|B_\ell\|_{\mathrm{Fr}}^2[\mathcal{C}+d]}{L}\right)$ in sample complexity, whereas the known result (1.2) scales as $\widetilde{O}\left(\frac{L^3\min(\mathcal{C},d)\sum_\ell\|B_\ell\|_{\mathrm{Fr}}^2}{L}\right)$ in [3]. Thus, the simplified analysis in the linear case removes a factor of $L^3$, which allows taking the limit as $L \to \infty$: in this case, our bound behaves more and more like a parameter-counting bound since $\frac{\sum\|B_\ell\|_{\mathrm{Fr}}^2}{L} \to \mathrm{rank}(A)$. On the other hand, the dependency on the input space dimension $d$ is absent in (1.2). Whilst it may be possible to improve the dependency on $d$ in Theorem 3.1 assuming the input data lies on a low-dimensional subspace, this would require significant modifications to the proofs, and is best left to future work.

## 3.2 Fully Connected Neural Networks

We consider neural networks of the following form

$$x \to F_A(x) := A_L\sigma_L(A^{L-1}\sigma_{L-1}(\ldots\sigma_1(A^1 x)\ldots)), \qquad (3.6)$$

where the matrices $A^\ell \in \mathbb{R}^{w_\ell \times w_{\ell-1}}$ are the weight matrices, $A = (A^L, A^{L-1}, \ldots, A^1)$ denotes the set of weight matrices considered together, $w_\ell$ denotes the width at layer $\ell$ and $\sigma_1, \ldots, \sigma_L$ denote elementwise activation functions with Lipschitz constants $\rho_\ell$ for $\ell = 1, \ldots, L$. For any pair of layers indices $\ell_1 < \ell_2$ we also use the notation $F^{\ell_1 \to \ell_2}$ for the function defined by $F^{\ell_1 \to \ell_2}(x) = A_{\ell_2}\sigma_{\ell_2}(A^{\ell_2-1}\sigma_{\ell_2-1}(\ldots\sigma_{\ell_1}(A^{\ell_1}x)\ldots))$. The result below follows from Theorem E.2 in $\widetilde{O}$ notation:

**Theorem 3.2.** *Fix reference matrices $M_1, \ldots, M_L$. For every $\delta > 0$, w.p. $\geqslant 1 - \delta$, the following generalization bound holds simultaneously over all values of $p_\ell \in [0, 2]$ for $\ell = 1, \ldots, L$:*

$$\text{GAP} \leqslant \tilde{O}\left( \mathcal{B}\sqrt{\frac{\log(1/\delta)}{N}} + \mathcal{B}\sqrt{\frac{L^3}{N}} + \mathcal{B}[L\,\mathrm{L}_1]^{\frac{p}{2+p}}\sqrt{\frac{L}{N}}\mathcal{R}_{F_A} \right), \quad \text{where} \tag{3.7}$$

$$\mathcal{R}_{F_A} := \left[ \sum_{\ell=1}^{L} \left[ \mathfrak{B}\prod_{i=1}^{L}\rho_i\|A_i\| \right]^{\frac{2p_\ell}{p_\ell+2}} \times \left[ \frac{\|A_\ell - M_\ell\|_{\mathrm{sc},p_\ell}^{p_\ell}}{\|A_\ell\|^{p_\ell}} \right]^{\frac{2}{p_\ell+2}} [w_\ell + w_{\ell-1}]^{1+\frac{p_\ell}{p_\ell+2}} \right]^{\frac{1}{2}}$$

*and the $\tilde{O}$ absorbs polylog factors of $\mathfrak{B}, \overline{W}, L\max_i\|A_i\|, \mathrm{L}_1$ and $N$. Here, $p := \max_{\ell=1}^{L}p_\ell$ denotes the maximum of all the indices $p_\ell$ and $\mathfrak{B} := \sup_{i=1}^{N}\|x_i\|$ is the maximum input $L^2$ norm.*

Next, using Loss Augmentation [5, 6], we obtain the following extension where norm-based terms are replaced by milder analogues relying on empirical estimates of intermediary activations. Since the product of spectral norms estimate $\|x\|\prod_{i=1}^{\ell}\|A^i\|$ is usually large compared to the activation norm $\|F^{0\to\ell}(x)\|$, this can often lead to substantial numerical improvements, as seen in Section D.1.

**Theorem 3.3** (Cf. Theorem E.6). *Fix reference matrices $M_1, \ldots, M_L$. For every $\delta > 0$, w.p. $\geqslant 1 - \delta$, the following bound holds simultaneously over all values of $p_\ell \in [0, 2]$ for $\ell = 1, \ldots, L$:*

$$\text{GAP} \leqslant \tilde{O}\left( \mathcal{B}\sqrt{\frac{\log(1/\delta)}{N}} + \mathcal{B}\sqrt{\frac{L^3}{N}} + \mathcal{B}[L\,\mathrm{L}_1]^{\frac{p}{2+p}}\sqrt{\frac{L}{N}}\mathcal{R}_{F_A} \right), \tag{3.8}$$

*where $\mathfrak{B}_{\ell-1,A} := \max\left(\max_{i\leqslant N}\|F_{A^1,\ldots,A^{\ell-1}}(x_i)\|, 1\right)$,*

$$\mathcal{R}_{F_A}^{emp} := \left[ \sum_{\ell=1}^{L} \left[ \mathfrak{B}_{\ell-1,A}\prod_{i=\ell}^{L}\rho_i\|A_i\| \right]^{\frac{2p_\ell}{p_\ell+2}} \times \left[ \frac{\|A_\ell - M_\ell\|_{\mathrm{sc},p_\ell}^{p_\ell}}{\|A_\ell\|^{p_\ell}} \right]^{\frac{2}{p_\ell+2}} [w_\ell + w_{\ell-1}]^{1+\frac{p_\ell}{p_\ell+2}} \right]^{\frac{1}{2}},$$

*and the $\tilde{O}$ notation absorbs polylogarithmic factors[6] of $\mathfrak{B}_{\ell-1,A}, \overline{W}, L, \max_i\|A_i\|, \mathrm{L}_1$ and $N$. Here, $p := \max_{\ell=1}^{L}p_\ell$ is the maximum of index $p_\ell$ and $\mathfrak{B} := \sup_{i=1}^{N}\|x_i\|$ is the maximum input $L^2$ norm.*

Theorem 3.2 holds for all values of $p_\ell$ simultaneously: they can be optimized after training. For $p_\ell = 0 \quad \forall \ell$, Thm 3.3 yields a sample complexity of $\tilde{O}\left(L^3 + L\sum_{\ell=1}^{L}[w_\ell + w_{\ell-1}]\mathrm{rank}(A_\ell)\right)$. However, the additive term of $L^3$ can easily be removed with a simpler argument dedicated to the situation where $p_\ell = 0$ (cf. Thm. E.8). Thus, our results provide a parametric complexity estimate for neural networks whose weight matrices satisfy a low rank property: for a neural network with fixed width $\overline{W}$, the sample complexity is $\tilde{O}\left(\overline{W}L^2\underline{r}\right)$ rather than $\tilde{O}\left(\overline{W}^2L^2\right)$ in a pure parameter counting bound: when applying Corollary 3.3 (or Thm. E.8) to a network with fixed width $\overline{W}$, each weight matrix $A_\ell \in \mathbb{R}^{\overline{W}\times\overline{W}}$ only contributes $\tilde{O}(\overline{W}\underline{r})$ 'parameters' to the sample complexity, instead of the full $\overline{W}^2$. Theorem 3.2 further incorporates *approximate low rank structure*. Indeed, in equation (3.7), the indices $p_\ell$ interpolate between the parametric and non parametric regimes. When $p$ increases, the term $\frac{\|A_\ell\|_{\mathrm{sc},p_\ell}^{p_\ell}}{\|A\|^{p_\ell}}$ captures low-rank structure inherent in very small singular values which are not quite equal to zero. However, the threshold $p_\ell$ must be carefully tuned due to a tradeoff with the scaling factor of $\left[\mathfrak{B}\prod_{i=1}^{L}\rho_i\|A_i\|\right]^{\frac{2p_\ell}{p_\ell+2}}$. As a partial limitation, we note that the bound does not exactly coincide with the norm-based result (1.3) when $p$ is set to 2, since there is always a parametric dependence on the input dimension from the term $[\mathcal{C} + d]^{\frac{2}{p+2}}$: whilst the bound is an interpolation between norm-based and parameter counting bounds, it maintains a slight bias towards parameter counting. However, this is achieved without being uniformly inferior

---

[6]In this paper, our $\tilde{O}$ notation absorbs factors of $\max_{\ell=1}^{L}\|A_\ell\|$, which is stricter than the alternative of accepting polylogarithmic factors of $\prod_{\ell=1}^{L}\|A_i\|$. In particular, one of the multiplicative factors of $L$ in our bound arises from a logarithmic factor in $\prod_{\ell=1}^{L}\|A_\ell\|$. We find this approach more natural as it makes the full-rank parameter counting complexity $\tilde{O}(\mathcal{W}L) = \tilde{O}(\overline{W}L^2)$ match classic *lower-bounds* on VC dimension [7, 82].

to either. To see this, consider the one-layer case in an idealized situation where all the weights take binary values $\in \{1, -1\}$. Here, the norm-based bound (1.3) yields a sample complexity of $\widetilde{O}(\mathcal{C}d)$, which agrees with parameter-counting. Theorem 3.2 also yields a sample complexity of $\widetilde{O}\left(\mathcal{M}^{\frac{2p}{p+2}}\min(\mathcal{C},d)^{\frac{p}{2+p}}\max(\mathcal{C},d)^{\frac{2}{2+p}}\right) = \widetilde{O}\left(\sqrt{\mathcal{C}d}\|A\|_{\mathrm{Fr}}\right) = \widetilde{O}\left(\mathcal{C}d\right)$.

### 3.2.1 Proof Sketches

To highlight our proof techniques, we provide a proof sketch for simplified versions of our results.

**Proposition 3.4** (Simplified form of Proposition F.2: covering number bound for one layer)**.** *Consider the following function class of linear maps from $\mathbb{R}^d \to \mathbb{R}^m$: $\mathcal{F}^p := \{M \in \mathbb{R}^{m \times d}, \|M\|_p \leq \mathcal{M}; \|M\| \leq s\}$. For any $\epsilon > 0$ and for any dataset $x_1, \ldots, x_N \in \mathbb{R}^d$ such that $\|x_i\| \leq B$ for all $i$, we have the following bound on the $L^\infty$ covering number (cf Prop. F2 for formal definition) of the class:*

$$\log(\mathcal{N}_\infty(\mathcal{F}_r^p, \epsilon)) \lesssim [m+d] \left[\frac{\mathcal{M}B}{\epsilon}\right]^{\frac{2p}{p+2}} \log(\frac{mdN\mathcal{M}}{\epsilon}).$$

*Proof Sketch.* For simplicity we assume $B = 1$ in the sketch. For any matrix $M \in \mathcal{F}^p$, write $\sum_{i=1}^{\min(m,d)} \rho_i u_i v_i^\top$ for its singular value decomposition (with the singular values ordered in (any) decreasing order, including zeros). For a threshold $\tau$ to be determined later, we decompose every $M \in \mathcal{F}^p$ into $M = M_1 + M_2$ where $M_1 = \sum_{i \leq T} \rho_i u_i v_i^\top$ and $M_2 = \sum_{i \geq T+1} \rho_i u_i v_i^\top$ where $T$ is the last index such that $\rho_T > \tau$. By Markov's inequality, since $\|M\|_p^p = \sum_i \rho_i^p \leq \mathcal{M}^p$, we have $\mathrm{rank}(M_1) \leq \tau \leq \frac{\mathcal{M}^p}{\tau^p}$. In addition, the spectral norm of $M_1$ is bounded as follows: $\|M_1\| = \rho_1 = \|M\| \leq \|M\|_{\mathrm{sc},p} \leq \mathcal{M}$. Thus, $M_1$ belongs to the set $\mathcal{F}_1^p := \left\{Z \in \mathcal{F}^p : \mathrm{rank}(Z) \leq \frac{\mathcal{M}^p}{\tau^p}, \quad \|Z\| \leq \mathcal{M}\right\}$ (a function class with few parameters its members are *low-rank*). Furthermore, it is clear that

$$\|M_2\|_{\mathrm{Fr}}^2 = \sum_{k=T+1}^{\min(m,d)} \rho_k^2 \leq \tau^2 \min(m,d).$$

Therefore $M_2$ belongs to the class $\mathcal{F}_2^p := \{Z \in \mathcal{F}^p : \|Z\|_{\mathrm{Fr}}^2 \leq \tau^2 \min(m,d)\}$ (whose members have *small norms*). By a parameter counting argument, we can bound the $L^\infty$ covering number of $\mathcal{F}_1^p$:

$$\mathcal{N}_\infty(\mathcal{F}_1^p, \epsilon/2) \lesssim [m+d][\mathrm{rank}] \log(\frac{\mathcal{M}}{\epsilon}) \lesssim [m+d]\frac{\mathcal{M}^p}{\tau^p} \log(\frac{\mathcal{M}}{\epsilon}).$$

By classic norm-based arguments (Thm. 4 in [83]), we can bound the covering number of $\mathcal{F}_2^p$ as:

$$\log(\mathcal{N}(\mathcal{F}_2^p, \epsilon/2)) \lesssim [\text{max Frob norm}] \log(\frac{\tau^2 mdN}{\epsilon}) = \frac{\tau^2\min(m,d)}{\epsilon^2} \log(\frac{\tau^2 mdN}{\epsilon}).$$

Combining both bounds gives $\log(\mathcal{N}_\infty(\mathcal{F}_r^p)) \leq \log(\mathcal{N}(\mathcal{F}_1^p, \epsilon/2)) + \log(\mathcal{N}(\mathcal{F}_2^p, \epsilon/2))$

$$\lesssim \frac{\tau^2\min(m,d)}{\epsilon^2} \log(\frac{\tau^2 mdN}{\epsilon}) + [m+d]\frac{\mathcal{M}^p}{\tau^p} \log(\frac{\mathcal{M}}{\epsilon})$$

Setting $\tau = \mathcal{M}^{\frac{p}{p+2}} \epsilon^{\frac{2}{p+2}}$ gives $\log(\mathcal{N}_\infty(\mathcal{F}_r^p)) \lesssim [m+d]\left[\frac{\mathcal{M}}{\epsilon}\right]^{\frac{2p}{p+2}} \log(\frac{mdN\mathcal{M}}{\epsilon})$ as expected.

$\square$

We then show how to extend the proof to the two-layer case. Here, we consider networks of the form $F_A : x \to A^2\sigma(A^1 x)$ where $\sigma$ is an elementwise 1-Lipschitz activation function (e.g. ReLU) and $A_1 \in \mathbb{R}^{m \times d}, A^2 \in \mathbb{R}^{C \times m}$ are weight matrices. We fix $p_1 = p_2 = p \in [0, 2]$ and $s_1, s_2, \mathcal{M}_1, \mathcal{M}_2 > 0$, and let $\mathcal{F}$ denote the class of such networks which further satisfy for all $i \in \{1, 2\}$: $\|A^i\| \leq s_i$; $\|A^i\|_p \leq \mathcal{M}_i$ (or, by abuse of notation, the corresponding matrices $(A^2, A^1)$). For simplicity, the loss $\ell : \mathbb{R}^C \times [C] \to \mathbb{R}^+$ is assumed to be 1-Lipschitz w.r.t. the $L^\infty$ norm.

**Proposition 3.5** (Simplified form of Proposition E.1). *Suppose we are given an i.i.d. training set $(x_1, y_1), \ldots, (x_N, y_N)$ from a joint distribution over $\mathbb{R}^d \times [C]$ such that $\|x_i\| \leqslant B$ w.p. 1. For any $\delta > 0$, w.p. $\geqslant 1 - \delta$, the generalization gap of any network $F_A \in \mathcal{F}$ is bounded by:*

$$\tilde{O}\left( [s_1 s_2]^{\frac{p}{2+p}} \left[ [r_1[m+d]^{1+\frac{p}{p+2}} + r_2[m+C] \right]^{\frac{1}{2}} \frac{1}{\sqrt{N}} + \sqrt{\frac{\log(1/\delta)}{N}} \right), \qquad (3.9)$$

*where for $i = 1, 2$, $r_i := \left[ \frac{\mathcal{M}_i^p}{s_i^p} \right]^{\frac{2}{p+2}}$ is a soft analogue of the rank and the $\tilde{O}$ notation incorporates polylogarithmic factors of $N, m, d, C, s_1, s_2, \mathcal{M}_1, \mathcal{M}_2$.*

*Proof Sketch.* Since the loss function is 1 Lipschitz, we only need to find an $L^\infty$ cover of $\mathcal{F}$, i.e., a set $\mathcal{C} \subset \mathcal{F}$ such that for any $A = (A^1, A^2) \in \mathcal{F}$, there exists a cover element $(\bar{A}^1, \bar{A}^2) \in \mathcal{C}$ such that

$$\forall i, \|F_A(x_i) - F_{\bar{A}}(x_i)\|_\infty \leqslant \epsilon$$

(here, recall $F_A(x_i) \in \mathbb{R}^C$ is a vector of scores for all classes). We achieve this by adapting standard chaining techniques (see [3,5]), with the caveat that we need to change the norm of the cover at the intermediary layer, incurring a factor of $\sqrt{m}$. More precisely, let $\mathcal{F}^1, \mathcal{F}^2$ denote the set of matrices $A^1 \in \mathcal{R}^{m \times d}$ and $A^2 \in \mathcal{R}^{C \times m}$ satisfying the relevant constraints above. First, we can apply Proposition 3.4 above, with $\epsilon$ set to $\epsilon_1 := \epsilon/[2\sqrt{m}s_2]$, to achieve a $\mathcal{C}_1 \subset \mathcal{F}_1$ such that for all $A \in \mathcal{F}$ there exists a $\bar{A} \in \mathcal{C}$ such that $\forall i, \|\sigma(A^1(x_i)) - \sigma(\bar{A}^1(x_i))\| \leqslant \|A^1 x_i - \bar{A}^1 x_i\| \leqslant$

$\sqrt{m}\| \leqslant \|A^1 x_i - \bar{A}^1 x_i\| \leqslant \sqrt{m}\epsilon_1 = \frac{\epsilon}{2s_2}$, and $\log(|\mathcal{C}|) \in \tilde{O}\left( [m+d]\left[ \frac{\mathcal{M}_1 B}{\epsilon_1} \right]^{\frac{2p}{p+2}} \right) = \tilde{O}\left( [m + \right.$

$\left. d] m^{\frac{p}{p+2}} [s_1 s_2 B]^{\frac{2p}{p+2}} \left[ \frac{\mathcal{M}_1}{s_2 \epsilon} \right]^{\frac{2p}{p+2}} \right)$. Similarly, for any $\bar{A}^1 \in \mathcal{C}_1$, $\|\sigma(\bar{A}^1 x_i)\| \leqslant s_1 B \quad \forall i$, and therefore

another application of Prop. 3.4 with $B$ replaced by $\tilde{B}s_1 B$ and $\epsilon$ replaced by $\epsilon_2 := \epsilon/2$ yields a cover $\mathcal{C}_2(\bar{A}^1)$ with size $\log(|\mathcal{C}_2(\bar{A}^1)|) \in \tilde{O}\left( \left[ \frac{\mathcal{M}_2[Bs_1]}{\epsilon_2} \right]^{\frac{2p}{p+2}} \right) = \tilde{O}\left( [Bs_1 s_2]^{\frac{2p}{p+2}} \left[ \frac{\mathcal{M}_2}{\epsilon s_2} \right]^{\frac{2p}{p+2}} \right)$.

The cover $\mathcal{C} := \bigcup_{\bar{A}^1 \in \mathcal{C}_1} \mathcal{C}_2(\bar{A}^1)$ is now an $\epsilon$ cover of $\mathcal{F}$. Indeed, for any $(A^1, A^2) \in \mathcal{F}$, we can define $\bar{A}^1$ to be the cover element associated to $A^1$ in $\mathcal{C}_1$ and subsequently $\bar{A}^2$ to be the cover element in $\mathcal{C}_1(\bar{A}^1)$ associated to $A^2$, which yields for any $i \leqslant N$:

$$\|A^2 \sigma(A^1 x_i) - \bar{A}^2 \sigma(\bar{A}^1 x_i)\|_\infty \leqslant \|A^2 \sigma(A^1 x_i) - A^2 \sigma(\bar{A}^1 x_i)\|_\infty + \|A^2 \sigma(\bar{A}^1 x_i) - \bar{A}^2 \sigma(\bar{A}^1 x_i)\|_\infty$$

$$\leqslant \epsilon_2 + \|A^2\| \|\sigma(A^1(x_i)) - \sigma(\bar{A}^1(x_i))\| \leqslant \epsilon/2 + s_2 \epsilon/2s_2 = \epsilon.$$

Furthermore, the resulting cover has cardinality bounded as

$$\log(|\mathcal{C}|) \in \tilde{O}\left( \frac{[Bs_1 s_2]^{\frac{2p}{p+2}}}{\epsilon} \left[ [m+d]^{1+\frac{p}{p+2}} \left[ \frac{\mathcal{M}_1}{\epsilon s_1} \right]^{\frac{2p}{p+2}} + [m+C] \left[ \frac{\mathcal{M}_2}{\epsilon s_2} \right]^{\frac{2p}{p+2}} \right] \right),$$

which yields eq (3.9) after substituting for $r_i$ and calculations based on Dudley's entropy integral. □

Thm E.2 is then derived from a union bound to ensure uniformity over $\mathcal{M}_1, \mathcal{M}_2$ etc. This step is standard for $s_1, s_2, \mathcal{M}_1, \mathcal{M}_2$ but involves a more tedious continuity argument for the parameter $p$.

### 3.3 Convolutional Neural Networks (CNNs)

In this section, we present an extension of our results to Convolutional Neural Networks (CNNs). Since CNNs are a well-known class of models, we leave a more thorough description of our notation for the forward pass to Appendix G. We denote the filters of each layer by $A_\ell$ (which incorporates each weight only once), whist we use $\text{op}(A_\ell)$ to denote matricization of the layer, i.e., the matrix which represents the linear operation performed by the convolution at layer $\ell$ with the weights $A_\ell$. We use $U_\ell$ for the number of channels and $d_\ell$ to refer to the dimension of the input patches. This includes both spatial and channel dimensions: if the input is an RGB image and the filter has spatial dimension $(2, 2)$, $d_0 = 2 \times 2 \times 3 = 12$. Thus, the number of trainable parameters at layer $\ell$ is $d_{\ell-1} \times U_\ell$, and write $w_\ell$ for the spatial dimension at layer $\ell$ (for instance, a $28 \times 28$ image will have $w_0 = 28 \times 28 = 784$). Thus, $U_\ell \times w_\ell$ is the number of preactivations at layer $\ell$.

**Theorem 3.6.** *Fix reference matrices $M_1, \ldots, M_L$. For every $\delta \in (0,1)$, w.p. $\geqslant 1 - \delta$, the following generalization bound holds simultaneously over all values of $0 \leqslant p_\ell \leqslant 2$ for $\ell = 1, \ldots, L$:*

$$\text{GAP} \leqslant \tilde{O}\left( \mathcal{B}\sqrt{\frac{\log(1/\delta)}{N}} + \mathcal{B}\sqrt{\frac{L^3}{N}} + \mathcal{B}[L\,\mathrm{L_l}]^{\frac{p}{2+p}}\sqrt{\frac{L}{N}}\mathcal{R}_{F_A}^{\mathfrak{C}} \right), \quad \text{where} \quad (3.10)$$

$$\mathcal{R}_{F_A}^{\mathfrak{C}} := \left[ \sum_{\ell=1}^{L} \left[ \mathfrak{B}\prod_{i=1}^{L} \rho_i \| \mathrm{op}(A_i)\| \right]^{\frac{2p_\ell}{p_\ell+2}} \times \left[ \frac{\|A_\ell - M_\ell\|_{\mathrm{sc},p_\ell}^{p_\ell}}{\|\mathrm{op}(A_\ell)\|^{p_\ell}} \right]^{\frac{2}{p_\ell+2}} [U_\ell + d_{\ell-1}]W_\ell^{\frac{p_\ell}{p_\ell+2}} \right]^{\frac{1}{2}},$$

$\mathfrak{B} := \sup_{i=1}^{N}\|x_i\|$ *and the $\tilde{O}$ notation absorbs multiplicative factors of logarithms of the quantities $\mathfrak{B}, \overline{W}, \mathcal{A}, \mathcal{W}, L, \max_i \|\mathrm{op}(A_i)\|, \mathrm{L_l}$ and $N$. Here, $W_\ell := U_\ell \times w_\ell$ (for $\ell \neq L$) and $W_L := 1$.*

This result incorporates weight sharing in the same sense as other CNN bounds [4, 16]; the term $\|A_\ell - M_\ell\|_{\mathrm{sc},p_\ell}^{p_\ell}$ only involves the weights once, irrespective of the number of patches.

**Theorem 3.7** (Cf. Theorem G.10). *For every $\delta \in (0,1)$, with probability greater than $1 - \delta$, the following generalization bound holds simultaneously over all values of $0 \leqslant p_\ell \leqslant 2$ for $\ell = 1, \ldots, L$:*

$$\text{GAP} \leqslant \tilde{O}\left( \mathcal{B}\sqrt{\frac{\log(1/\delta)}{N}} + \mathcal{B}\sqrt{\frac{L^3}{N}} + \mathcal{B}[L\,\mathrm{L_l}]^{\frac{p}{2+p}}\sqrt{\frac{L}{N}}\mathcal{R}_{F_A}^{emp,\mathfrak{C}} \right), \quad \text{where}$$

$$\mathcal{R}_{F_A}^{emp,\mathfrak{C}} := \left[ \sum_{\ell=1}^{L} \left[ \mathfrak{B}_{\ell-1,A}^{\mathfrak{C}}\prod_{i=\ell}^{L} \rho_i \| \mathrm{op}(A_i)\| \right]^{\frac{2p_\ell}{p_\ell+2}} \times \left[ \frac{\|A_\ell - M_\ell\|_{\mathrm{sc},p_\ell}^{p_\ell}}{\|\mathrm{op}(A_\ell)\|^{p_\ell}} \right]^{\frac{2}{p_\ell+2}} [U_\ell + d_{\ell-1}]W_\ell^{\frac{p_\ell}{p_\ell+2}} \right]^{\frac{1}{2}},$$

$\mathfrak{B}_{\ell-1,A}^{\mathfrak{C}} := \max\left(\max_{i \leqslant N, o} \|[F_{A^1,\ldots,A^{\ell-1}}(x_i)]_{S_{\ell-1,o}}\|, 1\right)$ *denotes the maximum norm of a convolutional patch at layer $\ell - 1$ and the $\tilde{O}$ notation absorbs multiplicative factors of logarithms of the quantities $\mathfrak{B}, \overline{W}, \mathcal{A}, \mathcal{W}, L, \max_i \|\mathrm{op}(A_i)\|, \mathrm{L_l}, N$. Here, $W_\ell := U_\ell \times w_\ell$ (for $\ell \neq L$) and $W_L := 1$.*

Compared to [4], Thm. 3.7 incorporates a low-rank structure: each layer's contribution to sample complexity is reduced from $L \times U_\ell \times d_\ell$ (the number of parameters in the layer) to $L[U_\ell + d_\ell]\mathrm{rank}(A_\ell)$ ($U_\ell$ is the number of channels at layer $\ell$ and $d_{\ell-1}$ is the dimension of patches at layer $\ell - 1$).

## 4 Conclusion and Future Directions

For linear networks, deep neural networks and convolutional neural networks, we have shown generalization bounds which capture the implications of approximate low-rank structure in weight matrices in terms of sample complexity. The bounds are expressed in terms of the Schatten $p$ quasi norms of the weight matrices, and are, to the best of our knowledge, the first bounds for deep learning which incorporate both norm-based and parameter counting qualities. When $p \to 0$, the bounds behave like a parameter-counting sample complexity of $\tilde{O}\left( L^3 + L\sum_{\ell=1}^{L}[w_\ell + w_{\ell-1}]\mathrm{rank}(A_\ell) \right)$, taking into account (exact) low-rank structure in the weight matrices. For more moderate values of $p$, the bounds incorporate norm-based quantities and incorporate *approximate* low-rank structure. Even in the case of linear maps, Subsection 3.1 provides original bounds for multi-class classification in the presence of *low-rank structure over the classes*, offering an alternative to classic results such as [65].

A limitation of our work is that we require the *weight matrices* to be low-rank, whereas most research on neural rank collapse [21, 27] demonstrates instead that the *activations* at each layer are low-rank. The two are very closely connected, but are not completely equivalent: it is possible that capturing the effect of low-rank *activations* could lead to further improvements and insights. Crucially, we believe a substantial modification of our proofs could tackle such situations. However, this would require a much more challenging network-wide approach to the simultaneous tuning of the parametric interpolation thresholds and is left to future work. Like other uniform convergence results which are agnostic to the training procedure, our bounds are vacuous at for large-scale networks in the absence of aggressive bound optimization, a limitation which may be addressed in future work by better controlling the norm-based components during training, manually truncating the ranks or incorporating data-dependent priors with a Bayesian approach. Lastly, we note that the low-rank assumption requires overparametrization and is not always satisfied for smaller architectures or layers (cf. Appendix D). Finally, other tantalizing future directions include proving fast rates in $N$ and the extension of our work to other architectures such as Resnets and transformers.

## Acknowledgements

Rodrigo Alves thanks Recombee for their support. The work of Yunwen Lei is partially supported by the Research Grants Council of Hong Kong [Project Nos. 17302624]. We thank the anonymous reviewers, Felix Biggs and Nadav Timor for their comments on our manuscript.

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

# A    Table of Notation

| Notation | Meaning |
|---|---|
| $\|\cdot\|, \|\cdot\|_\sigma$ | Spectral norm of a matrix |
| $\|\cdot\|, \|\cdot\|_2$ | (L2) Norm of a vector |
| $\|\cdot\|_{\mathrm{Fr}}$ | Frobenius norm of a matrix |
| $\|\cdot\|_{\mathrm{sc},p}$ | Schatten $p$ quasi norm of a matrix |
| $\|\cdot\|_*$ | Nuclear norm |
| $\|Z\|_{2,1}$ | $\sum_i \sqrt{\sum_j Z_{i,j}^2}$ |
| $\mathcal{C}$ | Number of classes |
| l | Loss function |
| $\mathrm{L_l}$ | Lipschitz constant of l (w.r.t. $L^\infty$ norm) |
| $\mathcal{B}$ | upper bound on l |
| $A_\ell \in \mathbb{R}^{w_\ell \times w_{\ell-1}}$ | Weight matrix at layer $\ell$ |
| $M_\ell \in \mathbb{R}^{w_\ell \times w_{\ell-1}}$ | Reference matrix at layer $\ell$ |
| $\sigma_\ell$ | (Elementwise) Activation function at layer $\ell$ |
| $\rho_\ell$ | Lipschitz constant of $\sigma_\ell$ |
| $F_A(x) := A_L\sigma_L(A^{L-1}\sigma_{L-1}(\ldots\sigma_1(A^1 x)\ldots)$ | NN with weight matrices $A_1,\ldots,A_L$ (Cf. Equation (3.6)) |
| $F_A^{\ell_1 \to \ell_2}(\tilde{x})$ | $\sigma_{\ell_2}(A_{\ell_2}\sigma_{\ell_2-1}(\ldots\sigma_{\ell+1}(A_{\ell+1}\tilde{x})\ldots))$ Map from layer $\ell_1$ to layer $\ell_2$ |
| **Architectural Quantities** | |
| $L$ | Depth |
| $N$ | Number of samples |
| $w_\ell$ | Width at layer $\ell$ |
| $\overline{W}$ | $\max_{\ell=0}^L w_\ell$ (Maximum width of the network) |
| $\underline{w}_\ell$ | $\min(w_\ell, w_{\ell-1})$ |
| $\bar{w}_\ell$ | $[w_\ell + w_{\ell-1}]$ |
| $\tilde{w}_\ell$ | $[w_\ell \underline{w}_\ell]$   if   $\ell \neq L$   $\underline{w}_L$   if   $\ell = L$ |
| $d$ | input dimension |
| **Function Classes** | |
| $\mathcal{F}_{\mathcal{M}_2}^2$ | $\{Z \in \mathbb{R}^{m \times d} : \|Z\|_{\mathrm{Fr}} \leqslant \mathcal{M}\}$ |
| $\mathcal{F}_{\mathrm{r}}^{\mathrm{p}}$ | $\{Z \in \mathbb{R}^{m \times d} : \|Z\|_{\mathrm{sc},p}^p \leqslant \mathcal{M}^{\mathrm{p}} = \mathrm{r}\,\mathrm{s}^p;$ $\|Z\| \leqslant \mathrm{s}\}$ |
| $\mathcal{E}_{\mathrm{r,s}}$ | $\{A \in \mathbb{R}^{m \times d} : \mathrm{rank}(A) \leqslant r, \|A\| \leqslant s\}$ |
| $\mathcal{B}_\ell$ | $\{A_\ell \in \mathbb{R}^{w_\ell \times w_{\ell-1}} : \|A_\ell - M_\ell\|_{\mathrm{sc},p_\ell} \leqslant \mathcal{M}_\ell; \|A_\ell\| \leqslant s_\ell\}$ |
| $\mathcal{F}_\mathcal{B}$ | $\{F_A = F_{(A_1,\ldots,A_L)} : A_\ell \in \mathcal{B}_\ell \quad \forall \ell \leqslant L\}$ (cf. Eq. (E.1)) $= \{F_A = F_{(A_1,\ldots,A_L)} : \|A_\ell - M_\ell\|_{\mathrm{sc}} \leqslant \mathcal{M}_\ell \|A\| \leqslant s_\ell \quad \forall \ell \leqslant L\}$ |
| $\mathcal{L}^{\lambda_{aug}}$ | $\{\lambda_{aug}(F_A, x, y) : F_A \in \mathcal{F}_\mathcal{B}\}$ |
| $\mathcal{L}^{\mathfrak{C},\lambda_{aug}}$ | $\{\lambda_{aug}(F_A, x, y) : F_A \in \mathcal{F}_\mathcal{B}^\mathfrak{C}\}$ |
| **Constraints/scaling factors** | |
| $p_\ell$ | Schatten index for layer $\ell$ |
| $p = \max_\ell p_\ell$ | Maximum Schatten index over all layers |

| | |
|---|---|
| s | Upper bound on $\|Z\|$ |
| $s_\ell$ | Upper bound on $\|A_\ell\|$ (fully connected) |
| $\mathcal{M}$ | Upper bound on $\|Z\|_{\mathrm{sc},p}$ |
| $\mathcal{M}_\ell$ | Upper bound on $\|A - M_\ell\|_{\mathrm{sc},p_\ell}$ |
| r | $\frac{\mathcal{M}^p}{\mathrm{s}^p}$ (rank proxy) |
| $r_\ell$ | $\frac{\mathcal{M}_\ell{}^p}{s_\ell{}^p}$ (rank proxy) |
| $\mathfrak{B}$ | $\sup_{i=1}^N \|x_i\|$ |
| $\tilde{r}_\ell$ | a priori upper bound on $\mathrm{rank}(A_\ell)$ |
| $\mathfrak{B}_{\ell-1,A}$ | $\max\left(\max_{i\leqslant N}\|F_{A^1,\ldots,A^{\ell-1}}(x_i)\|,1\right)$ |

**Constants/log factors**

| | |
|---|---|
| $\Gamma_{\mathcal{F}_r^p}$ | $\left(\frac{16[\mathcal{M}^p+1][b+1][1+s][m+d]}{\epsilon^{\frac{p}{p+2}}}+7\right)mN$ |
| $\Gamma'_{\mathcal{F}_r^p}$ | $\left(\frac{16[\mathcal{M}^p+1][b+1][1+s][m+d]^2}{\epsilon^{\frac{p}{p+2}}}+7\right)mN$ |
| $\Gamma_{\mathcal{F}_r^p,\ell}$ | $\left[\left(\frac{16[\mathcal{M}_\ell{}^{p_\ell}+1][b\prod_{i=1}^\ell \rho_i s_i+1][1+s_\ell][w_\ell+w_{\ell-1}]^2}{\epsilon_\ell^{\frac{p_\ell}{p_\ell+2}}}+7\right)w_\ell N\right]$ |
| $\Gamma_{\mathcal{F}_r^p,L}$ | $\left[\left(\frac{16[\mathcal{M}_L{}^{p_L}+1][b\prod_{i=1}^L \rho_i s_i+1][1+\sqrt{w_L}\,s_L][w_L+w_{L-1}]}{\epsilon_L^{\frac{p_L}{p_L+2}}}+7\right)w_L N\right]$ |
| $\overline{\Gamma}$ | $\left[128\prod_{i=1}^L[[\rho_i+1]s_i+1]^3[b+1][\mathrm{L}_l\,L+1]\overline{W}^4 N^2\right]$ |
| $\underline{\Gamma}$ | $\left[[b+1][L\,\mathrm{L}_l+1]\overline{W}N\prod_{i=1}^L[[\rho_i+1]s_i+1]\right]$ |
| | ($\overline{\Gamma}=40\underline{\Gamma}^6$, cf. eq (E.141)) |
| $\gamma_{FC}$ | $12\left[[b+1][L\,\mathrm{L}_l+1]\overline{W}N\prod_{i=1}^L[\rho_i\|A_i\|+1]\right]\left[4\overline{W}+b\prod_i \rho_i\|A_i\|\right]$ |
| $\Theta_{\log}$ | $2\log\left[4\overline{W}+b\prod_i \rho_i\|A_i\|\right]\left[|\log(b)|+L\left[\sum_{\ell=1}^L|\log(\|A_\ell\|)|+2\log(4\overline{W})\right]\right]$ |
| $\Gamma_R$ | $[\mathfrak{B}+2][L\,\mathrm{L}_l+1]\overline{W}N\prod_{i=1}^L[[\rho_i+1][\|A_i\|+2]+1]$ |
| $\Theta_{\log}^{\lambda_{aug}}$ | $2\log\left[4\overline{W}+b\prod_i \rho_i\|A_\ell\|\right]\times$ $\left[|\log(b)|+L\left[\sum_{\ell=1}^L|\log(\|A_\ell\|)|+\sum_{\ell=1}^{L-1}\log(\mathfrak{B}_{\ell,A})+2\log(4\overline{W})\right]\right]$ |

**Convolutional Neural Networks**

| | |
|---|---|
| $O_\ell$ | Number of convolutional patches at layer $\ell$ |
| $w_\ell$ | Spatial dimension at layer $\ell$ |
| $O_{\ell-1}$ | Number of spatial dimensions before pooling at layer $\ell$ |
| $U_\ell$ | Number of channel dimensions at layer $\ell$ |
| $d_\ell$ | Dimension of input patches at layer $\ell$ |
| $S^{\ell,o}\in[U_\ell]\times[w_\ell]$ | $o$th Convolutional Patch at layer $\ell$ |
| $A_\ell\in\mathbb{R}^{U_\ell\times d_\ell}$ | Weight matrix at layer $\ell$ |
| $M_\ell\in\mathbb{R}^{U_\ell\times d_\ell}$ | Initialized weight matrix at layer $\ell$ |
| $\Lambda_{A^\ell}$ | Convolution operation associated to $A_\ell$ Represented by the matrix $\mathrm{op}(A_\ell)$ $:\mathbb{R}^{U_\ell\times O_\ell}\to\mathbb{R}^{U_\ell\times w_\ell}$ |
| $\sigma_\ell$ | Activation function (including pooling) |
| $F_{A^1,A^2,\ldots,A^L}^{\mathfrak{C}}$ | $(\sigma_L\circ\Lambda_{A^L}\circ\sigma_{L-1}\circ\Lambda_{A^{L-1}}\circ\ldots\sigma_1\circ\Lambda_{A^1})(x)$ (Cf. Equation (G.1)) |

| | |
|---|---|
| $\mathcal{A}$ | $\sum_{\ell=0}^{L} U_\ell \times O_{\ell-1}$ 
 (Total number of preactivations) |
| $\mathcal{W}$ | $\sum_{\ell=1}^{L} d_{\ell-1} \times U_\ell$ 
 (Total number of parameters in the network) |
| $\mathcal{F}_{\mathcal{B}}^{\mathfrak{C}}$ | $\left\{ F_A = F_{(A_1,\ldots,A_L)}^{\mathfrak{C}} : \|A_\ell - M_\ell\|_{\mathrm{sc}} \leqslant \mathcal{M}_\ell, \right.$ 

 $\left. \|\mathrm{op}(A_\ell)\| \leqslant s_\ell \quad \forall \ell < L, \ \|A_L^\top\|_{2,\infty} \leqslant s_L \right\}$ |
| $\underline{w}_\ell$ | Spatial patch size at layer $\ell$ |
| $\underline{B}$ | $\max_{o,i} \|(x_i)_{S^{0,o}}\|$ 
 Maximum norm of a convolutional patch at input layer |
| **Constants/log factors for CNNs** | |
| $\gamma_{\mathfrak{C}}$ | $12 \left[ [\mathfrak{B}+1][L\,L_1+1] \mathcal{W}\mathcal{A}N \prod_{i=1}^{L} [\rho_i \|\mathrm{op}(A_i)\| + 1] \right]$ 
 $\times \left[ 4\overline{W} + \mathfrak{B} \prod_i \rho_i \|\mathrm{op}(A_i)\| \right]$ |
| $\gamma_{\mathfrak{C},\lambda_{aug}}$ | $12 \left[ \left[ [\mathfrak{B}+1][L\,L_1+1] \mathcal{W}\mathcal{A}N \prod_{i=1}^{L} [\rho_i \|\mathrm{op}(A_i)\| + 1] \right] \right] \times$ 
 $\left[ 4\overline{W} + \mathfrak{B} \prod_i \rho_i \|\mathrm{op}(A_i)\| \right]$ |
| $\Gamma_{\mathfrak{C}_r^p}$ | $\left( \dfrac{16[\mathcal{M}^p+1][b+1][1+s][U'+d]}{\epsilon^{\frac{p}{p+2}}} + 7 \right) U'NO$ |
| $\Gamma_{\mathcal{F}_r^p,\ell}^{\mathfrak{C}}$ | $\left[ \left( \dfrac{16[\mathcal{M}_\ell^{p_\ell}+1][b\prod_{i=1}^{\ell}\rho_i s_i + 1][1+\sqrt{\mathcal{A}}s_\ell]\mathcal{W}\mathcal{A}}{\epsilon_\ell^{\frac{p_\ell}{p_\ell+2}}} + 7 \right) N\mathcal{A} \right]$ |
| $\Gamma_{\mathfrak{C}_r^p}'$ | $\left( \dfrac{16[\mathcal{M}^p+1][b+1][1+s][U'+d]U'w'}{\epsilon^{\frac{p}{p+2}}} + 7 \right) U'NO$ |
| $\Gamma_R^{\mathfrak{C}}$ | $\left[ [\mathfrak{B}+2][L\,L_1+1]\mathcal{W}\mathcal{A}N \prod_{i=1}^{L} [[\rho_i+1][\|A_i\|+2]+1] \right]$ |
| $\underline{\Gamma}^{\mathfrak{C},\lambda_{aug}}$ | $[b+1][L(L_1+1))\mathcal{W}\mathcal{A}N \prod_{i=1}^{L} [[\rho_i+1]s_i+1]$ |

Table 1: Table of notations for quick reference

# B  Table of Comparison to Existing Literature

| **Linear Classification Linear Networks** | | |
|---|---|---|
| Proof Technique | Bound (excluding polylog factors) | Reference |
| Parameter Counting | $\mathrm{L_l}\sqrt{\frac{\mathcal{C}d}{N}}$ | [4] |
| Norm Based | $\mathrm{L_l}\,\mathfrak{B}\sqrt{\mathcal{C}\frac{\|A\|_{\mathrm{Fr}}^2}{N}}$ | [66], [67], [84], [64] |
| Norm-based/peeling from Linear Networks | $\mathrm{L_l}\,\mathfrak{B}\sqrt{\frac{\mathcal{C}\min_\ell(\mathrm{Rank}(B_\ell))}{N}}\prod_\ell\|B_\ell\|$ | [64] |
| Norm-based | $\mathrm{L_l}\,\mathfrak{B}\sqrt{\frac{\|A\|_{\mathrm{Fr}}^2}{N}}$ | [65] |
| Norm-based | $\mathrm{L_l}\,\mathfrak{B}L^{\frac{3}{2}}\|A\|^{\frac{L-1}{L}}\sqrt{\frac{\mathcal{C}\|A\|_{\mathrm{sc},\frac{2}{L}}^{\frac{2}{L}}}{N}}$ | Deduced from [3] or [1] cf. Corollary C.1 |
| Parametric Interpolation | $\mathcal{B}\,\mathrm{L_l}^{\frac{p}{2+p}}\sqrt{\frac{\mathfrak{B}^{\frac{2p}{2+p}}\|A\|_{\mathrm{sc},p}^{\frac{2p}{2+p}}[\mathcal{C}+d]}{N}}$ | cf. Theorem 3.1 |
| | $\mathcal{B}\sqrt{\frac{\mathrm{rank}(A)[\mathcal{C}+d]}{N}}$ | $p=0$ in above line |
| **DNNs, norm-based** | | |
| Peeling | $\mathrm{L_l}\sqrt{\frac{\mathfrak{B}^2 L\prod_{\ell=1}^L\|A_\ell\|_{\mathrm{Fr}}^2}{N}}$ | [13] |
| Peeling | $\dfrac{\mathrm{L_l}\,\mathfrak{B}\max_\ell\frac{\|A_\ell^\top\|_{2,1}}{\|A_\ell\|}\left[\prod_{\ell=1}^L\|A_\ell\|\right]}{N^{\frac{1}{2+3p}}}\left[\frac{L}{p}\right]^{\frac{2}{3}+p}$ | [13] |
| Peeling | $\mathrm{L_l}\dfrac{C_1\sqrt{w_1}\frac{\|A_1\|_{\mathrm{Fr}}}{\|A_1\|}+\sum_{\ell=2}^L C_1^{L-\ell}C_2\sqrt{w_\ell\bar{r}_\ell}\frac{\|A_\ell\|_{\mathrm{Fr}}}{\|A_\ell\|}}{\sqrt{N}}$ $\times\prod_{\ell=1}^L\|A_\ell\|\mathfrak{B}$ | [64] |
| (Peeling) | $\mathrm{L_l}\,C_1^L\mathfrak{B}\left[\prod_{\ell=1}^L\|A_\ell\|\right]L\underline{r}\sqrt{\frac{\overline{W}}{N}}$ | Particular case of above |
| Pac Bayes | $\mathrm{L_l}\dfrac{L\sqrt{\overline{W}}}{\sqrt{N}}\left(\prod_{\ell=1}^L\|A_\ell\|\right)\left(\sum_{\ell=1}^L\frac{\|A_\ell\|_{\mathrm{Fr}}^2}{\|A_\ell\|_\sigma^2}\right)^{\frac{1}{2}}$ | [3] |
| Covering Numbers | $\mathrm{L_l}\dfrac{1}{\sqrt{N}}\prod_{\ell=1}^L\|A_\ell\|\left(\sum_{\ell=1}^L\frac{\|(A_\ell-M_\ell)^\top\|_{2,1}^{\frac{2}{3}}}{\|A_\ell\|^{\frac{2}{3}}}\right)^{\frac{3}{2}}$ | [1] |
| **DNNs, Parameter Counting** | | |
| | $\mathcal{B}\sqrt{\frac{\mathcal{W}\mathcal{S}L}{N}}$ | [4] |
| | $\mathcal{B}\sqrt{\frac{\mathcal{W}L}{N}}$ | [74] |
| | $\mathcal{B}\sqrt{\frac{L\left[\sum_{\ell=1}^L[w_\ell+w_{\ell-1}]\mathrm{rank}(A_\ell)\right]}{N}}$ | Theorem E.8 |

| | | |
|---|---|---|
| **DNNs**
**Parametric Interpolation** | | |
| | $$\mathcal{B}[L\,\mathrm{L_l}]^{\frac{p}{2+p}}\sqrt{\frac{L}{N}}$$ $$\left[\sum_{\ell=1}^{L}\left[\mathfrak{B}\prod_{i=1}^{L}\rho_i\|A_i\|\right]^{\frac{2p_\ell}{p_\ell+2}}\times\right.$$ $$\left.\left[\frac{\|A_\ell-M_\ell\|_{\mathrm{sc},p_\ell}^{p_\ell}}{\|A_\ell\|^{p_\ell}}\right]^{\frac{2}{p_\ell+2}}[w_\ell+w_{\ell-1}]^{1+\frac{p_\ell}{p_\ell+2}}\right]^{\frac{1}{2}}$$ | Theorem 3.2 |
| Parametric Interpolation
& Loss augmentation | $$\mathcal{B}[L\,\mathrm{L_l}]^{\frac{p}{2+p}}$$ $$\left[\sum_{\ell=1}^{L}\left[\mathfrak{B}_{\ell-1,A}\prod_{i=\ell}^{L}\rho_i\|A_i\|\right]^{\frac{2p_\ell}{p_\ell+2}}\times\right.$$ $$\left.\left[\frac{\|A_\ell-M_\ell\|_{\mathrm{sc},p_\ell}^{p_\ell}}{\|A_\ell\|^{p_\ell}}\right]^{\frac{2}{p_\ell+2}}[w_\ell+w_{\ell-1}]^{1+\frac{p_\ell}{p_\ell+2}}\right]^{\frac{1}{2}}$$ | Theorem 3.3 |
| **CNNs,**
**Parameter Counting** | | |
| | $$\mathcal{B}\sqrt{\frac{\mathcal{W}\mathcal{S}L}{N}}$$ | [4] |
| | $$\mathcal{B}\sqrt{\frac{\mathcal{W}L}{N}}$$ | [74] |
| | $$\mathcal{B}\sqrt{\frac{L\left[\sum_{\ell=1}^{L}[U_\ell+d_{\ell-1}]\mathrm{rank}(A_\ell)\right]}{N}}$$ | Corollary G.6 |
| **CNNs,**
**Norm-based** | | |
| Covering Numbers | $$\frac{\mathrm{L_l}\prod_{\ell=1}^{L}\|\mathrm{op}(A_\ell)\|}{\sqrt{N}}\left[\sum_{\ell=1}^{L}\left(\frac{U_\ell w_\ell\|[A_\ell-M_\ell]^\top\|_{2,1}}{\|\mathrm{op}(A_\ell)\|}\right)^{\frac{2}{3}}\right]^{\frac{3}{2}}$$ | [16] |
| Peeling | $$\frac{\mathrm{L_l}\prod_{\ell=1}^{L}\|A_\ell\|_{\mathrm{Fr}}\sqrt{L}\prod_{\ell=1}^{L}\underline{w}_\ell B}{\sqrt{N}}$$ | [32] |
| **CNNs**
**Parametric Interpolation** | | |
| | $$\mathcal{B}[L\,\mathrm{L_l}]^{\frac{p}{2+p}}\sqrt{\frac{L}{N}}$$ $$\left[\sum_{\ell=1}^{L}\left[\mathfrak{B}\prod_{i=1}^{L}\rho_i\|\mathrm{op}(A_i)\|\right]^{\frac{2p_\ell}{p_\ell+2}}\times\right.$$ $$\left.\left[\frac{\|A_\ell-M_\ell\|_{\mathrm{sc},p_\ell}^{p_\ell}}{\|\mathrm{op}(A_\ell)\|^{p_\ell}}\right]^{\frac{2}{p_\ell+2}}[U_\ell+d_{\ell-1}]W_\ell^{\frac{p_\ell}{p_\ell+2}}\right]^{\frac{1}{2}}$$ | Theorem G.5
Note:
$W_\ell:=U_\ell\times w_\ell$
(for $\ell\neq L$)
and $W_L:=1$. |
| Parametric Interpolation
and Loss augmentation | $$\mathcal{B}[L\,\mathrm{L_l}]^{\frac{p}{2+p}}$$ $$\left[\sum_{\ell=1}^{L}\left[\mathfrak{B}_{\ell-1,A}^{\mathfrak{C}}\prod_{i=\ell}^{L}\rho_i\|\mathrm{op}(A_i)\|\right]^{\frac{2p_\ell}{p_\ell+2}}\times\right.$$ $$\left.\left[\frac{\|A_\ell-M_\ell\|_{\mathrm{sc},p_\ell}^{p_\ell}}{\|\mathrm{op}(A_\ell)\|^{p_\ell}}\right]^{\frac{2}{p_\ell+2}}[U_\ell+d_{\ell-1}]W_\ell^{\frac{p_\ell}{p_\ell+2}}\right]^{\frac{1}{2}}$$ | Theorem 3.7 |

Table 2: Summary of results and previous State of the Art. By convention, all results are expressed in $\widetilde{O}$ notation, which hides polylogarithmic factors of all relevant architectural ($\mathcal{W}, \mathcal{A}, \overline{W}, L$) and scaling ($\max_\ell \|A_\ell\|, \|A\|_{\mathrm{sc}}$, etc.) quantities. In addition, we only consider the dominant Rademacher complexity term: for simplicity we ignore both the missing terms of $O\left(\mathcal{B}\sqrt{\frac{\log(1/\delta)}{N}}\right)$ and additional terms of the form $\widetilde{O}\left(\sqrt{\frac{L}{N}}\right)$ which would occur from making the bounds post hoc w.r.t. to the norm constraints (some works do this explicitly whilst some do not).

## C    In-depth Comparison to Existing Works

In all the descriptions below, the $\widetilde{O}$ notation absorbs polylogarithmic factors in $\mathfrak{B}, \mathcal{W}, \max_i \|A_\ell\|, N, \mathrm{L_l}$ as well as the Lipschitz constant $\mathrm{L_l}$ of the loss or the margin $\gamma$.

Recall that in [3], it was shown that

$$\mathrm{GAP} - O\left(\mathcal{B}\sqrt{\frac{\log(1/\delta)}{N}}\right) \leqslant \widetilde{O}\left(\mathrm{L_l}\frac{L\sqrt{\overline{W}}}{\sqrt{N}}\left(\prod_{\ell=1}^{L}\|A_\ell\|\right)\left(\sum_{\ell=1}^{L}\frac{\|A_\ell\|_{\mathrm{Fr}}^2}{\|A_\ell\|_\sigma^2}\right)^{\frac{1}{2}}\right). \tag{C.1}$$

In [1], it was shown that with probability greater than $1 - \delta$ over the draw of the training set we have

$$\mathrm{GAP} - \mathcal{B}\sqrt{\frac{\log(1/\delta)}{N}} \leqslant \widetilde{O}\left(\mathrm{L_l}\frac{1}{\sqrt{N}}\prod_{\ell=1}^{L}\|A_\ell\|\left(\sum_{\ell=1}^{L}\frac{\|(A_\ell - M_\ell)^\top\|_{2,1}^{\frac{2}{3}}}{\|A_\ell\|^{\frac{2}{3}}}\right)^{\frac{3}{2}}\right). \tag{C.2}$$

Note that equation (C.1) was discovered concurrently with (C.2) with a different approach ([3] relied on a PAC Bayes approach).

### C.1    Bounds for Linear Neural Networks

In this subsection, we briefly list generalization bounds which can be obtained for linear maps in the multi-class classification case, focussing especially on the ones which apply to linear networks of the form $B_L \ldots B_1$.

Most of the early literature on generalization bounds for linear multi-class classification focused on kernels, which are a slightly more general setting.

For a single output ($\mathcal{C} = 1$), it is well known from early results [68, 83] that the Rademacher complexity [7] of linear classifiers is bounded as $\widetilde{O}\left(\sqrt{\frac{\|a\|^2 \sum_i \|x_i\|^2}{N^2}}\right) \leqslant \widetilde{O}\left(\mathfrak{B}\sqrt{\frac{\|A\|^2}{N}}\right)$. Early work on generalization bounds for multi-class classification [85, 68, 66, 67, 86, 87] used inequalities for Rademacher complexities of composite function classes to obtain bounds of the form $\widetilde{O}\left(\mathfrak{B}\sqrt{\frac{\mathcal{C}\|A\|_{\mathrm{Fr}}^2}{N}}\right)$. Later in [88, 65], the bound was refined to $\widetilde{O}\left(\mathfrak{B}\sqrt{\frac{\|A\|_{\mathrm{Fr}}^2}{N}}\right)$ by exploiting the $L^\infty$ continuity of common loss functions.

In [64], the authors proved the following bound specifically targetted at linear networks of the form $A = B_L \ldots B_2 B_1$:

$$\widetilde{O}\left(\mathfrak{B}\sqrt{\frac{\mathcal{C}\min_\ell(\mathrm{Rank}(B_\ell))}{N}}\prod_\ell \|B_\ell\|\right). \tag{C.3}$$

Note that $\prod_\ell \|B_\ell\| \geqslant \|A\|$ and $\|A\| \min_\ell \sqrt{\mathrm{rank}(B_\ell)} \geqslant \|A\|\sqrt{\mathrm{rank}(A)} \geqslant \|A\|_{\mathrm{Fr}}$. Thus, the bound in (C.3) behaves at least as $\widetilde{O}\left(\sqrt{\frac{\mathcal{C}\|A\|_{\mathrm{Fr}}^2}{N}}\right)$, which exhibits the same behavior as the results in [86, 66].

The results are expressed in terms of vector output Gaussian complexity, which has the advantage of yielding bounds for L2 Lipschitz loss functions. When the loss function is $L^\infty$ Lipschitz, the bound from [65] will be tighter by a factor of $\sqrt{\mathcal{C}}$ (ignoring any logarithmic factors). The main advantage of the bound in equation (C.3) is the introduction of the novel proof technique which also allows the authors to derive the bound (C.9) for neural networks with activation functions.

**Comparison between Bound** (C.1) **and Theorem 3.1 for linear networks:**

It is worth noting that the Bound (C.1) can be instantiated as follows:

---

[7]For simplicity, we use $\widetilde{O}$ notation, however, many of the result in this section do not require logarithmic factors when expressed as a function of Rademacher complexity only. However, making them post hoc w.r.t. to the constraints requires additional logarithmic factors.

**Corollary C.1.** *With probability greater than $1 - \delta$ over the draw of the training set, every matrix $A$ satisfies the following generalization bound*

$$\text{GAP} \leqslant \tilde{O}\left(\mathcal{B}\sqrt{\frac{\log(1/\delta)}{N}} + \mathrm{L_l}\,\mathfrak{B}\|A\|^{\frac{L-1}{L}}L^{\frac{3}{2}}\sqrt{\frac{\mathcal{C}\|A\|_{\mathrm{sc},\frac{2}{L}}^{\frac{2}{L}}}{N}}\right). \tag{C.4}$$

*Proof.* By the proof of Theorem 1.1, the values of $B_1, \ldots B_L$ which minimize the weight decay regulariser $\sum_{\ell=1}^{L}\|B_\ell\|_{\mathrm{Fr}}^2$ satisfy $\|B_\ell\| = \|A\|^{\frac{1}{L}}$ and even $\sigma_i(B_\ell) = \sigma_i(A)^{\frac{1}{L}}$ for all $\ell, i$ where $\sigma_i(Z)$ denotes the $i$th singular value of the matrix $Z$. It follows that for this choice of $B_\ell$s, we have $\prod_{i\neq\ell}\|B_i\| = \|A\|^{\frac{L-1}{L}}$ and $\|B_\ell\|_{\mathrm{Fr}}^2 = \|A\|_{\mathrm{sc},\frac{2}{L}}^{\frac{2}{L}}$. The theorem follows upon replacing the values into equation (C.1). $\qquad\square$

In contrast, for $p = \frac{2}{L}$, our Theorem 3.1 scales as

$$\tilde{O}\left(\mathrm{L_l}^{\frac{p}{2+p}}\sqrt{\frac{\mathfrak{B}^{\frac{2p}{2+p}}\|A\|_{\mathrm{sc},p}^{\frac{2p}{2+p}}\min(\mathcal{C},d)^{\frac{p}{p+2}}\max(\mathcal{C},d)^{\frac{2}{p+2}}}{N}}\right)$$

$$\leqslant \tilde{O}\left(\mathrm{L_l}^{\frac{1}{L+1}}\sqrt{\frac{\mathfrak{B}^{\frac{2}{L+1}}\|A\|_{\mathrm{sc},p}^{\frac{2}{L+2}}[\mathcal{C}+d]}{N}}\right). \tag{C.5}$$

The two are not directly comparable: for one thing, the bound in (C.4) is purely norm-based, which means that it enjoys more obvious scaling properties. This also makes it more vulnerable to large inputs. It is worth noting that both bounds potentially improve when $L$ grows, as this influences the schatten quasi norm index. However, this benefit is strongly mitigated in the case of (C.1) by the multiplicative factors of $L$, which are not present in our case. The explicit dependency on $\mathcal{C}$ is also worse in (C.1), but we believe this is an artefact of the $L^2$ based proof technique and might be improvable with an argument more carefully dedicated to the linear case.

## C.2 Norm-based Bounds for Fully Connected Neural Networks

In [13], extending work from [12], Rademacher complexity bounds of the following order were obtained:

$$\tilde{O}\left(\mathrm{L_l}\sqrt{\frac{\mathfrak{B}^2 L\prod_{\ell=1}^{L}\|A_\ell\|_{\mathrm{Fr}}^2}{N}}\right). \tag{C.6}$$

In addition, the same paper [13] contains other variants, which mostly consist of products of various norms of the weight matrices. For instance (Corollary 2 page 13) for any fixed value $p$: GAP $\leqslant$

$$\tilde{O}\left(\mathcal{B}\sqrt{\frac{\log(\frac{1}{\delta})}{N}} + \mathrm{L_l}\,\mathfrak{B}\mathcal{U}\left[\prod_{\ell=1}^{L}\|A_\ell\|\right]\left[\frac{L}{p}\right]^{\frac{1}{\frac{2}{3}+p}}\frac{1}{N^{\frac{1}{2+3p}}}\right), \tag{C.7}$$

where $\mathcal{U}$ is an apriori upper bound on $\max_\ell \frac{\|A_\ell^\top\|_{2,1}}{\|A_\ell\|}$. Note: we have simplified the expression to adapt it to our $\tilde{O}$ notation assuming that $\frac{M_p(A)}{M(A)} \leqslant O(\overline{W}^{\frac{1}{p}})$, where $M_p(A)$ (resp. $M(A)$) are upper bounds on the Schatten $p$ and spectral norms of $A$ respectively (such an assumption makes sense since for any matrix with dimensions bounded by $\overline{W}$, we have $\frac{\|A\|_{\mathrm{sc},p}}{\|A\|} \leqslant \overline{W}^{\frac{1}{p}}$). It is not possible to directly compare the factors of $L$ in our bounds and in (C.7) since a fully post hoc version of that result would likely involve further factors of $L$ as in our own proof.

Note that $\mathcal{U} \leqslant \overline{W}$, thus setting $p = \frac{1}{3}$ we obtain for instance

$$\text{GAP} - O\left(\mathcal{B}\sqrt{\frac{\log(1/\delta)}{N}}\right) \leqslant \tilde{O}\left(\mathrm{L_l}\,\mathfrak{B}\prod_{\ell=1}^{L}\|A_\ell\|\frac{\overline{W}L}{N^{\frac{1}{3}}}\right). \tag{C.8}$$

Still in the category of 'peeling' based bounds relying on vector contraction inequalities, the recent work of [64] (cf. Theorem 7), which appeared independently of the present work, has established bounds of the following form for the Gaussian complexity of neural networks with fixed norm and rank constraints

$$
O\left(\prod_{\ell=1}^{L} \|A_\ell\| \mathfrak{B} \frac{C_1 \sqrt{w_1} \frac{\|A_1\|_{\mathrm{Fr}}}{\|A_1\|} + \sum_{\ell=2}^{L} C_1^{L-\ell} C_2 \sqrt{w_\ell \tilde{r}_\ell} \frac{\|A_\ell\|_{\mathrm{Fr}}}{\|A_\ell\|}}{\sqrt{N}}\right),
$$

(C.9)

where $\tilde{r}_\ell$ is an upper bound on the rank of layer $\ell$. In fact, a notable property of the bound is that there are no logarithmic factors in the Rademacher complexity bound: however, note the Rademacher complexity bound into an excess risk bound would incur at least a logarithmic factor of $N$, and making the bound post hoc w.r.t. the norm constraints would incur additional logarithmic factors of the norms, as well as a non logarithmic factor of $\sqrt{L}$. Here, $C_1, C_2$ are unspecified constants inherited from [78] (which are described as 'rather large' on page 4 of [78]). However, the authors in [64] do conjecture that the constants and the accompanying explicitly exponential dependence on depth could be removed with a more careful analysis

$$
\mathfrak{G} \leqslant O\left(C_1^L \mathfrak{B} \left[\prod_{\ell=1}^{L} \|A_\ell\|\right] L\underline{r}\sqrt{\frac{\overline{W}}{N}}\right),
$$

(C.10)

where $\mathfrak{G}$ denotes the Gaussian complexity, and $\underline{r}$ the minimum rank of the weight matrices $A_1, \dots, A_L$.

**Comparison between (C.10) in [64] and (E.125)**

Both results take the low-rank structure of the weight matrices into account and improve when neural collapse occurs. However, both the proof techniques and the final results differ significantly. Whilst translating the bound (C.10) into an *excess risk* bound would imply at least a logarithmic factor of $N$, and making the bound post hoc w.r.t. the norm constraints would incur additional logarithmic factors of the norms, as well as a non logarithmic factor of $\sqrt{L}$, the bound for the Gaussian complexity is free of any logarithmic factors (in contrast, even our Rademacher complexity bound from the calculation in Proposition E.1 includes logarithmic factors of the norms arising from the application of $L^\infty$ results such at Lemma I.5), which could give it an advantage when $L$ is small. On the other hand, the bound C.10 and C.9 exhibit both explicit and implicit exponential dependence on depth through the factors $C_1^L$ and $\prod \|A_\ell\|$ respectively. In contrast, the result in G.7 has no non logarithmic dependency on norm based quantities. In fact, it belongs to the family of parameter counting bounds, whilst (C.10) is a purely norm based. In the case where all the spectral norms and the norms of the datapoints are considered to be $O(1)$, the bound (C.10) amounts to a sample complexity of $\widetilde{O}(C_1^{2L}\underline{r}^2 L^2 \overline{W})$, whilst Theorem G.7 amounts to a sample complexity of $\widetilde{O}\left(\underline{r}L^2\overline{W}\right)$: thus, the bounds are nearly identical in this particular situation apart from the removal of a factor of $\underline{r}C_1^L$ (which removes exponential dependence on depth) in our work at the cost of additional logarithmic factors. It is worth noting that the proofs are radically different: we rely on layer-wise covering numbers analogous to [1, 4, 74, 16] whilst [64] relies on vector contraction inequalities and is closer to the works of [12, 13]. In particular, one of the factors of $L$ inside the square root in Theorem E.8 arises from making the bounds post hoc w.r.t. the spectral norms. It would be interesting to investigate whether this factor can be removed when assuming that $\|A_\ell\| = 1$ for all $L$.

Several other comparable bounds are proved in [64], which appeared independently of the present work. The bounds only apply to the exactly low-rank case, whilst we provide a more refined analysis based on the singular value spectrum decay implicit in the low Schatten $p$ quasi norm constraints. In addition, we also consider convolutional neural networks, and the proof techniques are radically different.

## C.3 Norm-based Bounds for Convolutional Neural Networks:

In [16], it was shown that [8]

$$\text{GAP} - O\left(\mathcal{B}\sqrt{\frac{\log(1/\delta)}{N}}\right) \leqslant \tilde{O}\left(\sqrt{L}\frac{\left[\sum_{\ell=1}^{L}(T_\ell)^{2/3}\right]^{\frac{3}{2}}}{\sqrt{N}}\right) \tag{C.11}$$

where if $\ell \leqslant L - 1$,

$$T_\ell = B_{\ell-1}(X)\|(A_\ell - M_\ell)^\top\|_{2,1}\sqrt{w_\ell}\max_{U \leqslant L}\frac{\prod_{u=\ell+1}^{U}\|\operatorname{op}(\tilde{A}_u)\|}{B_U(X)}$$

and if $\ell = L$,

$$T_L = \frac{B_{L-1}(X)}{\gamma}\|A^L - M^L\|_{\text{Fr}}.$$

Here, we denote $B_\ell(X) = \max_i \|F_A^{0\to\ell}(x)\|_\ell$, where the norm $\|\cdot\|_\ell$ denotes the maximum $L^2$ norm of a convolutional patch for a vector of activations at layer $\ell$.

The above bound relies on loss function augmentation to control the L2 norm of patches at each layer. However, the following simpler bound (Theorem E.2) can also be of interest:
$\text{GAP} - O\left(\mathcal{B}\sqrt{\frac{\log(1/\delta)}{N}}\right) \leqslant$

$$\tilde{O}\left(\frac{L_1\sqrt{L}\prod_{\ell=1}^{L}\|\operatorname{op}(A_\ell)\|}{\sqrt{N}}\left[\sum_{\ell=1}^{L}\left(\frac{U_\ell w_\ell\|[A_\ell - M_\ell]^\top\|_{2,1}}{\|\operatorname{op}(A_\ell)\|}\right)^{\frac{2}{3}}\right]^{\frac{3}{2}}\right). \tag{C.12}$$

Recently, in [32] an ingenious approach was used to provide bounds for convolutional neural networks in terms of products of the Frobenius norms of the weight matrices. The bounds and proof techniques are closer to the school of [12, 13] than those of [1], and rely on vector concentration inequalities and direct calculations of Rademacher complexities rather than covering numbers, whilst incorporating many novel elements specific to the CNN situation. We have, considering a post hoc version of Theorem 3.2 in [32] and translating to our notation, $\text{GAP} - O\left(\mathcal{B}\sqrt{\frac{\log(1/\delta)}{N}}\right) \leqslant$

$$\tilde{O}\left(\mathcal{B}\sqrt{\frac{L^2}{N}} + \frac{L_1\prod_{\ell=1}^{L}\|A_\ell\|_{\text{Fr}}\sqrt{L\prod_{\ell=1}^{L}\underline{w_\ell}\underline{B}}}{\sqrt{N}}\right), \tag{C.13}$$

where $\underline{w_\ell}$ denotes the spatial size of the convolutional patches at layer $\ell$ and $\underline{B} = \max_{o,i}\|(x_i)_{S^{0,o}}\|$ is the maximum $L^2$ norm of a convolutional patch in the input.

The bounds (C.12) and (C.13) exploit the sparsity of connections at the first layer to obtain bounds which only depend on the maximum norm of an input convolutional patch (as opposed to the full norm of the input), a characteristic which is shared by our result (cf. theorem 3.6. In addition, the results in [16] exploit the weight sharing to involve only the norms of the weight matrices (rather than the linear operators $\operatorname{op}(A_\ell)$) in the numerator. Quite remarkably, the bound in [32] achieves a similar effect in the context of peeling type results by relying exclusively on the sparsity of connections. Our results for CNNs can be interpreted as a low-rank generalization of those in [16] incorporating low-rank structure in the weight matrices, just as our results for DNNs generalize those of [1]. However, the results in [32] are not directly comparable, since they belong to the peeling family, affording them a more favorable dependence on width but a poorer dependence on depth if the norms $\|A_\ell\|_{\text{Fr}}$ are moderately large.

---

[8] As explained in the main paper, our convention on $\tilde{O}$ notation is stricter than that in [16]: we allow the $\tilde{O}$ to absorb factors of $\log(\max_{\ell=1}^{L}\|A_\ell\|)$, but we treat $\log(\prod_{\ell=1}^{L}\|A_\ell\|)$ as $\tilde{O}(L)$, which explains the additional multiplicative factor of $\sqrt{L}$. This factor is absent in [74] due to the use of a purely $L^2$ covering number approach, though the implicit dependency on the number of classes at the last layer is stronger.

## C.4 Parameter Counting Bounds

In [7], generalization bounds based on VC dimensions were established for fully connected neural networks. In more recent research the focus of parameter counting bounds has been on different architectures such as Convolutional Neural Networks or ResNets. We focus on CNNs below as this work focuses on DNNs and CNNs.

In [4] (Theorem 3.1 ), the following bound is proved for CNNs with fixed norm constraints:

$$\text{GAP} \leqslant \widetilde{O}\left( \mathcal{B}\sqrt{\frac{\mathcal{W}\mathcal{S}L}{N}} + \mathcal{B}\sqrt{\frac{\log(1/\delta)}{N}} \right), \tag{C.14}$$

where $\mathcal{W} := \sum_{\ell=1}^{L} d_{\ell-1} \times U_\ell$ denotes the total number of parameters in the network and $\mathcal{S}$ is an a priori upper bound on $[[\max_\ell \|A_\ell\|] - 1]$.

Later in [74] (Theorem 3.4), the following result was shown[9]:

$$\text{GAP} \leqslant \widetilde{O}\left( \mathcal{B}\sqrt{\frac{\mathcal{W}L}{N}} + \mathcal{B}\sqrt{\frac{\log(1/\delta)}{N}} \right). \tag{C.15}$$

Note: in [4], the regime of interest is that where the spectral norms of the weight matrices approach 1, which explains the appearance of the quantity $\mathcal{S}$. In fact, the quantity appears through an upper bound on the product of spectral norms as $(1 + \nu)^L \leqslant \exp(\nu L)$, thus, the multiplicative factor of $[L\mathcal{S}]$ can relatively straightforwardly be replaced by $\widetilde{O}(L)$. This results in a bound which looks identical to equation (C.15) when expressed in $\widetilde{O}$ notation. However, the bound in [74] presents many other advantages, including applicability to Resnets and much smaller constants and logarithmic factors. We also note that [89] provide rank-sensitive bounds for neural networks. However, the bounds include a multiplicative factor of the Jacobian of the full network (both outside and inside the logarithmic factors): the bounds will coincide (up to log factors) with our pure parameter counting bounds in the particular case when all the spectral norms of the weights are exactly equal to 1, but behave less favorably in other cases.

# D    Experiments

In this section, we present some experiments to demonstrate that our bound can successfully capture the low rank structure in the weights to improve upon competing bounds. Although the numerical advantage of our bounds over the best compared method (the parameter counting bounds of [74]), the bounds generally show a much milder growth as the width of the layers grows compared to all existing bounds when the generalization gap is kept relatively constant. Thus, our bounds are able to adapt to capture the presence of unused function class capacity in the form of low-rank weights. We present results for both DNNs on the MNIST dataset and CNNs on the CIFAR-10 dataset. All experiments were run with 128 CPUs with 500GB ram and DGX A100.

## D.1    Fully-connected Neural Networks (MNIST)

To evaluate the bounds' behavior for fully connected neural networks, we conducted experiments on the MNIST dataset. We set the number of non-linear hidden layers to $L = 3$ and varied the width of the hidden layers $w_\ell$ in $\{300, 400, 500, 700\}$ to explore different regimes. To guarantee that we could analyze the phenomenon studied in this research, we enforced rank-sparsity through two complementary approaches: (1) implicitly, by preceding each non-linear hidden layer with four linear layers. (2) explicitly, by regularizing the neural network through weight decay. We selected the regularization parameter from $\{10^{-4}, 5 \times 10^{-4}, 10^{-3}, \ldots, 10^2\}$ and analyzed models with the

---

[9]Note, keeping in line with the convention used for the $\widetilde{O}$ notation in this paper, we interpret $[\max_\ell \|A_\ell\|]$ as a constant. This implies that the presence of the quantities $C_{i,j}$ in [74] introduces a logarithmic dependency in $[\max_\ell \|A_\ell\|]^L$ inside the square root, corresponding to a multiplicative factor of $L$ inside the square root.

highest weight decay that achieved an accuracy of at least $0.9$. We then maximize the margin $\gamma$ subject to $I_{\gamma,X} \leqslant 0.1$.

The results can be seen in Figure D.1. To facilitate comparison, denser bars are lower than less dense bars when compared to their counterparts, i.e, by considering models with the same width. In addition, we only plot the bounds in $\widetilde{O}$ terms, ignoring all polylogarithmic factors. This facilitates the comparison since some competing baselines are post hoc whilst some are not: our bounds hold uniformly over all values of $p_\ell, \|A_\ell - M_\ell\|_{scc,p_\ell}, \gamma$ etc., whilst some of the related works do not explicitly perform the required union bound to achieve this. The practice of evaluating the numerical values of generalization bounds only up to polylogarithmic factors [74, 64] or not at all [3, 4] is in line with the literature. When comparing to [64], we evaluate the bound for two values of the intractable constant $C_1 = 1, 2$. This is to account for the the authors' statement that the factor of $C_1^L$ could potentially be removed with a more refined proof, and certainly underestimates the numerical value of the established bound, since the constant $C_1$ is 'rather large' [78]. We include both versions of our bound: Theorem 3.2 (without loss function augmentation) and Theorem 3.3 (with loss function augmentation). A key observation is that as the width increases, our bounds do not increase as fast as the competing results, suggesting a successful use of the rank sparsity of the problem: in this experiment, the accuracy and generalization gap are constant (overparametrized regime), therefore, the more modest growth exhibited by our model indicates a better ability to capture true generalization error than the competing methods, which show a much sharper growth with width.

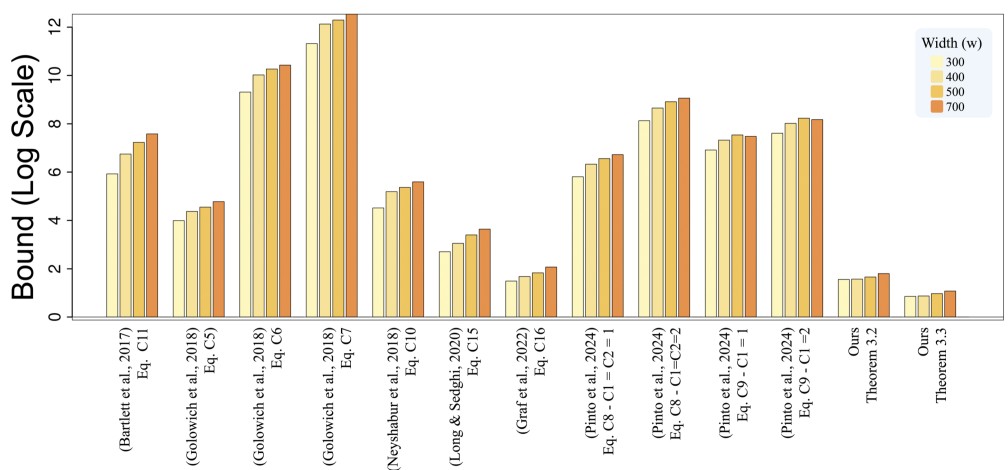

Figure D.1: Empirical comparison between our generalization results and previous works.

**Further Validation of the Low-rank Condition** Moreover, the singular value spectra plots from Figure D.6 below demonstrates the presence of rank sparsity in our trained models. This confirms the observations made by a multitude of recent papers in the optimization literature [21, 22, 27], which further explains why our bounds' advantage improves when the width increases. In this example, even the value $p_\ell = 0$ yields an advantage over classic parameter counting bounds due to the exact low-rank structure.

### D.2 Convolutional Neural Networks (CIFAR-10)

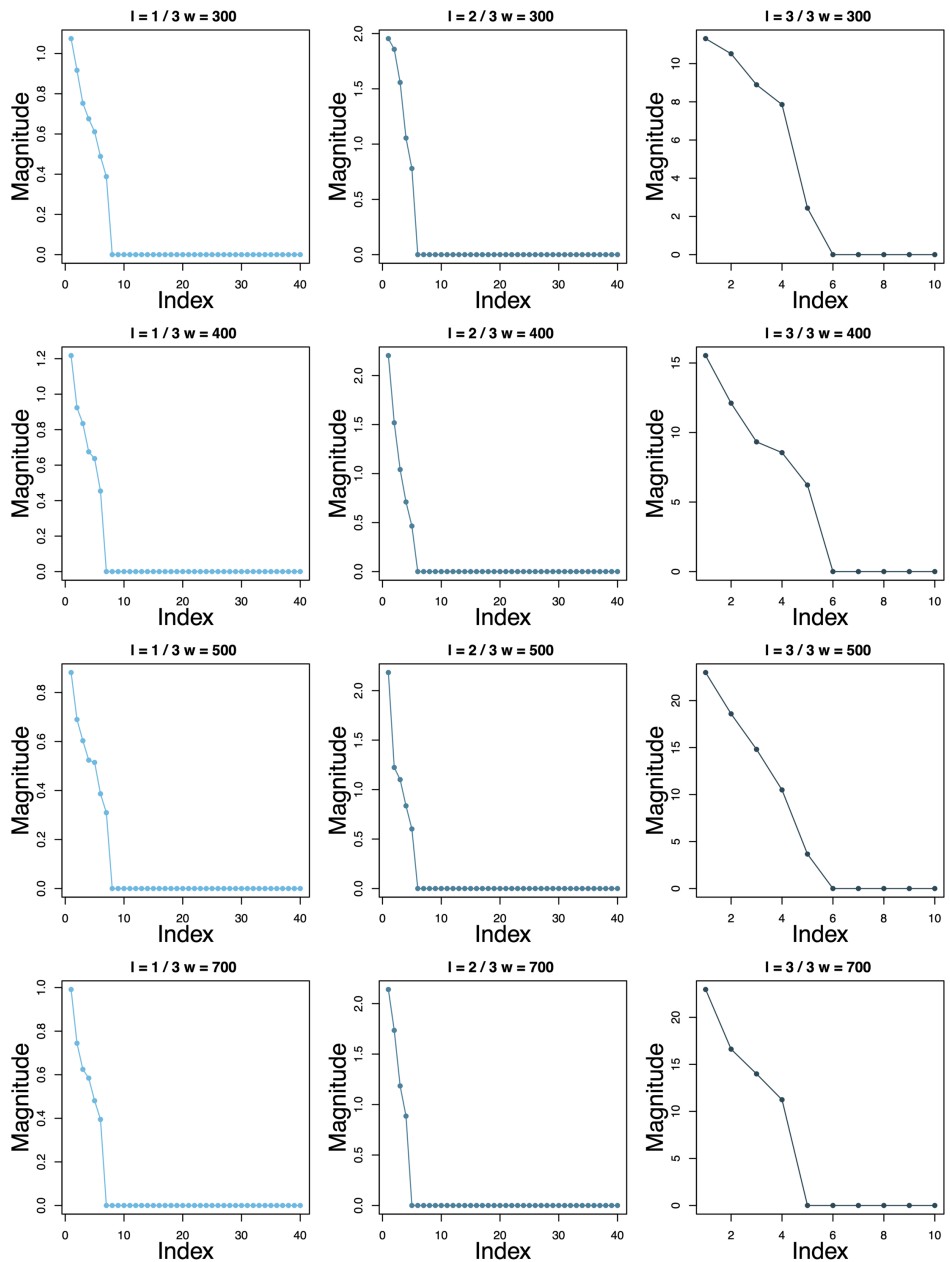

Figure D.2: Spectral Decay of Intermediary Layers in the MNIST Dataset (Fully Connected).

## D.3 Convolutional Neural Networks (CIFAR-10)

In this section, we present experiments on the CIFAR-10 dataset. We train a neural network with 3 convolutional layers with kernel size $(3, 3)$ each followed by 5 fully connected layers. The last (output) layer has width 10 since there are 10 classes, and the width of the intermediary fully connected layers is varied between 100 and 1000. Due to the additional computational burden of the task and the more pronounced overparametrization, we evaluate the bounds on a single trained network for each architecture. To moderate the effect of norm-based factors, we employ spectral regularization, constraining the spectral norms of all weight matrices to 1. We set the margin dynamically to ensure a loss of only 1 percent in accuracy compared to a margin of zero. As in the fully connected case, we note that as most of the bounds are asymptotic in nature (and many of the sources do not calculate the explicit constants) their 'practical evaluation' challenging. For this reason, we only evaluate dominant term of each bound (ignoring the constants and logarithmic factors) and we employ a logarithmic scale, which is better suited to this asymptotic analysis since the bounds differ from each other by very large factors. This is also standard practice in the related literature [74, 4, 1] whenever bounds are evaluated. As in the fully connected case, our bounds outperform the competing ones. However, more interesting is the fact that our bounds exhibit a milder dependency on the width, which better aligns with the true generalization error.

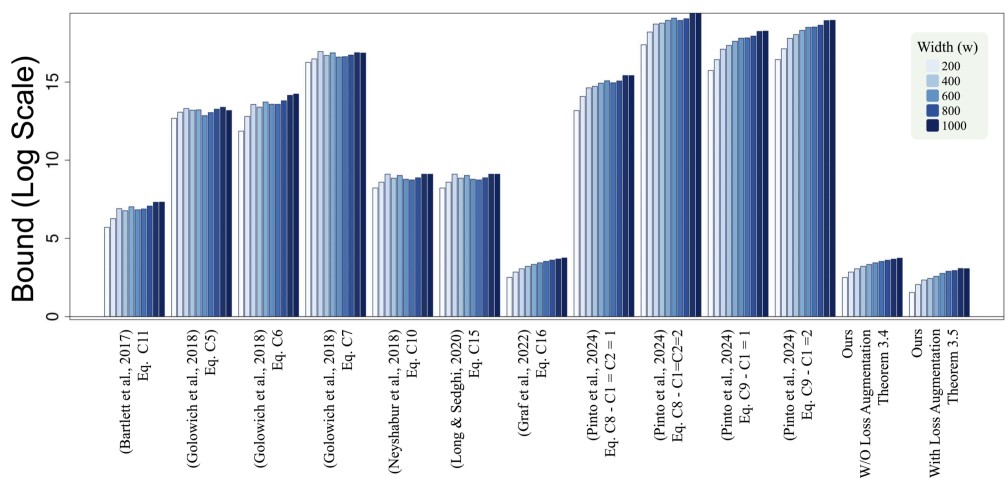

Figure D.3: Numerical Comparison of our Bounds with other Bounds from the Literature (CNNs, CIFAR-10).

Furthermore, to provide further evidence of spectral decay and strong *approximate low-rank structure* in our trained models, we provide the following plots of the singular spectral of the weights of our trained models. The plots confirm that the approximate low-rank structure is stronger for larger widths, which also explains the fact that our bounds' advantage over existing approaches widens in such overparametrization regimes.

We also provide a few additional comparisons to validate our conclusion that our bounds respond to varying degrees of overparametrization better than pure parameter counting bounds. As mentioned in the main paper, the choice of $p_\ell$ can be made after training, as the bounds hold uniformly over their choice. In fact, the choice of $p_\ell$ made after optimizing the bounds is a relevant factor in the interpretation of the bound in terms of overparametrization regime: a larger but moderate value of $p$ (in the range $[0.2, 1]$) indicates that the norm-based factors are moderate enough to be involved in the bound and that the low-rank structure in the weight matrices is sufficiently pronounced to be exploited. On the other hand, a value of $p = 0$ can indicate one of two things: either (1) the norm-based factors are too large for the low rank structure to be exploited, or (2) the weight matrices are exactly low-rank. In our experiments, we observe the optimization of the bound often results in the value $p_\ell = 0$ being chosen for convolutional layers, whilst the fully connected layers tend to correspond to more moderate values of $p_\ell$. This can be seen in Figure D.5 below, which corresponds

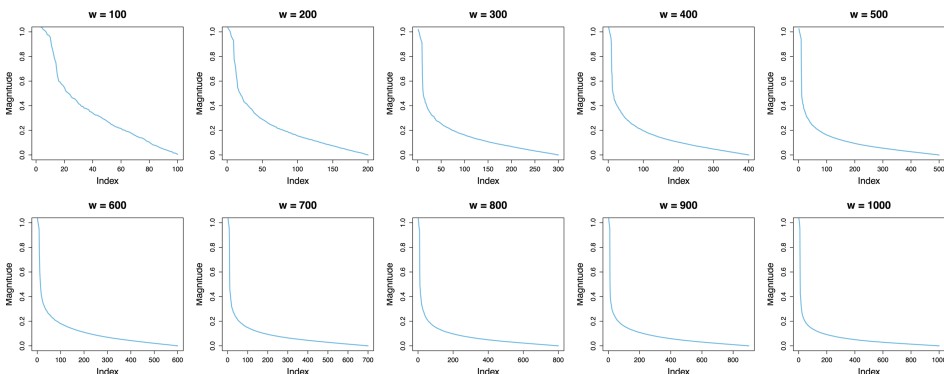

Figure D.4: Illustration of the Spectral Decay of the Trained Models on the Layer fc2.

to the most overparametrized CNN model ($w = 1000$) where the $p_\ell$s are selected based on the optimization of the bounds from Theorem 3.7 (with loss function augmentation).

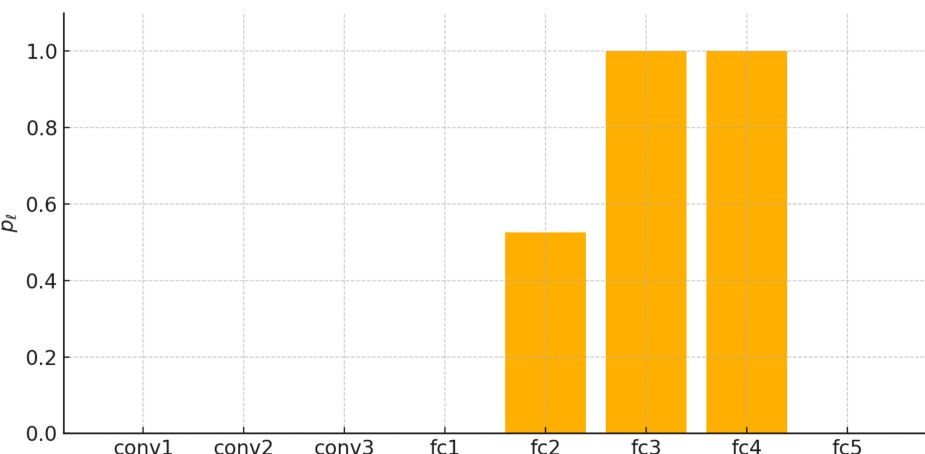

Figure D.5: Chosen Values of $p_\ell$ (with Loss Function Augmentation) for the most Overparametrized CNN Model (width $= 1000$)

The results are somewhat expected: convolutional layers are already quite parameter efficient, making the exploitation of the low-rank structure comparatively less necessary. Furthermore, the norm-based factors arising from the product $\prod_{i=\ell}^{L} s_i$ are larger since convolutional layers have a larger spectral norm due to the repetition over patches. On the other hand, the fully connected layers are able to select larger values of $p_\ell$ due to a more pronounced low-rank structure and moderate norm-based factors achieved through a combination of loss function augmentation and spectral regularization during training.

We note that the choice of $p_\ell$ generally introduces a tradeoff which is only imperfectly captured by the regime corresponding to the value of $p_\ell$ chosen when optimizing the bound. Indeed, lower values of $p_\ell$ correspond both to a bias towards the parameter-counting component of our bounds and stronger exploitation of the approximate low-rank structure (assuming the norm based factors are not too large). Thus, more moderate values of $p_\ell$ are preferable in terms of the bounds' responsivity to changes in architecture, whilst setting very close to zero can sometimes be beneficial in terms of overall numerical performance due to the convergence to a parameter counting result. To better isolate the bounds' ability to capture low-rank structure and its indirect effect on function class capacity, we evaluate the bounds when fixing the value of $p_\ell$ to various values for all layers. We then compare the behavior of our results from Theorem G.5 (without loss function augmentation, in blue) and Theorem 3.7 (with loss function augmentation, in turquoise) to the parameter counting baseline of [74] (in yellow). To isolate the behavior as the overparametrization increases, we plot the

ratio between the value of the bounds compared to the value of the same bound for a width of 100. Similarly, we also plot the ratio of the *test error* (in red) compared to the test error for a width of 100 (i.e. $0.2793$). The results are in Figure D.6 below.

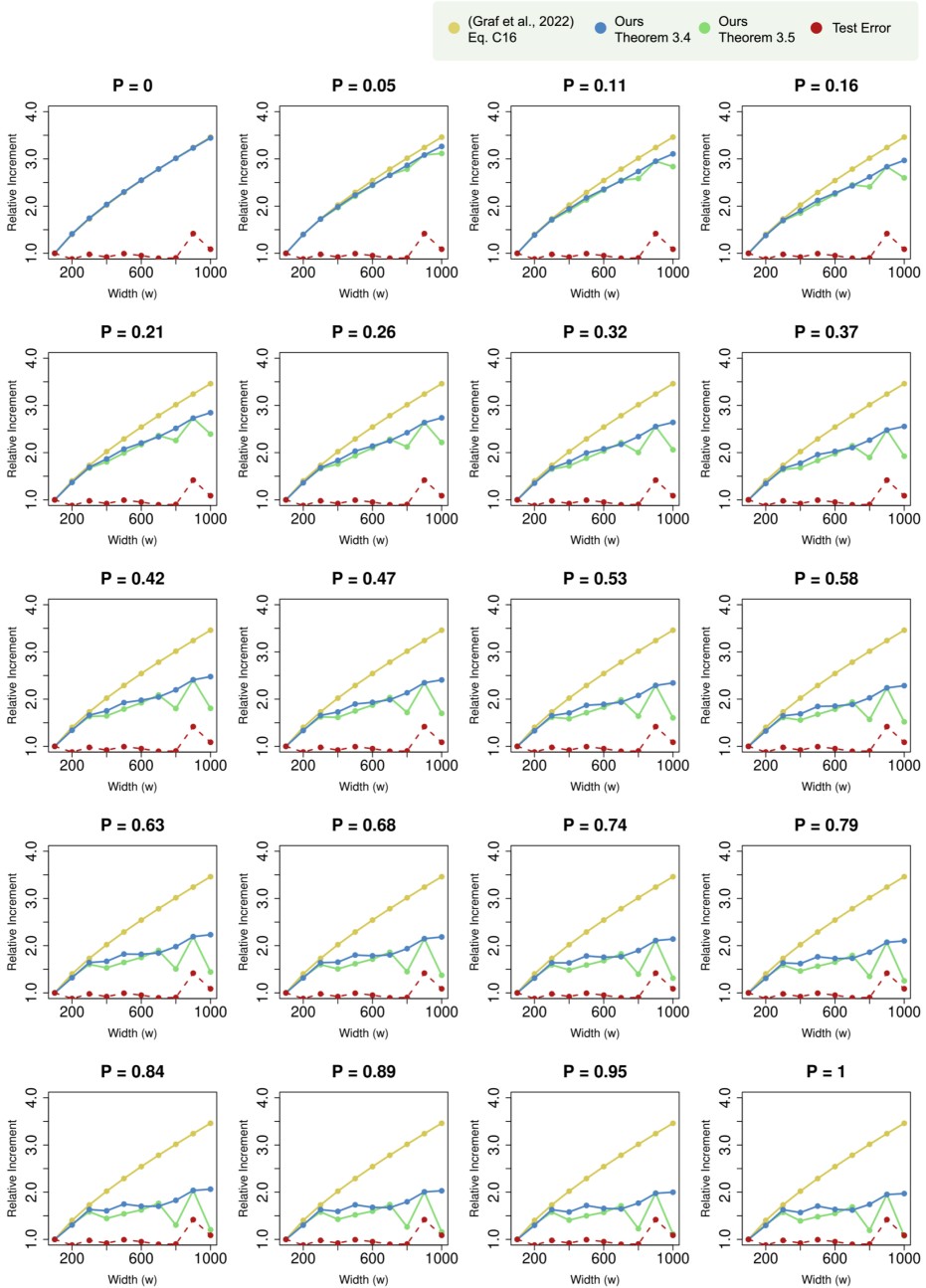

Figure D.6: Relative Growth of our Main Results as a Function of Width

We can directly observe that for moderate values of $p_\ell$, both our bounds are successful at capturing the implicit function class restriction that occurs through both norm-based effects and spontaneous rank-sparsity in the overparametrized regime: for small widths (100 to 300), the bounds all grow linearly, which correspond to a parameter-counting behavior (indeed, the parameter count grows quadratically in the width, which results in linear growth of the term $\sqrt{\frac{D}{N}}$ where $D$ is the number of parameters), at the regime of moderate overparametrization (width 300 to 700), our bounds maintain

relatively constant behavior, which matches the behavior of the test error. Then, at the extremely overparametrized regime (widths 800 to 1000), the bounds begin to grow again and exhibit less stable behavior, which also better matches the behavior of the test error compared to the parameter counting baseline, which continues to grow linearly.

# E   Generalization Bounds for Fully Connected Networks

In this section, we prove our generalization bounds for classes of neural networks with Schatten $p$ quasi norm constraints on the weight matrices. For each layer $\ell$, we have a set of admissible matrices $\mathbb{R}^{w_\ell \times w_{\ell-1}} \ni \mathcal{B}_\ell := \{A_\ell \in \mathbb{R}^{w_\ell \times w_{\ell-1}} : \|A_\ell - M_\ell\|_{\mathrm{sc},p_\ell} \leqslant \mathcal{M}_\ell; \|A_\ell\| \leqslant s_\ell\}$ and

$$\mathcal{F}_\mathcal{B} := \left\{ F_A = F_{(A_1,\ldots,A_L)} : A_\ell \in \mathcal{B}_\ell \quad \forall \ell \leqslant L \right\} \tag{E.1}$$

$$= \left\{ F_A = F_{(A_1,\ldots,A_L)} : \|A_\ell - M_\ell\|_{\mathrm{sc}} \leqslant \mathcal{M}_\ell \, \|A\| \leqslant s_\ell \quad \forall \ell \leqslant L \right\}.$$

We define similarly:

$$\mathcal{F}_\mathcal{B}^{0 \to \ell} := \left\{ F_A^{0 \to \ell} = F_{(A_1,\ldots,A_\ell)} : A_l \in \mathcal{B}_\ell \quad \forall l \leqslant \ell \right\}, \tag{E.2}$$

where we define

$$\mathcal{B}_\ell = \left\{ A_\ell \in \mathbb{R}^{w_\ell \times w_{\ell-1}} : \|A_\ell - M_\ell\|_{\mathrm{sc},p_\ell} \leqslant \mathcal{M}_\ell; \|A_\ell\| \leqslant s_\ell \right\}.$$

**Proposition E.1.** *Fix some reference/initialization matrices $M^1 \in \mathbb{R}^{w_1 \times w_0} = \mathbb{R}^{w_1 \times d}, \ldots, M^L \in \mathbb{R}^{w_L \times w_{L-1}} = \mathbb{R}^{\mathcal{C} \times w_{L-1}}$. Assume also that the inputs $x \in \mathbb{R}^d$ satisfy $\|x\| \leqslant b$ w.p. 1. Fix a set of constraint parameters $0 \leqslant p_\ell \leqslant 2, s_\ell, \mathcal{M}_\ell$ (for $\ell = 1, \ldots, L$).*

*With probability greater than $1 - \delta$ over the draw of an i.i.d. training set $x_1, \ldots, x_N$, every neural network $F_A(x) := A_L \sigma_L(A^{L-1}\sigma_{L-1}(\ldots \sigma_1(A^1 x) \ldots)$ satisfying the following conditions:*

$$\begin{aligned} \|A^\ell - M^\ell\|_{\mathrm{sc},p_\ell} &\leqslant \mathcal{M}_\ell && \forall 1 \leqslant \ell \leqslant L \\ \|A\| &\leqslant s_\ell && \forall 1 \leqslant \ell \leqslant L-1 \\ \|A_L^\top\|_{2,\infty} &\leqslant s_L, \end{aligned} \tag{E.3}$$

*also satisfies the following generalization bound:*

$$\mathbb{E}\left[ \mathrm{l}(F_A(x), y) \right] - \frac{1}{N} \sum_{i=1}^N \mathrm{l}(F_A(x_i), y_i)$$

$$\leqslant 6[\mathcal{B}+1]\sqrt{\frac{\log(\frac{4}{\delta})}{N}} + \frac{240\sqrt{\log_2(4\underline{\Gamma})}\log(N)[\mathcal{B}+1]}{\sqrt{N}}(1 + L\,\mathrm{L_l})^{\frac{p}{2+p}}\mathcal{R}_{\mathcal{M},\mathrm{s},p,b} \tag{E.4}$$

*where $\mathcal{R}_{\mathcal{M},\mathrm{s},p,b} :=$*

$$\left[ \sum_{\ell=1}^{L-1} \left[ b \prod_i \rho_i s_i \right]^{\frac{2p_\ell}{p_\ell+2}} \mathrm{r}_\ell^{\frac{2}{p_\ell+2}} \bar{w}_\ell^{\frac{2}{p_\ell+2}} [w_\ell \underline{w}_\ell]^{\frac{p_\ell}{p_\ell+2}} + \left[ b \prod_i \rho_i s_i \right]^{\frac{2p_L}{p_L+2}} \mathrm{r}_\ell^{\frac{2}{p_L+2}} \bar{w}_L^{\frac{2}{p_L+2}} \underline{w}_L^{\frac{p_L}{p_L+2}} \right]^{\frac{1}{2}}, \tag{E.5}$$

*$\underline{\Gamma} := \left[ [b+1][L\,\mathrm{L_l}+1]\overline{W}N \prod_{i=1}^L [[\rho_i+1]s_i+1] \right]$ and we use the shorthands $\mathrm{r}_\ell := \frac{\mathcal{M}_\ell^p}{s_\ell^p}$, $\underline{w}_\ell := \min(w_\ell, w_{\ell-1})$, $\bar{w}_\ell := [w_\ell + w_{\ell-1}]$.*

*Proof.* Write $\mathcal{F}_\mathcal{B}$ for the class of neural networks satisfying conditions (E.3). By Lemma I.7 with $\alpha = \frac{1}{N}$ and Proposition E.11, we have the following bound on the empirical Rademacher complexity of $\mathcal{F}_\mathcal{B}$

$$\mathfrak{R}_S(\mathrm{l}\circ\mathcal{F}_\mathcal{B})$$

$$\leqslant \frac{4}{N} + \frac{12}{\sqrt{N}}\int_{\frac{1}{N}}^{\mathcal{B}}\sqrt{96\log_2(4\underline{\Gamma})\left[\frac{(1+L\,\mathrm{L_l})}{\min(\varepsilon,1)}\right]^{\frac{2p}{2+p}}\sum_{\ell=1}^{L}\left[b\prod_i\rho_i s_i\right]^{\frac{2p_\ell}{p_\ell+2}}\mathrm{r}_\ell^{\frac{2}{p_\ell+2}}\,\bar{w}_\ell^{\frac{2}{p_\ell+2}}\,\tilde{w}_\ell^{\frac{p_\ell}{p_\ell+2}}}\,d\epsilon$$

$$\leqslant \frac{4}{N} + \frac{120\sqrt{\log_2(4\underline{\Gamma})}}{\sqrt{N}}(1+L\,\mathrm{L_l})^{\frac{p}{2+p}}\left[\sum_{\ell=1}^{L}\left[b\prod_i\rho_i s_i\right]^{\frac{2p_\ell}{p_\ell+2}}\mathrm{r}_\ell^{\frac{2}{p_\ell+2}}\,\bar{w}_\ell^{\frac{2}{p_\ell+2}}\,\tilde{w}_\ell^{\frac{p_\ell}{p_\ell+2}}\right]^{\frac{1}{2}}\left[\int_{\frac{1}{N}}^{1}\frac{1}{\epsilon}d\epsilon+\mathcal{B}-1\right]$$

$$\leqslant \frac{4}{N} + \frac{120\sqrt{\log_2(4\underline{\Gamma})}[\log(N)+\mathcal{B}]}{\sqrt{N}}(1+L\,\mathrm{L_l})^{\frac{p}{2+p}}\left[\sum_{\ell=1}^{L}\left[b\prod_i\rho_i s_i\right]^{\frac{2p_\ell}{p_\ell+2}}\mathrm{r_L}^{\frac{2}{p_\ell+2}}\,\bar{w}_\ell^{\frac{2}{p_\ell+2}}\,\tilde{w}_\ell^{\frac{p_\ell}{p_\ell+2}}\right]^{\frac{1}{2}}.$$

(E.6)

Next, by Theorem I.6, we have w.p. $\geqslant 1-\delta$,

$$\mathbb{E}\left[\mathrm{l}(F_A(x),y)\right] - \frac{1}{N}\sum_{i=1}^{N}\mathrm{l}(F_A(x_i),y_i) \leqslant 6\,\mathcal{B}\sqrt{\frac{\log(4/\delta)}{2N}} + \frac{8}{N}$$

(E.7)

$$+ \frac{240\sqrt{\log_2(4\underline{\Gamma})}[\log(N)+\mathcal{B}]}{\sqrt{N}}(1+L\,\mathrm{L_l})^{\frac{p}{2+p}}\left[\sum_{\ell=1}^{L}\left[b\prod_i\rho_i s_i\right]^{\frac{2p_\ell}{p_\ell+2}}\mathrm{r}_\ell^{\frac{2}{p_\ell+2}}\,\bar{w}_\ell^{\frac{2}{p_\ell+2}}\,\tilde{w}_\ell^{\frac{p_\ell}{p_\ell+2}}\right]^{\frac{1}{2}}$$

$$\leqslant 6[\mathcal{B}+1]\sqrt{\frac{\log(\frac{4}{\delta})}{N}}$$

$$+ \frac{240\sqrt{\log_2(4\underline{\Gamma})}\log(N)[\mathcal{B}+1]}{\sqrt{N}}(1+L\,\mathrm{L_l})^{\frac{p}{2+p}}\left[\sum_{\ell=1}^{L}\left[b\prod_i\rho_i s_i\right]^{\frac{2p_\ell}{p_\ell+2}}\mathrm{r}_\ell^{\frac{2}{p_\ell+2}}\,\bar{w}_\ell^{\frac{2}{p_\ell+2}}\,\tilde{w}_\ell^{\frac{p_\ell}{p_\ell+2}}\right]^{\frac{1}{2}},$$

where the last inequality holds for $N \geqslant 9$ (if this doesn't hold, inequality (E.4) holds trivially). Plugging equation (E.7) into equation (E.6) yields the result from equation (E.4) as expected.

$\square$

Using Proposition E.1 we can prove the following post hoc version, where the values of $p_\ell$ can be optimized.

**Theorem E.2.** *With probability greater than $1-\delta$, for any $b \in \mathbb{R}^+$, every trained neural network $F_A$ and sampling distribution with $\|x\| \leqslant b$ w.p. 1 satisfy the following generalization gap for all possible values of the $p_\ell s$:*

$$\mathbb{E}\left[\mathrm{l}(F_A(x),y)\right] - \frac{1}{N}\sum_{i=1}^{N}\mathrm{l}(F_A(x_i),y_i)$$

$$\leqslant 6[\mathcal{B}+1]\sqrt{\frac{\log(1/\delta)}{N}} + 6\,\mathcal{B}\sqrt{\frac{5+\Theta_{\log}}{N}} + 480[\mathcal{B}+1](1+L\,\mathrm{L_l})^{\frac{p}{2+p}}\frac{\sqrt{\log(\gamma_{FC})}\log(N)}{\sqrt{N}}\tilde{\mathcal{R}}_{F_A},$$

*where* $\Theta_{\log} := 2\log\left[4\overline{W}+b\prod_i\rho_i\|A_i\|\right]\left[|\log(b)|+L\left[\sum_{\ell=1}^{L}|\log(\|A_\ell\|)|+2\log(4\overline{W})\right]\right]$, $\gamma_{FC} := 12\left[[b+1][L\,\mathrm{L_l}+1]\overline{W}N\prod_{i=1}^{L}[\rho_i\|A_i\|+1]\right]\left[4\overline{W}+b\prod_i\rho_i\|A_i\|\right]$, *$b$ is an a priori bound on the input samples' $L^2$ norm and where*

$$\widetilde{\mathcal{R}}_{F_A} :=$$

$$\left[ \sum_{\ell=1}^{L-1} \left[ b\rho_L \|A_L^\top\|_{2,\infty} \prod_{i=1}^{L-1} \rho_i \|A_i\| \right]^{\frac{2p_\ell}{p_\ell+2}} \left[ \frac{\|A_\ell - M_\ell\|_{\mathrm{sc},p_\ell}^{p_\ell}}{\|A_\ell\|^{p_\ell}} \right]^{\frac{2}{p_\ell+2}} \bar{w}_\ell^{\frac{2}{p_\ell+2}} [w_\ell \underline{w}_\ell]^{\frac{p_\ell}{p_\ell+2}} \right.$$

$$\left. + \left[ b\rho_L \|A_L^\top\|_{2,\infty} \prod_{i=1}^{L-1} \rho_i \|A_i\| \right]^{\frac{2p_L}{p_L+2}} \left[ \frac{\|A_L - M_L\|_{\mathrm{sc},p_L}^{p_L}}{\|A_L^\top\|_{2,\infty}^{p_L}} \right]^{\frac{2}{p_L+2}} \bar{w}_L^{\frac{2}{p_L+2}} \underline{w}_L^{\frac{p_L}{p_L+2}} \right]^{\frac{1}{2}}.$$

*Furthermore, the result also holds with $b$ replaced by the empirical quantity $\tilde{b} = \max_{i \leqslant N} \|x_i\|$.*

*Proof.* Let $\Delta_1, \Delta_2 > 0$ be positive real numbers to be determined later. For every set of values $k^{(b)}, k^{(\mathcal{M}_\ell)}, k^{(s_\ell)} \in \mathbb{Z}$ and $1 \leqslant k^{(p_\ell)} \leqslant \frac{2}{\Delta}$ (where $\ell = 1, \dots, L$) we apply proposition E.11 for the following values of $b, k^{(s_\ell)}, \mathcal{M}_\ell, p_\ell, \delta$:

$$b \leftarrow \tilde{b} = \exp(\Delta_1 k^{(b)}) \tag{E.8}$$

$$s_\ell \leftarrow \tilde{s}_\ell = \exp(\Delta_1 k^{(s_\ell)}/L) \tag{E.9}$$

$$\mathcal{M}_\ell \leftarrow \tilde{\mathcal{M}}_\ell = \tilde{s}_\ell \tilde{r}_\ell^{\frac{1}{p}} \quad \text{with} \tag{E.10}$$

$$\tilde{r}_\ell := \exp(\Delta_1 k^{(\mathcal{M}_\ell)}) \tag{E.11}$$

$$p_\ell \leftarrow \tilde{p}_\ell := \Delta_2 k^{(p_\ell)} \tag{E.12}$$

$$\delta_{k^{(b)}, k^{(s_\ell)}, k^{(\mathcal{M}_\ell)}, k^{(p_\ell)}} = \delta \Delta_2^L \frac{1}{3^{3L+1}} 2^{-|k^{(b)}| - \sum_\ell |k^{(s_\ell)}| - \sum_\ell |k^{(\mathcal{M}_\ell)}|}. \tag{E.13}$$

This implies that with probability greater than $1 - \delta_{k^{(b)}, k^{(s_\ell)}, k^{(\mathcal{M}_\ell)}, k^{(p_\ell)}}$,

$$\mathbb{E}\left[ \mathbb{1}(F_A(x), y) \right] - \frac{1}{N} \sum_{i=1}^{N} \mathbb{1}(F_A(x_i), y_i) \leqslant 6\mathcal{B} \sqrt{ \frac{\log\left( \frac{4}{\delta_{k^{(b)}, k^{(s_\ell)}, k^{(\mathcal{M}_\ell)}, k^{(p_\ell)}}} \right)}{N} }$$

$$+ \frac{240 \sqrt{\log_2(4\underline{\Gamma}_{k^{(b)}, k^{(s_\ell)}})} \log(N)[\mathcal{B}+1]}{\sqrt{N}} (1 + L\,\mathrm{L_l})^{\frac{p}{2+p}} \mathcal{R}_{k^{(p_\ell)}, k^{(\mathcal{M}_\ell)}, k^{(b)}, k^{(s_\ell)}}, \tag{E.14}$$

where $\mathcal{R}_{k^{(p_\ell)}, k^{(\mathcal{M}_\ell)}, k^{(b)}, k^{(s_\ell)}}$ is defined as the value of $\mathcal{R}_{\mathcal{M}, s, p, b}$ for $b, s_\ell, \mathcal{M}_\ell, p_\ell$ taking the values defined in equation (E.8) and $\underline{\Gamma}_{k^{(b)}, k^{(s_\ell)}} = \left[ [\exp(k^{(b)}\Delta) + 1][\mathrm{L_l} + 1]\overline{W} N \prod_{\ell=1}^{L} [\rho_\ell \exp(k^{(s_\ell)}\Delta) + 1] \right]$.

Note that

$$\sum_{k^{(b)}, k^{(s_\ell)}, k^{(\mathcal{M}_\ell)}, k^{(p_\ell)}} \delta_{k^{(b)}, k^{(s_\ell)}, k^{(\mathcal{M}_\ell)}, k^{(p_\ell)}} = \delta \Delta_2^L \frac{1}{3^{3L+1}} \left[ \frac{2}{\Delta_2} \right]^L [\sum_{i \in \mathbb{Z}} \frac{1}{2^i}]^{2L+1} \leqslant \frac{2}{3}\delta < \delta. \tag{E.15}$$

Thus, equation (E.14) holds simultaneously over all admissible values of $k^{(b)}, k^{(s_\ell)}, k^{(\mathcal{M}_\ell)}, k^{(p_\ell)}$ with probability $\geqslant 1 - \delta$.

For a given neural network $F_A$ and choice of $p_\ell$s, we utilise the result above for the following values of $k^{(b)}, k^{(s_\ell)}, k^{(\mathcal{M}_\ell)}, k^{(p_\ell)}$:

$$k^{(b)} := \lceil \log(b)/\Delta_1 \rceil \tag{E.16}$$

$$k^{(s_\ell)} = \lceil \log(s_\ell)L/\Delta_1 \rceil$$

$$k^{(\mathcal{M}_\ell)} = \lceil \log(\mathrm{r}_\ell)/\Delta_1 \rceil$$

$$k^{(p_\ell)} := \left\lceil \frac{p_\ell}{\Delta_2} \right\rceil$$

where $b \leftarrow \max_i \|x_i\|$, $s_\ell \leftarrow \|A_\ell\|$, $r_\ell \leftarrow \|A_\ell - M_\ell\|_{\mathrm{sc},p_\ell}^{p_\ell}/\|A\|^{p_\ell} = \sum_i \sigma_i(A - M_\ell)^{p_\ell}$ (for $\ell \neq L$) and $r_L \leftarrow \|A_L - M_L\|_{\mathrm{sc},p_L}^{p_L}/\|A_L^\top\|_{2,\infty}^{p_L} \leqslant \sqrt{w_L} \sum_i \sigma_i(A_L - M_L)^{p_L}$.

For these values, we certainly have:

$$\underline{\Gamma}_{k^{(b)},k^{(s_\ell)}} \leqslant \left[ [\exp(\Delta_1)b + 1][L_l + 1]\overline{W}N \prod_{i=1}^{L} [\rho_i s_i \exp(\Delta_1/L) + 1] \right] \tag{E.17}$$

$$\leqslant \exp(2\Delta_1) \left[ [b + 1][L\,L_l + 1]\overline{W}N \prod_{i=1}^{L} [[\rho_i + 1]s_i + 1] \right] = \exp(2\Delta_1)\underline{\Gamma}, \tag{E.18}$$

because $\underline{\Gamma} := \left[ [b + 1][L\,L_l + 1]\overline{W}N \prod_{i=1}^{L} [[\rho_i + 1]s_i + 1] \right]$. Further, we have

$$\tilde{r}_\ell \leqslant r_\ell\, e^{\Delta_1}. \tag{E.19}$$

Note also that

$$\frac{2\tilde{p}_\ell}{\tilde{p}_\ell + 2} - \frac{2p_\ell}{p_\ell + 2} \leqslant \frac{2[\tilde{p}_\ell - p_\ell]}{\tilde{p}_\ell + 2} \leqslant 2[\tilde{p}_\ell - p_\ell]/2 \leqslant \Delta_2 \tag{E.20}$$

and

$$\frac{2}{\tilde{p}_\ell + 2} - \frac{2}{p_\ell + 2} = \frac{2[\tilde{p}_\ell - p_\ell]}{[\tilde{p}_\ell + 2][p_\ell + 2]} \leqslant \frac{2\Delta_2}{4} \leqslant \Delta_2/2. \tag{E.21}$$

Thus, we can continue:

$$\mathcal{R}_{k^{(p_\ell)},k^{(\mathcal{M}_\ell)},k^{(b)},k^{(s_\ell)}} \leqslant \tag{E.22}$$

$$\left[ \sum_{\ell=1}^{L} \left[ \exp(\Delta_1)b \prod_i \rho_i s_i \exp(\Delta_1/L) \right]^{\frac{2\tilde{p}_\ell}{\tilde{p}_\ell + 2}} \tilde{r}_\ell^{\frac{2}{\tilde{p}_\ell + 2}} \bar{w}_\ell^{\frac{2}{\tilde{p}_\ell + 2}} \tilde{w}_\ell^{\frac{\tilde{p}_\ell}{\tilde{p}_\ell + 2}} \right]^{\frac{1}{2}} \tag{E.23}$$

$$\leqslant \exp(\Delta_1) \left[ \sum_{\ell=1}^{L} \left[ b \prod_i \rho_i s_i \right]^{\frac{2\tilde{p}_\ell}{\tilde{p}_\ell + 2}} \tilde{r}_\ell^{\frac{2}{\tilde{p}_\ell + 2}} \bar{w}_\ell^{\frac{2}{\tilde{p}_\ell + 2}} \tilde{w}_\ell^{\frac{\tilde{p}_\ell}{\tilde{p}_\ell + 2}} \right]^{\frac{1}{2}} \tag{E.24}$$

$$\leqslant \left[ \exp(\Delta_1) \left[ b \prod_i \rho_i s_i \right]^{\Delta_2/2} \right] \left[ \sum_{\ell=1}^{L} \left[ b \prod_i \rho_i s_i \right]^{\frac{2p_\ell}{p_\ell + 2}} \tilde{r}_\ell^{\frac{2}{\tilde{p}_\ell + 2}} \bar{w}_\ell^{\frac{2}{\tilde{p}_\ell + 2}} \tilde{w}_\ell^{\frac{\tilde{p}_\ell}{\tilde{p}_\ell + 2}} \right]^{\frac{1}{2}} \tag{E.25}$$

$$\leqslant \left[ e^{\Delta_1} \left[ b \prod_i \rho_i s_i \right]^{\frac{\Delta_2}{2}} e^{\Delta_1/2} \right] \left[ \sum_{\ell=1}^{L} \left[ b \prod_i \rho_i s_i \right]^{\frac{2p_\ell}{p_\ell + 2}} [r_\ell\, \bar{w}_\ell]^{\frac{2}{\tilde{p}_\ell + 2}} \tilde{w}_\ell^{\frac{\tilde{p}_\ell}{\tilde{p}_\ell + 2}} \right]^{\frac{1}{2}} \tag{E.26}$$

$$\leqslant \left[ e^{1.5\Delta_1} \left[ b \prod_i \rho_i s_i \right]^{\frac{\Delta_2}{2}} \right] \left[ \sum_{\ell=1}^{L} \left[ b \prod_i \rho_i s_i \right]^{\frac{2p_\ell}{p_\ell + 2}} r_\ell^{\frac{2}{\tilde{p}_\ell + 2}} \bar{w}_\ell^{\frac{2}{\tilde{p}_\ell + 2}} \tilde{w}_\ell^{\frac{p_\ell}{p_\ell + 2}} \right]^{\frac{1}{2}} \tag{E.27}$$

$$\leqslant \left[ e^{1.5\Delta_1} \left[ b \prod_i \rho_i s_i \right]^{\frac{\Delta_2}{2}} \bar{w}_\ell^{\frac{\Delta_2}{2}} \right] \left[ \sum_{\ell=1}^{L} \left[ b \prod_i \rho_i s_i \right]^{\frac{2p_\ell}{p_\ell + 2}} [r_\ell\, \bar{w}_\ell]^{\frac{2}{p_\ell + 2}} \tilde{w}_\ell^{\frac{p_\ell}{p_\ell + 2}} \right]^{\frac{1}{2}} \tag{E.28}$$

$$\leqslant \left[ e^{1.5\Delta_1} \left[ b \prod_i \rho_i s_i \right]^{\frac{\Delta_2}{2}} \bar{w}_\ell^{\frac{\Delta_2}{2}} \right] \widetilde{\mathcal{R}}_{F_A} \tag{E.29}$$

$$\leqslant \left[ 4\overline{W} + b \prod_i \rho_i s_i \right]^{\max(\Delta_1,\Delta_2)} \widetilde{\mathcal{R}}_{F_A} \tag{E.30}$$

where at equation (E.25) we have used equation (E.20), at equation (E.26) we have used equation (E.19), at equation (E.27) we have used the trivial fact that $\frac{\tilde{p}_\ell}{\tilde{p}_\ell + 2} \geqslant \frac{p_\ell}{2 + p_\ell}$ and at equation (E.28) we have used equation (E.21) and the fact that $r_\ell \leqslant \bar{w}_\ell$.

Equation (E.30) motivates the choice

$$\Delta_1 = \Delta_2 = \frac{\log(2)}{\log\left[4\overline{W} + b\prod_i \rho_i s_i\right]}, \tag{E.31}$$

which ensures that

$$\mathcal{R}_{k^{(p_\ell)},k^{(\mathcal{M}_\ell)},k^{(b)},k^{(s_\ell)}} \leqslant 2\widetilde{\mathcal{R}}_{F_A}. \tag{E.32}$$

Note that for this choice of $\Delta_1$ and $\Delta_2$, we can write first (from equation (E.17),

$$\underline{\Gamma}_{k^{(b)},k^{(s_\ell)}} \leqslant \underline{\Gamma}4\left[4\overline{W} + b\prod_i \rho_i \|A_\ell\|\right] \tag{E.33}$$

and then:

$$
\log\left(\frac{1}{\delta_{k^{(b)},k^{(s_\ell)},k^{(\mathcal{M}_\ell)},k^{(p_\ell)}}}\right)
$$

$$
\leqslant \log(\frac{1}{\delta}) + [3L+1]\log(3) + \left[|k^{(b)}| + \sum_\ell |k^{(s_\ell)}| + \sum_\ell |k^{(\mathcal{M}_\ell)}|\right] + L\log(1/\Delta_2)
$$

$$
\leqslant \log(\frac{1}{\delta}) + L\left[4.3 + \log\log\left[4\overline{W} + b\prod_i \rho_i s_i\right]\right] + \left[|k^{(b)}| + \sum_\ell |k^{(s_\ell)}| + \sum_\ell |k^{(\mathcal{M}_\ell)}|\right] \tag{E.34}
$$

$$
\leqslant \log(\frac{1}{\delta}) + L\left[4.3 + \log\log\left[4\overline{W} + b\prod_i \rho_i \|A_\ell\|\right]\right] + 2 \tag{E.35}
$$

$$
+ \left[|\log(b)| + L\left[\sum_{\ell=1}^{L} |\log(\|A_\ell\|)| + \log(\overline{W})\right]\right]/\Delta_1 \tag{E.36}
$$

$$
\leqslant \log(\frac{1}{\delta}) + L\left[4.3 + \log\left[4\overline{W} + b\prod_i \rho_i \|A_\ell\|\right]\right] + 2 \tag{E.37}
$$

$$
+ 2\log\left[4\overline{W} + b\prod_i \rho_i \|A_\ell\|\right]\left[|\log(b)| + L\left[\sum_{\ell=1}^{L} |\log(\|A_\ell\|)| + \log(\overline{W})\right]\right]
$$

$$
\leqslant \log(\frac{1}{\delta}) + 2 + 2\log\left[4\overline{W} + b\prod_i \rho_i \|A_\ell\|\right]\left[|\log(b)| + L\left[\sum_{\ell=1}^{L} |\log(\|A_\ell\|)| + 2\log(4\overline{W})\right]\right]
$$

$$
= \log(\frac{1}{\delta}) + 2 + \Theta_{\log} \tag{E.38}
$$

where at equation (E.34) we have used the definition of $\Delta_2$ (i.e. equation (E.31)) and the fact $\log(\log(2)) \geqslant -1$, at equation (E.36) we have used equations (E.16) , and at equation (E.37) we have used the fact that $\log(2) \geqslant \frac{1}{2}$ and the inequality $\log\log(x) \leqslant \log(x)$ and at equation (E.38) we have used the definition of $\Theta_{\log}$.

We now plug equations (E.30), (E.32), (E.38) and (E.33) into equation (E.14) to obtain:

$$\mathbb{E}\left[\mathrm{l}(F_A(x), y)\right] - \frac{1}{N}\sum_{i=1}^{N}\mathrm{l}(F_A(x_i), y_i) \leqslant 6[\mathcal{B}+1]\sqrt{\frac{\log\left(\frac{4}{\delta_{k^{(b)}, k^{(s_\ell)}, k^{(\mathcal{M}_\ell)}, k^{(p_\ell)}}}\right)}{N}}$$

$$+ \frac{240\sqrt{\log_2(2\underline{\Gamma}_{k^{(b)}, k^{(s_\ell)}})}\log(N)[\mathcal{B}+1]}{\sqrt{N}}(1 + L\,\mathrm{L_l})^{\frac{p}{2+p}}\mathcal{R}_{k^{(p_\ell)}, k^{(\mathcal{M}_\ell)}, k^{(b)}, k^{(s_\ell)}}$$

$$\leqslant 6[\mathcal{B}+1]\sqrt{\frac{\log(1/\delta)}{N}} + 6[\mathcal{B}+1]\sqrt{\frac{2 + \Theta_{\log}}{N}}$$

$$+ 480\,\mathcal{B}(1 + L\,\mathrm{L_l})^{\frac{p}{2+p}}\frac{\sqrt{\log_2(8\underline{\Gamma}\left[4\overline{W} + b\prod_i \rho_i\|A_\ell\|\right])}\log(N)}{N}\widetilde{\mathcal{R}}_{F_A} \qquad (E.39)$$

$$\leqslant 6[\mathcal{B}+1]\sqrt{\frac{\log(1/\delta)}{N}} + 6[\mathcal{B}+1]\sqrt{\frac{2 + \Theta_{\log}}{N}} + 480\,\mathcal{B}(1 + L\,\mathrm{L_l})^{\frac{p}{2+p}} \times$$

$$\frac{\sqrt{\log(12\left[[b+1][L\,\mathrm{L_l}+1]\overline{W}N\prod_{i=1}^{L}[[\rho_i + 1]s_i + 1]\right]\left[4\overline{W} + b\prod_i \rho_i\|A_\ell\|\right])}\log(N)}{N}\widetilde{\mathcal{R}}_{F_A}$$

$$\leqslant 6[\mathcal{B}+1]\sqrt{\frac{\log(1/\delta)}{N}} + 6[\mathcal{B}+1]\sqrt{\frac{2 + \Theta_{\log}}{N}}$$

$$+ 480\,\mathcal{B}(1 + L\,\mathrm{L_l})^{\frac{p}{2+p}}\frac{\sqrt{\log(\gamma_{FC})}\log(N)}{N}\widetilde{\mathcal{R}}_{F_A}, \qquad (E.40)$$

where at line (E.39) we have used equation (E.32), and at the last line we have used the definition

$$\gamma_{FC} := 12\left[[b+1][L\,\mathrm{L_l}+1]\overline{W}N\prod_{i=1}^{L}[\rho_i\|A_i\| + 1]\right]\left[4\overline{W} + b\prod_i \rho_i\|A_i\|\right].$$

This concludes the proof of the first statement.

For the second statement, it suffices to use an elementary form of loss function augmentation by considering the following augmented loss for any value of $b$ greater than 1: $l'(x, y) = \max(\mathrm{l}(F_A(x), y), \mathcal{B}\,1_{\|x\|>b})$. Indeed, given any training set $x_1, \ldots, x_N$ (not necessarily satisfying $\|x_i\| \leqslant b$), we can apply Proposition E.11 to the training set $\{x_i : \|x_i\| \leqslant b\}$, which results in a cover of $l' \circ \mathcal{F}_\mathcal{B}$ with respect to the whole set $\{x_i\}$ since $l' = \mathcal{B}$ whenever $\|x_i\| \geqslant b$. The cover is bounded above by the RHS of equation (E.131) and we can continue the argument as in the proof above and conclude that

$$\mathbb{E}\left[l'(F_A(x), y)\right] - \frac{1}{N}\sum_{i=1}^{N}l'(F_A(x_i), y_i) \leqslant \mathcal{Q}, \qquad (E.41)$$

where $\mathcal{Q}$ is the right hand side of equation E.40. Note that the argument above already uses a posthoc argument to show that the bound holds for all values of $b$ simultaneously with a single failure probability. Thus, the same is true for equation (E.41). Note that it is not necessary for $l'$ to be Lipschitz in the input $x$, since no cover of input space is required.

Next, we have:

$$\mathbb{E}\left[\mathrm{l}(F_A(x), y)\right] - \frac{1}{N}\sum_{i=1}^{N}\mathrm{l}(F_A(x_i), y_i)$$

$$\leqslant \mathbb{E}\left[l'(F_A(x), y)\right] - \frac{1}{N}\sum_{i=1}^{N}\mathrm{l}(F_A(x_i), y_i) \qquad (E.42)$$

$$\leqslant \mathcal{Q} + \frac{1}{N}\sum_{i=1}^{N}l'(F_A(x_i)) - \frac{1}{N}\sum_{i=1}^{N}\mathrm{l}(F_A(x_i), y_i) \qquad (E.43)$$

$$\leqslant \mathcal{Q}. \qquad (E.44)$$

where at the second line E.42, we have used the fact that $l' \geqslant l$, at the third line (E.43) we have used equation (E.41) and at the fourth line (E.44) we have used the fact that $b = \max_i \|x_i\|$. This completes the proof.

$\square$

## E.1 Tighter Results with Loss Function Augmentation

In this section, we use the technique of Loss Function Augmentation to improve the scaling of the bound with respect to norm based quantities.

We first recall the following definition of the ramp loss (defined for any B), which we use in controlling the intermediary activations of the neural network:

$$\lambda_B(x) = \min(\mathcal{B}\frac{(x - B)_+}{B}, \mathcal{B}). \tag{E.45}$$

We now consider the following augmented loss function:

$$l_{aug}(F_A, x) := \max\left(\max_{\ell \leqslant L-1}\left[\lambda_{b_\ell}(\|F_A^{0 \to \ell}(x)\|)\right], 1(F_A(x), y), \mathcal{B}\,1_{\|x\| > b_0}\right). \tag{E.46}$$

We first prove the following extension of Lemma E.12 to incorporate the loss function augmentation over activations.

**Lemma E.3.** *Assume $s_\ell \geqslant 1$ and let $b_\ell$ (for $\ell = 0, 1, \dots, L$) be some positive real numbers with $b_L = 1/\mathrm{L_l}$. Define $\hat{\rho}_\ell = \sup_{u \geqslant l} \rho_{l \to u}/b_u$ where as before, $\rho_{l \to u} = \rho_l \prod_{m=l+1}^u s_m \rho_m$.*

*Assume as in Lemma E.12 that we have a fixed dataset $x_1, \dots, x_N$ with $\|x_i\| \leqslant b_0$ for all $i \leqslant N$. Let $\mathcal{B}_1, \dots, \mathcal{B}_L$ be arbitrary subsets of the spaces $\mathbb{R}^{w_\ell \times w_{\ell-1}}$ and instantiate the rest of the notations from Lemma E.12, and explicitly set $\epsilon_\ell = \frac{b_\ell \epsilon}{L \hat{\rho}_\ell}$.*

*There exist covers $\mathcal{C}_1 \subset \mathcal{B}_1, \dots, \mathcal{C}_{0 \to \ell} \subset \mathcal{B}_1 \times \dots \mathcal{B}_\ell, \dots, \mathcal{C}_L \subset \mathcal{B}_1 \times \dots \mathcal{B}_L$ such that for all $A = (A_1, \dots, A_L) \in \mathcal{B}_1 \times \dots \mathcal{B}_L$, there exist $\bar{A}^1, \dots, \bar{A}^L$ such that for all $\ell \leqslant L$, $\bar{A} = \bar{A}^{0 \to L} := (\bar{A}^1, \dots, \bar{A}^L) \in \mathcal{C}_{0 \to L}$ and for all $i \leqslant N$, $\ell \leqslant L$:*

$$|\lambda_{aug}(F_A, x_i, y) - \lambda_{aug}(F_{\bar{A}}, x_i, y)| \leqslant \epsilon. \tag{E.47}$$

*Furthermore the covers satisfy the following covering number bound:*

$$\log(|\mathcal{C}_{0 \to L}|) \leqslant \sum_{\ell=1}^L \log(\mathcal{N}_{\infty,\ell}(\mathcal{B}_\ell, \epsilon_\ell, N, 3b_{\ell-1})), \tag{E.48}$$

*where as before, for any $\bar{b} > 0$ and $\bar{N} \in \mathbb{N}$, $\mathcal{N}_{\infty,\ell}(\mathcal{B}_\ell, \epsilon_\ell, N, \bar{b})$ is defined as the minimum number $\mathcal{N}$ such that for any dataset $x_1^{\ell-1}, \dots, x_{\bar{N}}^{\ell-1}$ with $\bar{N} \leqslant N$ and $\|x_i^{\ell-1}\|_\ell \leqslant \bar{b}$ for all $i \leqslant \bar{N}$, where $\|\cdot\|_\ell = \|\cdot\|$ for $\ell \neq L$ and $\|\cdot\|_L = \|\cdot\|_\infty$.*

*Proof.* The proof is quite similar to the proof of Proposition 8 in [42], itself inspired from [74, 5, 16].

We will construct the covers such that they satisfy the following claim:

*Claim 1:* There exists a cover $\mathcal{C}_{0 \to L}$ satisfying equation (E.48) such that for any $A = (A^1, \dots, A^L) \in \mathcal{B}_1 \times \dots \mathcal{B}_L$, there exists a $\bar{A}^1, \dots, \bar{A}^L$ such that for each $x \in X$ and each $\ell \leqslant L$, one of the following two conditions is satisfied:

$$\forall \ell_1 \leqslant \ell, \|F_{\bar{A}^1, \dots, \bar{A}^{\ell_1-1}, A^{\ell_1}}^{0 \to \ell_1}(x) - F_{\bar{A}^1, \dots, \bar{A}^{\ell_1-1}, \bar{A}^{\ell_1}}^{0 \to \ell}(x)\|_\ell \leqslant \epsilon_\ell \rho_\ell \tag{E.49}$$

or there exists $\ell_1 < \ell$ such that

$$\|F_{\bar{A}^1, \dots, \bar{A}^{\ell_1}}^{0 \to \ell_1}(x)\|_\ell > 3b_{\ell_1} \tag{E.50}$$

(Here as usual $\|\cdot\|_\ell = \|\cdot\|_2$ if $\ell \neq L$ and $\|\cdot\|_L = \|\cdot\|_\infty$).

*Proof of Claim 1:*

For the **first layer**, writing $X := \{x_1, \ldots, x_N\}$, we know by assumption that there exists a cover $\mathcal{C}_1(X) = \mathcal{C}_{0\to1} \subset \mathcal{B}_1$ such that for all $A_1 \in \mathcal{B}_1$ , there exists a $\bar{A}_1 \in \mathcal{C}_1(X_0)$ such that for all $i \leqslant N$,

$$\|(A^1 - \bar{A}^1)(x_i)\| \leqslant \epsilon_1, \tag{E.51}$$

from which it also follows that

$$\|F_{A^1}^{0\to1}(x_1) - F_{\bar{A}^1}^{0\to1}(x_1)\| = \|\sigma_1(A^1 x_i) - \sigma_1(\bar{A}^1(x_i))\| \leqslant \epsilon_1 \rho_1. \tag{E.52}$$

It follows that $\left\{F_{A^1}^{0\to1} : A^1 \in \mathcal{B}_1\right\} \subset \mathcal{F}_{\mathcal{B}}^{0\to1}$ is an $L^{\infty,2}$ cover of $\mathcal{F}_{\mathcal{B}}^{0\to1}$ as required: equation (E.49) holds for $\ell = 1$.

For the **inductive case**:

Assume that the covers $\mathcal{C}_{0\to1}, \ldots, \mathcal{C}_{0\to\ell}$ have been constructed and satisfy the claim up to and including layer $\ell$.

For each element $\bar{A}^{0\to\ell}$ of $\mathcal{C}_{0\to\ell}$, we can split the dataset $X$ into two sets $X_{\ell, \bar{A}^{0\to\ell}} \subset X$ and $[X_{\ell, \bar{A}^{0\to\ell}}]^c$ as follows:

$$X_{\ell, \bar{A}^{0\to\ell}} = \left\{x \in X : \forall \ell_1 \leqslant \ell : \|F_{\bar{A}^1, \ldots, \bar{A}^\ell}^{0\to\ell_1}(x_i)\|_{\ell_1} \leqslant 3b_{\ell_1}\right\}. \tag{E.53}$$

Next, by the definition of $\mathcal{N}_{\infty,\ell}(\mathcal{B}_\ell, \epsilon_\ell, N, 3b_{\ell-1})$, there exists a cover $\mathcal{C}_{\ell+1, \bar{A}^{0\to\ell}} \subset \mathcal{B}_{\ell+1}$ such that for any $A^{\ell+1} \in \mathcal{B}_{\ell+1}$, there exists $\bar{A}^{\ell+1} \in \mathcal{C}_{\ell, \bar{A}^{0\to\ell}}$ such that for any $x \in X_{\ell, \bar{A}^{0\to\ell}}$,

$$\|A^{\ell+1} F_{\bar{A}^{0\to\ell}}^{0\to\ell}(x) - \bar{A}^{\ell+1} F_{\bar{A}^{0\to\ell}}^{0\to\ell}(x)\|_\ell \leqslant \epsilon_\ell. \tag{E.54}$$

Accordingly, we define the cover

$$\mathcal{C}_{0\to\ell+1} = \bigcup_{\bar{A}^{0\to\ell} \in \mathcal{C}_\ell} \left\{\bar{A}^{0\to\ell}\right\} \times \mathcal{C}_{\ell+1, \bar{A}^{0\to\ell}}. \tag{E.55}$$

For any $A^{0\to\ell+1} = (A^1, \ldots, A^{\ell+1}) \in \mathcal{B}_1 \times \ldots \times \mathcal{B}_{\ell+1}$, the associated cover element $\bar{A}^{0\to\ell+1}$ is defined by $(\bar{A}^1, \ldots, \bar{A}^\ell, \bar{A}^{\ell+1})$ where $(\bar{A}^1, \ldots, \bar{A}^\ell)$ is the cover element of $\mathcal{C}_{0\to\ell}$ associated to $(A^1, \ldots, A^\ell)$ defined by the induction hypothesis, and $\bar{A}^{\ell+1}$ is the cover element of $\mathcal{C}_{\ell+1}$ associated to $A^{\ell+1}$ by the definition of $\mathcal{N}_{\infty,\ell}(\mathcal{B}_\ell, \epsilon_\ell, N, 3b_{\ell-1})$ to satisfy Condition (E.54).

Equation (E.54) implies that for any $x \in X_{\ell, \bar{A}^{0\to\ell}}$, condition (E.49) holds for the value $\ell_1 = \ell$. In addition, we already know from the induction hypothesis that the same condition holds for $\ell_1 < \ell$. Therefore, the condition holds for all $\ell_1 < \ell + 1$ for all $x \in X_{\ell, \bar{A}^{0\to\ell}}$. Furthermore, by definition of $X_{\ell, \bar{A}^{0\to\ell}}$, any $x \in [X_{\ell, \bar{A}^{0\to\ell}}]^c$ has to satisfy equation (E.50) for some $\ell_1 < \ell + 1$. Thus, every $x \in X$ indeed satisfies either condition (E.49) or condition (E.50), as desired.

This concludes the proof of the *claim*.

To finish the proof of the *lemma*, it only remains to prove equation E.47. Consider an arbitrary $A \in \mathcal{B}_1 \times \ldots \mathcal{B}_L$ and the associated cover element $\bar{A} = (\bar{A}^1, \bar{A}^2, \ldots, \bar{A}^L)$.

Let $x \in X$ be an arbitrary sample. Let $\ell^*$ be the smallest number less than $L - 1$ such that

$$\|F_{\bar{A}^1, \ldots, \bar{A}^\ell}^{0\to\ell}(x)\|_\ell > 3b_\ell. \tag{E.56}$$

and let $\ell^* = L$ if the above equation doesn't hold for any value of $\ell$.

By instantiating the claim for any $\ell \leqslant \ell^*$, we know that equation (E.49) holds for all $\ell_1 \leqslant \ell$. Thus,

$$\left| \lambda_{b_\ell}(\|F^{0\to\ell}_{A^0\to\ell}(x)\|) - \lambda_{b_\ell}(\|F^{0\to\ell}_{\bar{A}^0\to\ell}(x)\|) \right| \tag{E.57}$$

$$\leqslant \frac{1}{b_\ell} \left\| F^{0\to\ell}_{A^0\to\ell}(x) - F^{0\to\ell}_{\bar{A}^0\to\ell}(x) \right\| \tag{E.58}$$

$$\leqslant \frac{1}{b_\ell} \sum_{\ell_1=1}^{\ell} \left\| F^{0\to\ell}_{\bar{A}^1,\ldots,\bar{A}^{\ell_1},A^{\ell_1+1},\ldots A^\ell}(x) - F^{0\to\ell}_{\bar{A}^1,\ldots,\bar{A}^{\ell_1-1},A^{\ell_1},\ldots A^\ell}(x) \right\| \tag{E.59}$$

$$\leqslant \frac{1}{b_\ell} \sum_{\ell_1=1}^{\ell} \epsilon_{\ell_1} \rho_{\ell_1\to\ell} \tag{E.60}$$

$$= \frac{1}{b_\ell} \sum_{\ell_1=1}^{\ell} \frac{\epsilon}{L\rho_{\hat{\rho}_{\ell_1}}} \rho_{\ell_1\to\ell} \tag{E.61}$$

$$\leqslant \frac{1}{b_\ell} \sum_{\ell_1=1}^{\ell} \frac{\epsilon}{L\rho_{\ell_1\to\ell}/b_\ell} \rho_{\ell_1\to\ell} \leqslant \frac{\ell}{L}\epsilon \leqslant \epsilon, \tag{E.62}$$

where at equation (E.60) we have used the claim, at line (E.61) we have used the definition of $\epsilon_{\ell_1}$, and at line (E.62) we have used the definition of $\hat{\rho}_{\ell_1}$.

In particular, it certainly follows that if $\ell \leqslant L-1$,

$$\left\| F^{0\to\ell}_{A^0\to\ell}(x) \right\| \geqslant \left\| F^{0\to\ell}_{\bar{A}^0\to\ell}(x) \right\| - \left\| F^{0\to\ell}_{A^0\to\ell}(x) - F^{0\to\ell}_{\bar{A}^0\to\ell}(x) \right\| > 3b_\ell - b_\ell = 2b_\ell, \tag{E.63}$$

which implies that $\lambda_{aug}(F_A,x,y) = \lambda_{aug}(F_{\bar{A}},x,y) = 0$ and equation (E.47) indeed holds. On the other hand, if $\ell^* = L$, then since $b_L = 1/L_1$, equation (E.62) instantiated for $\ell = \ell^* = L$ also shows that

$$|\mathrm{l}(F_A(x),y) - \mathrm{l}(F_{\bar{A}}(x),y)| \leqslant \epsilon, \tag{E.64}$$

whilst (E.62) instantiated for any $\ell \leqslant L-1$ also shows that

$$\left| \max_{\ell\leqslant L-1} \left[ \lambda_{b_\ell}(\|F^{0\to\ell}_A(x)\|) \right] - \max_{\ell\leqslant L-1} \left[ \lambda_{b_\ell}(\|F^{0\to\ell}_{\bar{A}}(x)\|) \right] \right| \leqslant \epsilon. \tag{E.65}$$

Together, equations (E.64) and (E.65) imply that equation (E.47) holds, as expected.

$$\square$$

**Proposition E.4.** *For any granularity $\epsilon \geqslant \frac{1}{N}$ and any values of $1 \leqslant b_{\ell_1},\ldots b_{L-1}, s_\ell, \rho_\ell$, there exists a cover $\mathcal{C}$ of the augmented loss class $\{\lambda_{aug}(F_A,x,y) : A = (A^1,\ldots,A^L) \in \mathcal{B}_1 \times \ldots \times \mathcal{B}_L\}$ with cardinality bounded as follows:*

$$\log(|\mathcal{C}|) \leqslant 288\log_2(4\Gamma) \left[ \frac{L(1+L_1)}{\min(\epsilon,1)} \right]^{\frac{2p}{2+p}} \sum_{\ell=1}^{L} \left[ b_{\ell-1} \prod_{i=\ell}^{L} s_i\rho_i \right]^{\frac{2p_\ell}{p_\ell+2}} \mathrm{r}_\ell^{\frac{2}{p_\ell+2}} \bar{w}_\ell^{\frac{2}{p_\ell+2}} \tilde{w}_\ell^{\frac{p_\ell}{p_\ell+2}}. \tag{E.66}$$

*Proof.* The proof is analogous to that of proposition E.11 and consists in plugging in the one layer covering number estimates (Propositions (F.4) and (F.2)) into equation (E.48), taking into account our new definition of $\epsilon_\ell = \frac{b_\ell\epsilon}{L\hat{\rho}_\ell}$:

$$\log(|\mathcal{C}|) \leqslant \sum_{\ell=1}^{L} \log(\mathcal{N}_{\infty,\ell}(\mathcal{B}_\ell, \epsilon_\ell, N, 3b_{\ell-1}))$$

$$\leqslant 72 \left[ \sum_{\ell=1}^{L-1} \left[ \frac{\mathcal{M}_\ell b_{\ell-1}}{\epsilon_\ell} \right]^{\frac{2p_\ell}{p_\ell+2}} \bar{w}_\ell^{\frac{2}{p_\ell+2}} [w_\ell \underline{w}_\ell]^{\frac{p_\ell}{p_\ell+2}} \log_2(\Gamma_{\mathcal{F}^{\mathrm{p}}_{\mathrm{r}},\ell}) \right.$$

$$\tag{E.67}$$

$$\left. + \left[ \frac{\mathcal{M}_L b_{L-1}}{\epsilon_L} \right]^{\frac{2p_L}{p_L+2}} \bar{w}_L^{\frac{2}{p_L+2}} \underline{w}_L^{\frac{p_L}{p_L+2}} \log_2(\Gamma_{\mathcal{F}^{\mathrm{p}}_{\mathrm{r}},L}) \right]. \tag{E.68}$$

Note also that since we are assuming that $s_\ell \geqslant 1$ for all $\ell$ and $b_\ell \geqslant 1$ for all $\ell \neq L$, we also have $\epsilon_\ell = \frac{b_\ell \epsilon}{L \hat{\rho}_\ell} \geqslant \epsilon \rho_{\ell \to L}$, thus, the bounds on the logarithmic factors $\Gamma_{\mathcal{F}_r^p, L}, \Gamma_{\mathcal{F}_r^p, \ell}$ derived in equations (E.141) still hold. Thus, we can continue:

$$\log\left(|\mathcal{C}|\right) \leqslant 72 \log_2(\bar{\Gamma}) \sum_{\ell=1}^{L} \left[\frac{\mathcal{M}_\ell \, b_{\ell-1}}{\epsilon_\ell}\right]^{\frac{2 p_\ell}{p_\ell+2}} \bar{w}_\ell^{\frac{2}{p_\ell+2}} \tilde{w}_\ell^{\frac{p_\ell}{p_\ell+2}} \tag{E.69}$$

$$\leqslant 72 \log_2(\bar{\Gamma}) \sum_{\ell=1}^{L} \left[\frac{\mathcal{M}_\ell \, b_{\ell-1} \rho_\ell L[1 + \mathrm{L_l}] \prod_{i=\ell+1}^{L} s_i \rho_i}{\epsilon}\right]^{\frac{2 p_\ell}{p_\ell+2}} \bar{w}_\ell^{\frac{2}{p_\ell+2}} \tilde{w}_\ell^{\frac{p_\ell}{p_\ell+2}} \tag{E.70}$$

$$\leqslant 72 \log_2(\bar{\Gamma}) \left[\frac{L(1 + \mathrm{L_l})}{\min(\epsilon, 1)}\right]^{\frac{2p}{2+p}} \sum_{\ell=1}^{L} \left[\frac{\mathcal{M}_\ell}{s_\ell} b_{\ell-1} \prod_{i=\ell}^{L} s_i \rho_i\right]^{\frac{2 p_\ell}{p_\ell+2}} \bar{w}_\ell^{\frac{2}{p_\ell+2}} \tilde{w}_\ell^{\frac{p_\ell}{p_\ell+2}} \tag{E.71}$$

$$= 288 \log_2(4\underline{\Gamma}) \left[\frac{L(1 + \mathrm{L_l})}{\min(\epsilon, 1)}\right]^{\frac{2p}{2+p}} \sum_{\ell=1}^{L} \left[b_{\ell-1} \prod_{i=\ell}^{L} s_i \rho_i\right]^{\frac{2 p_\ell}{p_\ell+2}} \mathrm{r}_\ell^{\frac{2}{p_\ell+2}} \bar{w}_\ell^{\frac{2}{p_\ell+2}} \tilde{w}_\ell^{\frac{p_\ell}{p_\ell+2}}, \tag{E.72}$$

as expected.

$\square$

Then, we can proceed with the following straightforward analogue of Proposition E.1:

**Proposition E.5.** *Fix some reference/initialization matrices $M^1 \in \mathbb{R}^{w_1 \times w_0} = \mathbb{R}^{w_1 \times d}, \ldots, M^L \in \mathbb{R}^{w_L \times w_{L-1}} = \mathbb{R}^{C \times w_{L-1}}$. Assume also that the inputs $x \in \mathbb{R}^d$ satisfy $\|x\| \leqslant b$ w.p. 1. Fix a set of constraint parameters $0 \leqslant p_\ell \leqslant 2, 1 \leqslant s_\ell, \mathcal{M}_\ell$ (for $\ell = 1, \ldots, L$). Also fix some numbers $1 \leqslant b_{\ell_1}, \ldots b_{L-1}$ and $b_L = \frac{1}{\mathrm{L_l}}$.*

*With probability greater than $1 - \delta$ over the draw of an i.i.d. training set $x_1, \ldots, x_N$, every neural network $F_A(x) := A_L \sigma_L(A^{L-1} \sigma_{L-1}(\ldots \sigma_1(A^1 x) \ldots))$ satisfying the following conditions:*

$$\|A^\ell - M^\ell\|_{\mathrm{sc}, p_\ell} \leqslant \mathcal{M}_\ell \qquad \forall 1 \leqslant \ell \leqslant L$$
$$\|A\| \leqslant s_\ell \qquad \forall 1 \leqslant \ell \leqslant L - 1$$
$$\|A_L^\top\|_{2, \infty} \leqslant s_L \tag{E.73}$$
$$\|F_A^{0 \to \ell}(x_i)\| \leqslant b_\ell, \qquad \forall \ell \leqslant L - 1 \quad \forall i \leqslant N \tag{E.74}$$

*also satisfies the following generalization bound:*

$$\mathbb{E}\left[\mathrm{l}(F_A(x), y)\right] - \frac{1}{N} \sum_{i=1}^{N} \mathrm{l}(F_A(x_i), y_i)$$

$$\leqslant 6[\mathcal{B} + 1]\sqrt{\frac{\log(\frac{4}{\delta})}{N}} + \frac{416\sqrt{\log_2(4\underline{\Gamma})} \log(N)[\mathcal{B} + 1]}{\sqrt{N}} [L(1 + \mathrm{L_l})]^{\frac{p}{2+p}} \mathcal{R}_{\mathcal{M}, \mathrm{s}, p, b_\ell} \tag{E.75}$$

*where $\mathcal{R}_{\mathcal{M}, \mathrm{s}, p, b_\ell} :=$*

$$\left[\sum_{\ell=1}^{L-1} \left[b_{\ell-1} \prod_{i=\ell}^{L} \rho_i s_i\right]^{\frac{2 p_\ell}{p_\ell+2}} \mathrm{r}_\ell^{\frac{2}{p_\ell+2}} \bar{w}_\ell^{\frac{2}{p_\ell+2}} [w_\ell \underline{w}_\ell]^{\frac{p_\ell}{p_\ell+2}} + [b_{L-1} \rho_L]^{\frac{2 p_L}{p_L+2}} \mathrm{r}_L^{\frac{2}{p_L+2}} \bar{w}_L^{\frac{2}{p_L+2}} \underline{w}_L^{\frac{p_L}{p_L+2}}\right]^{\frac{1}{2}}, \tag{E.76}$$

*and as usual, $\underline{\Gamma} := \left[[b + 1][L \, \mathrm{L_l} + 1] \overline{W} N \prod_{i=1}^{L}[[\rho_i + 1] s_i + 1]\right]$ and we use the shorthands $\mathrm{r}_\ell := \frac{\mathcal{M}_\ell^p}{s_\ell^p}$, $\underline{w}_\ell := \min(w_\ell, w_{\ell-1})$, $\bar{w}_\ell := [w_\ell + w_{\ell-1}]$.*

*Proof.* We first note that with a calculation nearly identical to that of the proof of Proposition E.1 relying on Proposition E.4 instead of Proposition E.11 with an additional factor of $\sqrt{3}$ and $(1 + L \, \mathrm{L_l})$

replaced by $L(1 + \mathrm{L_l})$ as well as $b\prod_{i=1}^{L} s_i\rho_i$ replaced by $b_{\ell-1}\prod_{i=\ell}^{L} s_i\rho_l$, we obtain the following bound on the augmented loss class $\lambda_{aug}$:

$$\mathbb{E}\left[\lambda_{aug}(F_A, x, y)\right] - \frac{1}{N}\sum_{i=1}^{N}\lambda_{aug}(F_A, x_i, y_i)$$

$$\leqslant 6[\mathcal{B}+1]\sqrt{\frac{\log(\frac{4}{\delta})}{N}} + \frac{832\sqrt{\log_2(4\Gamma)}\log(N)[\mathcal{B}+1]}{\sqrt{N}}[L(1+\mathrm{L_l})]^{\frac{p}{2+p}}\mathcal{R}_{\mathcal{M},s,p,b_\ell}. \tag{E.77}$$

However, we know by definition of $\lambda_{aug}$ that

$$\mathbb{E}\left[\lambda_{aug}(F_A, x, y)\right] \geqslant \mathbb{E}\,\mathrm{l}(F_A(x), y). \tag{E.78}$$

Furthermore, since the last condition in (E.73) holds, we certainly also have:

$$\frac{1}{N}\sum_{i=1}^{N}\lambda_{aug}(F_A, x_i, y_i) = \frac{1}{N}\,\mathrm{l}(F_A(x_i), y_i). \tag{E.79}$$

The result now follows.

$\square$

We can now apply the above results to obtain a post hoc version using argument similar to those used in the proof of Theorem E.2

**Theorem E.6.** *With probability greater than $1 - \delta$, for any $b \in \mathbb{R}^+$, every trained neural network $F_A$ satisfies the following generalization gap for all possible values of the $p_\ell$s:*

$$\mathbb{E}\left[\mathrm{l}(F_A(x), y)\right] - \frac{1}{N}\sum_{i=1}^{N}\mathrm{l}(F_A(x_i), y_i) \tag{E.80}$$

$$\leqslant 6[\mathcal{B}+1]\sqrt{\frac{\log(1/\delta)}{N}} + 6\,\mathcal{B}\sqrt{\frac{1.5 + \Theta_{\log}^{\lambda_{aug}}}{N}}$$

$$+ 832[\mathcal{B}+1][L(1+\mathrm{L_l})]^{\frac{p}{2+p}}\frac{\sqrt{\log(\gamma_{FC})}\log(N)}{\sqrt{N}}\widetilde{\mathcal{R}}_{F_A,\lambda_{aug}},$$

*where $\Theta_{\log}^{\lambda_{aug}} :=$*

$$2\log\left[4\overline{W} + b\prod_i\rho_i\|A_\ell\|\right]\left[|\log(b)| + L\left[\sum_{\ell=1}^{L}|\log(\|A_\ell\|)| + \sum_{\ell=1}^{L-1}\log(\mathfrak{B}_{\ell,A}) + 2\log(4\overline{W})\right]\right],$$

*$\gamma_{FC} := 12\left[[b+1][L\,\mathrm{L_l}+1]\overline{W}N\prod_{i=1}^{L}[\rho_i\|A_i\| + 1]\right]\left[4\overline{W} + b\prod_i\rho_i\|A_i\|\right]$, $b = \max_{i=1}^{N}\|x_i\|$, and where*

$$\widetilde{\mathcal{R}}_{F_A,\lambda_{aug}} := \tag{E.81}$$

$$\left[\sum_{\ell=1}^{L-1}\left[\mathfrak{B}_{\ell-1,A}\rho_L\|A_L^\top\|_{2,\infty}\prod_{i=\ell}^{L-1}\rho_i\|A_i\|\right]^{\frac{2p_\ell}{p_\ell+2}}\left[\frac{\|A_\ell - M_\ell\|_{\mathrm{sc},p_\ell}^{p_\ell}}{\|A_\ell\|^{p_\ell}}\right]^{\frac{2}{p_\ell+2}}\bar{w}_\ell^{\frac{2}{p_\ell+2}}[w_\ell\underline{w}_\ell]^{\frac{p_\ell}{p_\ell+2}} \tag{E.82}$$

$$+\left[\mathfrak{B}_{L-1,A}\rho_L\|A_L^\top\|_{2,\infty}\right]^{\frac{2p_L}{p_L+2}}\left[\frac{\|A_L - M_L\|_{\mathrm{sc},p_L}^{p_L}}{\|A_L^\top\|_{2,\infty}^{p_L}}\right]^{\frac{2}{p_L+2}}\bar{w}_L^{\frac{2}{p_L+2}}\underline{w}_L^{\frac{p_L}{p_L+2}}\right]^{\frac{1}{2}}. \tag{E.83}$$

*where $\mathfrak{B}_{\ell-1,A} := \max\left(\max_{i\leqslant N}\|F_{A^1,\dots,A^{\ell-1}}(x_i)\|, 1\right)$.*

*Proof.* Let $\Delta_1, \Delta_2 > 0$ be positive real numbers to be determined later. For every set of values $k^{(b)}, k^{(b_\ell)}, k^{(\mathcal{M}_\ell)}, k^{(s_\ell)} \in \mathbb{Z}$ and $0 \leqslant k^{(p_\ell)} \leqslant \frac{2}{\Delta_2}$ (where $\ell = 1, \dots, L$) we apply proposition E.11

for the following values of $b, k^{(s_\ell)}, \mathcal{M}_\ell, p_\ell, \delta$:

$$b \leftarrow \tilde{b} = \exp(\Delta_1 k^{(b)}) \tag{E.84}$$

$$b_\ell \leftarrow \tilde{b}_\ell = \exp(\Delta_1 k^{(b_\ell)})$$

$$s_\ell \leftarrow \tilde{s}_\ell = \exp(\Delta_1 k^{(s_\ell)}/L)$$

$$\mathcal{M}_\ell \leftarrow \tilde{\mathcal{M}}_\ell = \tilde{s}_\ell \tilde{r}_\ell^{\frac{1}{p}} \quad \text{with}$$

$$\tilde{r}_\ell := \exp(\Delta_1 k^{(\mathcal{M}_\ell)})$$

$$p_\ell \leftarrow \tilde{p}_\ell := \Delta_2 k^{(p_\ell)}$$

$$\delta_{k^{(b)}, k^{(s_\ell)}, k^{(\mathcal{M}_\ell)}, k^{(p_\ell)}} = \delta \Delta_2^L \frac{1}{3^{4L+1}} 2^{-|k^{(b)}| - \sum_\ell |k^{(b_\ell)}| - \sum_\ell |k^{(s_\ell)}| - \sum_\ell |k^{(\mathcal{M}_\ell)}|}. \tag{E.85}$$

Using Proposition E.5, this implies that with probability greater than $1 - \delta_{k^{(b)}, k^{(b_\ell)}, k^{(s_\ell)}, k^{(\mathcal{M}_\ell)}, k^{(p_\ell)}}$,

$$\mathbb{E}\left[\mathrm{l}(F_A(x), y)\right] - \frac{1}{N} \sum_{i=1}^N \mathrm{l}(F_A(x_i), y_i) \leq 6\,\mathcal{B} \sqrt{\frac{\log\left(\frac{4}{\delta_{k^{(b)}, k^{(b_\ell)}, k^{(s_\ell)}, k^{(\mathcal{M}_\ell)}, k^{(p_\ell)}}}\right)}{N}}$$

$$+ \frac{416\sqrt{\log_2(4\underline{\Gamma}_{k^{(b)}, k^{(s_\ell)}})}\log(N)[\mathcal{B}+1]}{\sqrt{N}}[L(1+\mathrm{L}_\mathrm{l})]^{\frac{p}{2+p}} \mathcal{R}_{k^{(p_\ell)}, k^{(\mathcal{M}_\ell)}, k^{(b)}, k^{(s_\ell)}}, \tag{E.86}$$

where $\mathcal{R}_{k^{(b)}, k^{(b_\ell)}, k^{(s_\ell)}, k^{(\mathcal{M}_\ell)}, k^{(p_\ell)}}$ is defined as the value of $\mathcal{R}_{\mathcal{M}, s, p, b_\ell}$ for $b, b_\ell, s_\ell, \mathcal{M}_\ell, p_\ell$ taking the values defined in equations (E.84) and $\underline{\Gamma}_{k^{(b)}, k^{(s_\ell)}} = \left[[\exp(k^{(b)}\Delta) + 1][\mathrm{L}_\mathrm{l}+1]\overline{W}N \prod_{\ell=1}^L [\rho_\ell \exp(k^{(s_\ell)}\Delta) + 1]\right]$.

Note that

$$\sum_{k^{(b)}, k^{(b_\ell)}, k^{(s_\ell)}, k^{(\mathcal{M}_\ell)}, k^{(p_\ell)}} \delta_{k^{(b)}, k^{(b_\ell)}, k^{(s_\ell)}, k^{(\mathcal{M}_\ell)}, k^{(p_\ell)}} = \delta \Delta_2^L \frac{1}{3^{4L+1}}\left[\frac{2}{\Delta_2}\right]^L [\sum_{i \in \mathbb{Z}} \frac{1}{2^i}]^{3L+1} \leq \delta. \tag{E.87}$$

Thus, equation (E.86) holds simultaneously over all admissible values of $k^{(b)}, k^{(b_\ell)}, k^{(s_\ell)}, k^{(\mathcal{M}_\ell)}, k^{(p_\ell)}$ with probability $\geq 1 - \delta$.

For a given neural network $F_A$ and choice of $p_\ell$s, we utilise the result above for the following values of $k^{(b)}, k^{(s_\ell)}, k^{(\mathcal{M}_\ell)}, k^{(p_\ell)}$:

$$k^{(b)} := \lceil \log(b)/\Delta_1 \rceil \tag{E.88}$$

$$k^{(b_\ell)} = \lceil \log(\mathfrak{B}_{\ell,A})/\Delta_1 \rceil \tag{E.89}$$

$$k^{(s_\ell)} = \lceil \log(s_\ell)L/\Delta_1 \rceil$$

$$k^{(\mathcal{M}_\ell)} = \lceil \log(\mathrm{r}_\ell)/\Delta_1 \rceil$$

$$k^{(p_\ell)} := \left\lceil \frac{p_\ell}{\Delta_2} \right\rceil,$$

where $b \leftarrow \max_i \|x_i\|$, $b_\ell \leftarrow \mathfrak{B}_{\ell,A}$, $s_\ell \leftarrow \|A_\ell\|$, $\mathrm{r}_\ell \leftarrow \|A_\ell - M_\ell\|_{\mathrm{sc}, p_\ell}^{p_\ell}/\|A\|^{p_\ell} = \sum_i \sigma_i(A - M_\ell)^{p_\ell}$ (for $\ell \neq L$) and $\mathrm{r}_\mathrm{L} \leftarrow \|A_L - M_L\|_{\mathrm{sc}, p_L}^{p_L}/\|A_L^\top\|_{2,\infty}^{p_L} \leq \sqrt{w_L} \sum_i \sigma_i(A_L - M_L)^{p_L})$.

As in the case of Theorem E.2, the following equation still holds as it doesn't involve $k^{(b_\ell)}$ (cf. Eq. (E.17))

$$\underline{\Gamma}_{k^{(b)}, k^{(s_\ell)}} \leq \left[[\exp(\Delta_1)b + 1][\mathrm{L}_\mathrm{l}+1]\overline{W}N \prod_{i=1}^L [\rho_i s_i \exp(\Delta_1/L) + 1]\right] \tag{E.90}$$

$$\leq \exp(2\Delta_1)\left[[b+1][L\,\mathrm{L}_\mathrm{l}+1]\overline{W}N \prod_{i=1}^L [[\rho_i + 1]s_i + 1]\right] = \exp(2\Delta_1)\underline{\Gamma}, \tag{E.91}$$

because $\underline{\Gamma} := \left[[b+1][L\,\mathrm{L}_1+1]\overline{W}N\prod_{i=1}^{L}[[\rho_i+1]s_i+1]\right]$. For the same reason, we still have (cf. equations (E.19), (E.20) and (E.21)):

$$\tilde{r}_\ell \leqslant \mathrm{r}_\ell\, e^{\Delta_1}, \tag{E.92}$$

$$\frac{2\tilde{p}_\ell}{\tilde{p}_\ell+2} - \frac{2p_\ell}{p_\ell+2} \leqslant \frac{2[\tilde{p}_\ell - p_\ell]}{\tilde{p}_\ell+2} \leqslant 2[\tilde{p}_\ell - p_\ell]/2 \leqslant \Delta_2 \tag{E.93}$$

and

$$\frac{2}{\tilde{p}_\ell+2} - \frac{2}{p_\ell+2} = \frac{2[\tilde{p}_\ell - p_\ell]}{[\tilde{p}_\ell+2][p_\ell+2]} \leqslant \frac{2\Delta_2}{4} \leqslant \Delta_2/2. \tag{E.94}$$

Thus, plugging in the new values of $k^{(b)}, k^{(b_\ell)}, k^{(s_\ell)}, k^{(\mathcal{M}_\ell)}, k^{(p_\ell)}$ into the definition of $\mathcal{R}_{\mathcal{M},s,p,b_\ell} = \left[\sum_{\ell=1}^{L}\left[b_{\ell-1}\prod_{i=\ell}\rho_i s_i\right]^{\frac{2p_\ell}{p_\ell+2}}\mathrm{r}_\ell^{\frac{2}{p_\ell+2}}\bar{w}_\ell^{\frac{2}{p_\ell+2}}\tilde{w}_\ell^{\frac{p_\ell}{p_\ell+2}}\right]^{\frac{1}{2}}$ we now obtain:

$$\mathcal{R}_{k^{(b)},k^{(b_\ell)},k^{(s_\ell)},k^{(\mathcal{M}_\ell)},k^{(p_\ell)}} \tag{E.95}$$

$$\leqslant \left[\sum_{\ell=1}^{L}\left[\exp(\Delta_1)b_{\ell-1}\prod_{i=\ell}^{L}\rho_i s_i\exp(\Delta_1/L)\right]^{\frac{2\tilde{p}_\ell}{\tilde{p}_\ell+2}}\tilde{r}_\ell^{\frac{2}{\tilde{p}_\ell+2}}\bar{w}_\ell^{\frac{2}{\tilde{p}_\ell+2}}\tilde{w}_\ell^{\frac{\tilde{p}_\ell}{\tilde{p}_\ell+2}}\right]^{\frac{1}{2}} \tag{E.96}$$

$$\leqslant \left[\sum_{\ell=1}^{L}\left[[b_{\ell-1}\exp(\Delta_1)]\prod_{i=\ell}^{L}\rho_i[s_i\exp(\Delta_1/L)]\right]^{\frac{2\tilde{p}_\ell}{\tilde{p}_\ell+2}}\tilde{r}_\ell^{\frac{2}{\tilde{p}_\ell+2}}\bar{w}_\ell^{\frac{2}{\tilde{p}_\ell+2}}\tilde{w}_\ell^{\frac{\tilde{p}_\ell}{\tilde{p}_\ell+2}}\right]^{\frac{1}{2}} \tag{E.97}$$

$$\leqslant \exp(\Delta_1)\left[\sum_{\ell=1}^{L}\left[b_{\ell-1}\prod_{i=\ell}^{L}\rho_i s_i\right]^{\frac{2\tilde{p}_\ell}{\tilde{p}_\ell+2}}\tilde{r}_\ell^{\frac{2}{\tilde{p}_\ell+2}}\bar{w}_\ell^{\frac{2}{\tilde{p}_\ell+2}}\tilde{w}_\ell^{\frac{\tilde{p}_\ell}{\tilde{p}_\ell+2}}\right]^{\frac{1}{2}} \tag{E.98}$$

$$\leqslant [\exp(\Delta_1)]\left[\sum_{\ell=1}^{L}\left[b_{\ell-1}\prod_{i=\ell}^{L}\rho_i s_i\right]^{\frac{\Delta_2}{2}}\left[b_{\ell-1}\prod_{i=\ell}^{L}\rho_i s_i\right]^{\frac{2p_\ell}{p_\ell+2}}\tilde{r}_\ell^{\frac{2}{\tilde{p}_\ell+2}}\bar{w}_\ell^{\frac{2}{\tilde{p}_\ell+2}}\tilde{w}_\ell^{\frac{\tilde{p}_\ell}{\tilde{p}_\ell+2}}\right]^{\frac{1}{2}} \tag{E.99}$$

$$\leqslant \left[\exp(\Delta_1)\left[b\prod_i\rho_i s_i\right]^{\frac{\Delta_2}{2}}\right]\left[\sum_{\ell=1}^{L}\left[b_{\ell-1}\prod_{i=\ell}^{L}\rho_i s_i\right]^{\frac{2p_\ell}{p_\ell+2}}\tilde{r}_\ell^{\frac{2}{\tilde{p}_\ell+2}}\bar{w}_\ell^{\frac{2}{\tilde{p}_\ell+2}}\tilde{w}_\ell^{\frac{\tilde{p}_\ell}{\tilde{p}_\ell+2}}\right]^{\frac{1}{2}} \tag{E.100}$$

$$\leqslant \left[e^{\Delta_1}\left[b\prod_i\rho_i s_i\right]^{\frac{\Delta_2}{2}}e^{\Delta_1/2}\right]\left[\sum_{\ell=1}^{L}\left[b_{\ell-1}\prod_{i=\ell}^{L}\rho_i s_i\right]^{\frac{2p_\ell}{p_\ell+2}}[\mathrm{r}_\ell\,\bar{w}_\ell]^{\frac{2}{\tilde{p}_\ell+2}}\tilde{w}_\ell^{\frac{\tilde{p}_\ell}{\tilde{p}_\ell+2}}\right]^{\frac{1}{2}} \tag{E.101}$$

$$\leqslant \left[e^{1.5\Delta_1}\left[b\prod_i\rho_i s_i\right]^{\frac{\Delta_2}{2}}\right]\left[\sum_{\ell=1}^{L}\left[b_{\ell-1}\prod_{i=\ell}^{L}\rho_i s_i\right]^{\frac{2p_\ell}{p_\ell+2}}\mathrm{r}_\ell^{\frac{2}{\tilde{p}_\ell+2}}\bar{w}_\ell^{\frac{2}{\tilde{p}_\ell+2}}\tilde{w}_\ell^{\frac{p_\ell}{p_\ell+2}}\right]^{\frac{1}{2}} \tag{E.102}$$

$$\leqslant \left[e^{1.5\Delta_1}\left[b\prod_i\rho_i s_i\right]^{\frac{\Delta_2}{2}}\bar{w}_\ell^{\frac{\Delta_2}{2}}\right]\left[\sum_{\ell=1}^{L}\left[b_{\ell-1}\prod_{i=\ell}^{L}\rho_i s_i\right]^{\frac{2p_\ell}{p_\ell+2}}[\mathrm{r}_\ell\,\bar{w}_\ell]^{\frac{2}{p_\ell+2}}\tilde{w}_\ell^{\frac{p_\ell}{p_\ell+2}}\right]^{\frac{1}{2}} \tag{E.103}$$

$$\leqslant \left[e^{1.5\Delta_1}\left[b\prod_i\rho_i s_i\right]^{\frac{\Delta_2}{2}}\bar{w}_\ell^{\frac{\Delta_2}{2}}\right]\widetilde{\mathcal{R}}_{F_A,\lambda_{aug}} \tag{E.104}$$

$$\leqslant \left[4\overline{W}+b\prod_i\rho_i s_i\right]^{\max(\Delta_1,\Delta_2)}\widetilde{\mathcal{R}}_{F_A,\lambda_{aug}}, \tag{E.105}$$

where at equation (E.99) we have used equation (E.93), at equation (E.100) we have used the fact that $b_\ell \leqslant b\prod_{i=1}^{\ell}\rho_i s_i$ (differently from the proof of Theorem E.2). The rest of the calculation is analogous

to the proof of Theorem E.2: at equation (E.101) we have used equation (E.92), at equation (E.102) we have used the trivial fact that $\frac{\tilde{p}_\ell}{\tilde{p}_\ell+2} \geqslant \frac{p_\ell}{2+p_\ell}$ and at equation (E.103) we have used equation (E.94) and the fact that $r_\ell \leqslant \bar{w}_\ell$.

Thus, since the multiplicative error term is unchanged, we can proceed, as in the proof of Theorem E.2, to set

$$\Delta_1 = \Delta_2 = \frac{\log(2)}{\log\left[4\overline{W} + b\prod_i \rho_i s_i\right]}, \tag{E.106}$$

ensuring

$$\mathcal{R}_{k^{(p_\ell)},k^{(\mathcal{M}_\ell)},k^{(b)},k^{(s_\ell)}} \leqslant 2\widetilde{\mathcal{R}}_{F_A}. \tag{E.107}$$

Similarly, as before, we can write first (from equation (E.90))

$$\underline{\Gamma}_{k^{(b)},k^{(s_\ell)}} \leqslant \underline{\Gamma} 4\left[4\overline{W} + b\prod_i \rho_i\|A_\ell\|\right] \tag{E.108}$$

and then:

$$\log\left(\frac{1}{\delta_{k^{(b)},k^{(s_\ell)},k^{(\mathcal{M}_\ell)},k^{(p_\ell)}}}\right) \tag{E.109}$$

$$\leqslant \log(\frac{1}{\delta}) + [4L+1]\log(3) + \left[|k^{(b)}| + \sum_\ell |k^{(s_\ell)}| + \sum_\ell |k^{(b_\ell)}| + \sum_\ell |k^{(\mathcal{M}_\ell)}|\right] + L\log(1/\Delta_2) \tag{E.110}$$

$$\leqslant \log(\frac{1}{\delta}) + L\left[5.5 + \log\log\left[4\overline{W} + b\prod_i \rho_i s_i\right]\right] \tag{E.111}$$

$$+ \left[|k^{(b)}| + \sum_\ell |k^{(s_\ell)}| + \sum_\ell |k^{(b_\ell)}| + \sum_\ell |k^{(\mathcal{M}_\ell)}|\right] \tag{E.112}$$

$$\leqslant \log(\frac{1}{\delta}) + L\left[5.5 + \log\log\left[4\overline{W} + b\prod_i \rho_i\|A_\ell\|\right]\right] + 1.5 \tag{E.113}$$

$$+ \left[|\log(b)| + L\left[\sum_{\ell=1}^{L}|\log(\|A_\ell\|)| + \sum_{\ell=1}^{L-1}\log(\mathfrak{B}_{\ell,A}) + \log(\overline{W})\right]\right]/\Delta_1 \tag{E.114}$$

$$\leqslant \log(\frac{1}{\delta}) + L\left[5.5 + \log\left[4\overline{W} + b\prod_i \rho_i\|A_\ell\|\right]\right] + 1.5 \tag{E.115}$$

$$+ 2\log\left[4\overline{W} + b\prod_i \rho_i\|A_\ell\|\right]\left[|\log(b)| + L\left[\sum_{\ell=1}^{L}|\log(\|A_\ell\|)| + \sum_{\ell=1}^{L-1}\log(\mathfrak{B}_{\ell,A}) + \log(\overline{W})\right]\right]$$

$$\leqslant \log(\frac{1}{\delta}) + 1.5 + \tag{E.116}$$

$$2\log\left[4\overline{W} + b\prod_i \rho_i\|A_\ell\|\right]\left[|\log(b)| + L\left[\sum_{\ell=1}^{L}|\log(\|A_\ell\|)| + \sum_{\ell=1}^{L-1}\log(\mathfrak{B}_{\ell,A}) + 2\log(4\overline{W})\right]\right]$$

$$= \log(\frac{1}{\delta}) + 1.5 + \Theta_{\log}^{\lambda_{aug}}, \tag{E.117}$$

where at equation (E.111) we have used the definition of $\Delta_2$ (i.e. equation (E.106)) and the fact that $\log(\log(2)) \geqslant -1$, at equation (E.114) we have used equations (E.88) , and at equation (E.115) we

have used the fact that $\log(2) \geqslant \frac{1}{2}$ and the inequality $\log\log(x) \leqslant \log(x)$, at equation (E.116) we have used the fact that $4\log(4) \geqslant 5.5$, and at equation (E.117) we have used the definition of $\Theta_{\log}^{\lambda_{aug}}$.

We now plug equations (E.105), (E.107), (E.117) and (E.108) into equation (E.86) to obtain:

$$\mathbb{E}\left[\mathrm{l}(F_A(x), y)\right] - \frac{1}{N}\sum_{i=1}^{N}\mathrm{l}(F_A(x_i), y_i) \leqslant 6[\mathcal{B}+1]\sqrt{\frac{\log\left(\frac{4}{\delta_{k(b)},k^{(s_\ell)},k^{(\mathcal{M}_\ell)},k^{(p_\ell)}}\right)}{N}}$$

$$+ \frac{832\sqrt{\log_2(2\underline{\Gamma}_{k^{(b)},k^{(s_\ell)}})}\log(N)[\mathcal{B}+1]}{\sqrt{N}}[L(1+\mathrm{L_l})]^{\frac{p}{2+p}}\mathcal{R}_{k^{(b)},k^{(b_\ell)},k^{(s_\ell)},k^{(\mathcal{M}_\ell)},k^{(p_\ell)}} \tag{E.118}$$

$$\leqslant 6[\mathcal{B}+1]\sqrt{\frac{\log(1/\delta)}{N}} + 6[\mathcal{B}+1]\sqrt{\frac{1.5 + \Theta_{\log}^{\lambda_{aug}}}{N}} \tag{E.119}$$

$$+ 832\,\mathcal{B}\,L[(1+\mathrm{L_l})]^{\frac{p}{2+p}}\frac{\sqrt{\log_2(8\underline{\Gamma}\left[4\overline{W} + b\prod_i\rho_i\|A_\ell\|\right])}\log(N)}{N}\widetilde{\mathcal{R}}_{F_A,\lambda_{aug}} \tag{E.120}$$

$$\leqslant 6[\mathcal{B}+1]\sqrt{\frac{\log(1/\delta)}{N}} + 6[\mathcal{B}+1]\sqrt{\frac{1.5 + \Theta_{\log}}{N}} + 832\,\mathcal{B}[L[1+\mathrm{L_l}]]^{\frac{p}{2+p}} \times \tag{E.121}$$

$$\frac{\sqrt{\log\left(12\left[[b+1][L\,\mathrm{L_l}+1]\overline{W}N\prod_{i=1}^{L}[[\rho_i+1]s_i+1]\right]\left[4\overline{W}+b\prod_i\rho_i\|A_\ell\|\right]\right)\log(N)}}{N}\widetilde{\mathcal{R}}_{F_A,\lambda_{aug}} \tag{E.122}$$

$$\leqslant 6[\mathcal{B}+1]\sqrt{\frac{\log(1/\delta)}{N}} + 6[\mathcal{B}+1]\sqrt{\frac{1.5 + \Theta_{\log}}{N}} \tag{E.123}$$

$$+ 832\,\mathcal{B}[L(1+\mathrm{L_l})]^{\frac{p}{2+p}}\frac{\sqrt{\log(\gamma_{FC})}\log(N)}{N}\widetilde{\mathcal{R}}_{F_A,\lambda_{aug}}, \tag{E.124}$$

where at line (E.120) we have used equation (E.107), and at the last line we have used the definition

$$\gamma_{FC} := 12\left[[b+1][L\,\mathrm{L_l}+1]\overline{W}N\prod_{i=1}^{L}[\rho_i\|A_i\|+1]\right]\left[4\overline{W}+b\prod_i\rho_i\|A_i\|\right].$$

This concludes the proof.

$\square$

## E.2 The Case $p_\ell = 0$ for all $\ell$

We note that both of the above results hold for all values of the $p_\ell$s, including $0$. However, the fact that Theorem E.2 is posthoc with respect to all combinations of values of $p_\ell$ introduces an additional term of $\sqrt{\frac{L^3}{N}}$. In this subsection, we show that this can be avoided with a more dedicated proof for the case $p_\ell = 0$.

We first note the following immediate corollary of Proposition E.1:

**Proposition E.7.** *Instate the assumptions of Proposition E.1, and assume additionally that $p_\ell = 0$ for all $\ell$. With probability at least $1 - \delta$ over the draw of the training set,*

$$\mathbb{E}\left[\mathrm{l}(F_A(x), y)\right] - \frac{1}{N}\sum_{i=1}^{N}\mathrm{l}(F_A(x_i), y_i)$$

$$\leqslant 6[\mathcal{B}+1]\sqrt{\frac{\log(\frac{4}{\delta})}{N}} + \frac{240\sqrt{\log_2(4\underline{\Gamma})}\log(N)[\mathcal{B}+1]}{\sqrt{N}}\left[\sum_{\ell=1}^{L}[w_\ell + w_{\ell-1}]\tilde{r}_\ell\right], \tag{E.125}$$

*where $\tilde{r}_\ell$ is an a priori upper bound on $\mathrm{rank}(A_\ell)$.*

Next, we consider the following post hoc version.

**Theorem E.8.** *W.p.* $\geqslant 1 - \delta$ *over the draw of the training set, every neural network satisfies:*

$$\mathbb{E}\left[l(F_A(x), y)\right] - \frac{1}{N} \sum_{i=1}^{N} l(F_A(x_i), y_i)$$

$$\leqslant 6[\mathcal{B}+1]\sqrt{\frac{\log(\frac{4}{\delta})}{N}} + \frac{240\sqrt{\log_2(4\Gamma_R)}\log(N)[\mathcal{B}+1]}{\sqrt{N}} \left[\sum_{\ell=1}^{L}[w_\ell + w_{\ell-1}]rank(A_\ell)\right]$$

$$+ \sqrt{\frac{\log(4/\delta) + 2\log\left[[\mathcal{B}+2]\prod_{\ell=1}^{L}(\|A_\ell\|+2)\right] + L\log(\overline{W})}{N}},$$

*where* $\Gamma_R = [\mathcal{B}+2][L\,\mathrm{L}_1+1]\overline{W}N\prod_{i=1}^{L}[[\rho_i+1][\|A_i\|+2]+1].$

*Proof.* We apply Proposition E.7 simultaneously for all the values of $b, s_\ell \in \mathbb{N}_+$ and $\tilde{r}_\ell \in [\overline{W}]$ after setting

$$\delta_{b,s_\ell,\tilde{r}_\ell} = \frac{\delta}{[b+1]^2 \prod_\ell [s_\ell+1]^2 \overline{W}^L}. \tag{E.126}$$

Note that

$$\sum_{b\in\mathbb{N}_+, s_\ell\in\mathbb{N}_+, \tilde{r}_\ell\leqslant\overline{W}} \delta_{b,s_\ell,\tilde{r}_\ell} \leqslant \frac{\delta\overline{W}^L[\pi^2/6-1]^{1+L}}{\overline{W}^L} \leqslant \delta. \tag{E.127}$$

Thus, inequality (E.125) holds simultaneously over all values of $b, s_\ell \in \mathbb{N}_+$ and $\tilde{r}_\ell \in [\overline{W}]$ with failure probability $\delta$ with the loss function l replaced by $l_{aug}$ as in the proof of Theorem E.2.

We apply this to $s_\ell = \lceil\|A_\ell\|\rceil$, $b = \lceil\mathcal{B}\rceil$, $\tilde{r}_\ell = \mathrm{rank}(A^\ell)$. This immediately yields

$$\mathbb{E}\left[l(F_A(x), y)\right] - \frac{1}{N} \sum_{i=1}^{N} l(F_A(x_i), y_i)$$

$$\leqslant 6[\mathcal{B}+1]\sqrt{\frac{\log(\frac{4}{\delta_{b,s_\ell,\tilde{r}_\ell}})}{N}} + \frac{240\sqrt{\log_2(4\Gamma_R)}\log(N)[\mathcal{B}+1]}{\sqrt{N}} \left[\sum_{\ell=1}^{L}[w_\ell + w_{\ell-1}]rank(A_\ell)\right], \tag{E.128}$$

where

$$\Gamma_R \leqslant \left[[b+1][L\,\mathrm{L}_1+1]\overline{W}N\prod_{i=1}^{L}[[\rho_i+1]s_i+1]\right]$$

$$\leqslant [\mathcal{B}+2][L\,\mathrm{L}_1+1]\overline{W}N\prod_{i=1}^{L}[[\rho_i+1][\|A_i\|+2]+1].$$

Next, we note that

$$\log(\frac{4}{\delta_{b,s_\ell,\tilde{r}_\ell}}) \leqslant \log(4/\delta) + 2\log\left[[\mathcal{B}+2]\prod_{\ell=1}^{L}(\|A_\ell\|+2)\right] + L\log(\overline{W}). \tag{E.129}$$

Plugging this back into equation (E.128) yields the result as expected.

$\square$

### E.3 Covering Number Bound for Fully Connected Networks

In this subsection, we use the results of Section F to construct a covering number bound for the class of fully connected neural networks. First, we briefly recall the following definitions.

**Definition E.9** (Covering number). Let $V \subset \mathbb{R}^n$ and $\|\cdot\|$ be a norm in $\mathbb{R}^n$. The covering number w.r.t. $\|\cdot\|$, denoted by $\mathcal{N}(V, \epsilon, \|\cdot\|)$, is the minimum cardinality $m$ of a collection of vectors $\mathbf{v}^1, \ldots, \mathbf{v}^m \in \mathbb{R}^n$ such that $\sup_{\mathbf{v} \in V} \min_{j=1,\ldots,m} \|\mathbf{v} - \mathbf{v}^j\| \leqslant \epsilon$. In particular, if $\mathcal{F} \subset \mathbb{R}^{\mathcal{X}}$ is a function class and $X = (x_1, x_2, \ldots, x_n) \in \mathcal{X}^n$ are data points, $\mathcal{N}(\mathcal{F}(X), \epsilon, (1/\sqrt{n})\|\cdot\|_2)$ is the minimum cardinality $m$ of a collection of functions $\mathcal{F} \ni f^1, \ldots, f^m : \mathcal{X} \to \mathbb{R}$ such that for any $f \in \mathcal{F}$, there exists $j \leqslant m$ such that $\sum_{i=1}^{n}(1/n)\left|f^j(x_i) - f(x_i)\right|^2 \leqslant \epsilon^2$. Similarly, $\mathcal{N}(\mathcal{F}(X), \epsilon, \|\cdot\|_\infty)$ is the minimum cardinality $m$ of a collection of functions $\mathcal{F} \ni f^1, \ldots, f^m : \mathcal{X} \to \mathbb{R}$ such that for any $f \in \mathcal{F}$, there exists $j \leqslant m$ such that $i \leqslant n$, $\left|f^j(x_i) - f(x_i)\right| \leqslant \epsilon$.

**Definition E.10.** Let $\mathcal{F}$ be a class of real-valued functions with range $X$. Let also $S = (x_1, x_2, \ldots, x_n) \in X$ be $n$ samples from the domain of the functions in $\mathcal{F}$. The empirical Rademacher complexity $\mathfrak{R}_S(\mathcal{F})$ of $\mathcal{F}$ with respect to $x_1, x_2, \ldots, x_n$ is defined by

$$\mathfrak{R}_S(\mathcal{F}) := \mathbb{E}_\delta \sup_{f \in \mathcal{F}} \frac{1}{n} \sum_{i=1}^{n} \delta_i f(x_i), \tag{E.130}$$

where $\delta = (\delta_1, \delta_2, \ldots, \delta_n) \in \{\pm 1\}^n$ is a set of $n$ iid Rademacher random variables (which take values $1$ or $-1$ with probability $0.5$ each).

We now have the following result on the covering number of the class $\mathcal{F}_\mathcal{B}$.

**Proposition E.11.** let $x_1, \ldots, x_N \in \mathbb{R}^d$ be an arbitrary set of inputs satisfying $\|x_i\| \leqslant b$ for all $i$. We have the following bound on the $l^\infty$ covering number of the class $\mathcal{F}_\mathcal{B}$ for any granularity $\frac{1}{N} \leqslant \varepsilon$:

$$\log\left(\mathcal{N}_\infty(1 \circ \mathcal{F}_\mathcal{B}, \epsilon)\right)$$

$$\leqslant 24 \log_2(\bar{\Gamma}) \left[\frac{(1 + L\,\mathrm{L_l})}{\min(\varepsilon, 1)}\right]^{\frac{2p}{2+p}} \sum_{\ell=1}^{L} \left[b \prod_i \rho_i s_i\right]^{\frac{2p_\ell}{p_\ell+2}} \left[\frac{\mathcal{M}_\ell^{p_\ell}}{s_\ell^{p_\ell}}\right]^{\frac{2}{p_\ell+2}} \bar{w}_\ell^{\frac{2}{p_\ell+2}} \tilde{w}_\ell^{\frac{p_\ell}{p_\ell+2}}$$

$$\leqslant 96 \log_2(4\underline{\Gamma}) \left[\frac{(1 + L\,\mathrm{L_l})}{\min(\varepsilon, 1)}\right]^{\frac{2p}{2+p}} \sum_{\ell=1}^{L} \left[b \prod_i \rho_i s_i\right]^{\frac{2p_\ell}{p_\ell+2}} \mathrm{r}_\ell^{\frac{2}{p_\ell+2}} \bar{w}_\ell^{\frac{2}{p_\ell+2}} \tilde{w}_\ell^{\frac{p_\ell}{p_\ell+2}}, \tag{E.131}$$

where $p = \max_\ell(p_\ell)$, $\underline{w}_\ell = \min(w_\ell, w_{\ell-1})$, $\bar{w}_\ell = [w_\ell + w_{\ell-1}]$, $\mathrm{r}_\ell = \frac{\mathcal{M}_\ell^p}{s_\ell^p}$ and $\tilde{w}_\ell := [w_\ell \underline{w}_\ell]$ if $\ell \neq L$ $\underline{w}_L$ if $\ell = L$. Here we also have $\bar{\Gamma} = 128\underline{\Gamma}^4$ and $\underline{\Gamma} = \left[[b+1][L\,\mathrm{L_l}+1]\overline{W} N \prod_{i=1}^{L}[[\rho_i + 1]s_i + 1]\right]$ where $\overline{W} := \max_{\ell=0,1,\ldots,L} w_\ell$.

Before proving Proposition E.11, we will need the following Lemma, which we will later apply to the following choice: $\mathcal{B}_\ell := \{A_\ell \in \mathbb{R}^{w_\ell \times w_{\ell-1}} : \|A_\ell - M_\ell\|_{\mathrm{sc}, p_\ell} \leqslant \mathcal{M}_\ell; \|A_\ell\| \leqslant s_\ell\}$.

**Lemma E.12.** Let $\epsilon_1, \ldots, \epsilon_L > 0$ be some arbitrary positive real numbers, and assume we have a fixed dataset $x_1, \ldots, x_N$ with $\|x_i\| \leqslant b$ for all $i \leqslant N$. Let $\mathcal{B}_1, \ldots, \mathcal{B}_L$ be arbitrary subsets of the spaces $\mathbb{R}^{w_\ell \times w_{\ell-1}}$.

There exist covers $\mathcal{C}_1 \subset \mathcal{B}_1, \ldots, \mathcal{C}_{0 \to \ell} \subset \mathcal{B}_1 \times \ldots \mathcal{B}_\ell, \ldots, \mathcal{C}_L \subset \mathcal{B}_1 \times \ldots \mathcal{B}_L$ such that for all $(A_1, \ldots, A_L) \in \mathcal{B}_1 \times \ldots \mathcal{B}_L$, there exist $\bar{A}^1, \ldots, \bar{A}^L$ such that for all $\ell \leqslant L$, $\bar{A}^{0 \to \ell} := (\bar{A}^1, \ldots, \bar{A}^\ell) \in \mathcal{C}_{0 \to \ell}$ and for all $i \leqslant N$,

$$\|F_{A^1, \ldots, A^\ell}^{0 \to \ell}(x_i) - F_{\bar{A}^1, \ldots, \bar{A}^\ell}^{0 \to \ell}(x_i)\|_2 \leqslant \sum_{l=1}^{\ell} \epsilon_l \rho_{l \to \ell} \qquad \text{if} \quad \ell \neq L$$

$$\|F_{A^1, \ldots, A^\ell}^{0 \to L}(x_i) - F_{\bar{A}^1, \ldots, \bar{A}^\ell}^{0 \to L}(x_i)\|_\infty \leqslant \sum_{l=1}^{L} \epsilon_l \rho_{l \to L}. \tag{E.132}$$

Here $\rho_{\ell_1 \to \ell_2} := \rho_{\ell_1} \prod_{\ell=\ell_1+1}^{\ell_2} \rho_\ell s_\ell$ where $\rho_0 = 1$ by convention.

*Furthermore the covers satisfy the following covering number bound:*

$$\log(|\mathcal{C}_{0\to L}|) \leq \sum_{\ell=1}^{L} \log(\mathcal{N}_{\infty,\ell}(\mathcal{B}_\ell, \epsilon_\ell, N, b\rho_{0\to\ell-1})), \tag{E.133}$$

*where for any $\bar{b} > 0$ and $\bar{N} \in \mathbb{N}$, $\mathcal{N}_{\infty,\ell}(\mathcal{B}_\ell, \epsilon_\ell, N, \bar{b})$ is defined as the minimum number $\mathcal{N}$ such that for any dataset $x_1^{\ell-1}, \ldots, x_{\bar{N}}^{\ell-1}$ with $\bar{N} \leq N$ and $\|x_i^{\ell-1}\|_\ell \leq \bar{b}$ for all $i \leq \bar{N}$, where $\|\cdot\|_\ell = \|\cdot\|$ for $\ell \neq L$ and $\|\cdot\|_L = \|\cdot\|_\infty$.*

*Proof.* The proof consists in stringing together the covering numbers of each individual layer with arguments analogous to those in the comparable literature on generalization bounds for neural networks [1, 16, 74].

For the **first layer**, writing $X := \{x_1, \ldots, x_N\}$, we know by assumption that there exists a cover $\mathcal{C}_1(X) = \mathcal{C}_{0\to1} \subset \mathcal{B}_1$ such that for all $A_1 \in \mathcal{B}_1$, there exists a $\bar{A}_1 \in \mathcal{C}_1(X_0)$ such that for all $i \leq N$,

$$\|(A^1 - \bar{A}^1)(x_i)\| \leq \epsilon_1, \tag{E.134}$$

from which it also follows that

$$\|F_{A^1}^{0\to1}(x_1) - F_{\bar{A}^1}^{0\to1}(x_1)\| = \|\sigma_1(A^1 x_i) - \sigma_1(\bar{A}^1(x_i))\| \leq \epsilon_1 \rho_1. \tag{E.135}$$

It follows that $\{F_{A^1}^{0\to1} : A^1 \in \mathcal{B}_1\} \subset \mathcal{F}_{\mathcal{B}}^{0\to1}$ is an $L^{\infty,2}$ cover of $\mathcal{F}_{\mathcal{B}}^{0\to1}$ as required: equation E.132 holds for $\ell = 1$.

For the **inductive case**:

Assume that the covers $\mathcal{C}_{0\to1}, \ldots, \mathcal{C}_{0\to\ell}$ have been constructed and satisfy equation (E.132). For each element $\bar{A}^{0\to\ell}$ of $\mathcal{C}_{0\to\ell}$, we have

$$\|F_{\bar{A}^{0\to\ell}}^{0\to\ell}(x_i)\|_2 \leq \rho_{0\to\ell} b.$$

Thus, we can construct a cover $\mathcal{C}_{\ell+1}(F_{\bar{A}^{0\to\ell}}^{0\to\ell}(X)) \subset \mathcal{B}_{\ell+1}$ such that for all $A^{\ell+1} \in \mathcal{B}_{\ell+1}$ there exists a $\bar{A}^{\ell+1} \in \mathcal{B}_{\ell+1}$ such that for all $i \leq N$, we have

$$\|F_{(\bar{A}^1,\ldots,\bar{A}^\ell,A^{\ell+1})}^{0\to\ell+1}(x_i) - F_{(\bar{A}^1,\ldots,\bar{A}^\ell,\bar{A}^{\ell+1})}^{0\to\ell+1}(x_i)\|_{\ell+1} \leq \epsilon_{\ell+1} \tag{E.136}$$

and

$$\log\left(|\mathcal{C}_{\ell+1}(F_{\bar{A}^{0\to\ell}}^{0\to\ell}(X))|\right) \leq \log(\mathcal{N}_{\infty,\ell+1}(\mathcal{B}_{\ell+1}, \epsilon_{\ell+1}, N, b\rho_{0\to\ell})). \tag{E.137}$$

We now construct the cover

$$\mathcal{C}_{0\to\ell+1} := \bigcup_{\bar{A}^{0\to\ell} \in \mathcal{C}_{0\to\ell}} \mathcal{C}_{\ell+1}(F_{\bar{A}^{0\to\ell}}^{0\to\ell}(X)). \tag{E.138}$$

For any $A^1, \ldots, A^{\ell+1}$, by the induction hypothesis [10], there exists $(\bar{A}^1, \ldots, \bar{A}^\ell) \in \mathcal{C}_{0\to\ell}$ such that for all $i \leq N$,

$$\|F_{A^1,\ldots,A^\ell}^{0\to\ell}(x_i) - F_{\bar{A}^1,\ldots,\bar{A}^\ell}^{0\to\ell}(x_i)\| \leq \sum_{l=1}^{\ell} \epsilon_l \rho_{l\to\ell}.$$

Thus we have

$$\|F_{A^1,\ldots,A^{\ell+1}}^{0\to\ell+1}(x_i) - F_{\bar{A}^1,\ldots,\bar{A}^{\ell+1}}^{0\to\ell+1}(x_i)\|$$

$$\leq \|F_{A^1,\ldots,A^{\ell+1}}^{0\to\ell+1}(x_i) - F_{\bar{A}^1,\ldots,\bar{A}^\ell,A^{\ell+1}}^{0\to\ell+1}(x_i)\| + \|F_{\bar{A}^1,\ldots,\bar{A}^\ell,A^{\ell+1}}^{0\to\ell+1}(x_i) - F_{\bar{A}^1,\ldots,\bar{A}^{\ell+1}}^{0\to\ell+1}(x_i)\|$$

$$\leq \rho_{\ell+1}s_{\ell+1}\|F_{A^1,\ldots,A^\ell}^{0\to\ell}(x_i) - F_{\bar{A}^1,\ldots,\bar{A}^\ell}^{0\to\ell}(x_i)\| + \rho_{\ell+1}\|A^{\ell+1}(F_{\bar{A}^{0\to\ell}}^{0\to\ell}(x_i)) - \bar{A}^{\ell+1}(F_{\bar{A}^{0\to\ell}}^{0\to\ell}(x_i))\|$$

$$\leq \rho_{\ell+1}s_{\ell+1}\sum_{l=1}^{\ell} \epsilon_l \rho_{l\to\ell} + \rho_{\ell+1}\epsilon_{\ell+1} = \sum_{l=1}^{\ell+1} \epsilon_l,$$

---

[10] if $\ell = L$ the theorem is proved, so we can assume $\ell \neq L$ in which case $\|\cdot\|_\ell = \|\cdot\|$

as expected. Furthermore, by construction (E.138), the cardinality of $\mathcal{C}_{0\to\ell+1}$ certainly satisfies:

$$|\mathcal{C}_{0\to\ell+1}| \leqslant |\mathcal{C}_{0\to\ell}| \times \mathcal{N}_{\infty,\ell+1}(\mathcal{B}_{\ell+1}, \epsilon_{\ell+1}, N, b\rho_{0\to\ell+1})$$

$$\leqslant \prod_{l=1}^{\ell} \mathcal{N}_{\infty,l}(\mathcal{B}_{\ell+1}, \epsilon_l, N, b\rho_{0\to l}) \times \mathcal{N}_{\infty,\ell+1}(\mathcal{B}_{\ell+1}, \epsilon_{\ell+1}, N, b\rho_{0\to\ell+1})$$

$$= \prod_{l=1}^{\ell+1} \mathcal{N}_{\infty,l}(\mathcal{B}_{\ell+1}, \epsilon_l, N, b\rho_{0\to l}),$$

where at the last line we have used the induction hypothesis. $\qquad\square$

We can now proceed with the Proof of Proposition E.11.

*Proof of Proposition E.11.* For any choice of $\epsilon_1, \ldots, \epsilon_L$, by Lemma E.12, we know that $l \circ \mathcal{F}_\mathcal{B}$ admits an $L^\infty$ cover $\mathcal{C}$ with granularity $\epsilon = \sum_{\epsilon=1}^{L} \epsilon_\ell \rho_{\ell\to L}$ and cardinality

$$\log(|\mathcal{C}|) \leqslant \sum_{\ell=1}^{L} \log(\mathcal{N}_{\infty,\ell}(\mathcal{B}_\ell, \epsilon_\ell, N, b\rho_{0\to\ell-1})). \tag{E.139}$$

Let us denote $\bar{w}_\ell := [w_\ell + w_{\ell-1}]$, $\underline{w}_\ell = \min(w_\ell, w_{\ell-1})$ and we further define $\tilde{w}_\ell = [w_\ell \underline{w}_\ell]$ if $\ell \neq L$ $\underline{w}_L$ if $\ell = L$.

Further, by Propositions (F.4) (for $\ell \neq L$) and (F.2) (for layer $\ell = L$), we can further continue from equation (E.139) as follows:

$$\log(|\mathcal{C}|) \leqslant \sum_{\ell=1}^{L} \log(\mathcal{N}_{\infty,\ell}(\mathcal{B}_\ell, \epsilon_\ell, N, b\rho_{0\to\ell-1}))$$

$$\leqslant 24 \left[ \sum_{\ell=1}^{L-1} \left[ \frac{\mathcal{M}_\ell\, b\rho_{0\to\ell-1}}{\epsilon_\ell} \right]^{\frac{2p_\ell}{p_\ell+2}} \bar{w}_\ell^{\frac{2}{p_\ell+2}} [w_\ell \underline{w}_\ell]^{\frac{p_\ell}{p_\ell+2}} \log_2(\Gamma_{\mathcal{F}_r^P,\ell}) \right.$$

$$\left. + \left[ \frac{\mathcal{M}_L\, b\rho_{0\to L-1}}{\epsilon_L} \right]^{\frac{2p_L}{p_L+2}} \bar{w}_L^{\frac{2}{p_L+2}} \underline{w}_L^{\frac{p_L}{p_L+2}} \log_2(\Gamma_{\mathcal{F}_r^P,L}) \right]$$

$$\leqslant 24 \log_2(\bar{\Gamma}) \left[ \sum_{\ell=1}^{L-1} \left[ \frac{\mathcal{M}_\ell\, b\rho_{0\to\ell-1}}{\epsilon_\ell} \right]^{\frac{2p_\ell}{p_\ell+2}} \bar{w}_\ell^{\frac{2}{p_\ell+2}} [w_\ell \underline{w}_\ell]^{\frac{p_\ell}{p_\ell+2}} + \left[ \frac{\mathcal{M}_L\, b\rho_{0\to L-1}}{\epsilon_L} \right]^{\frac{2p_L}{p_L+2}} \bar{w}_L^{\frac{2}{p_L+2}} \underline{w}_L^{\frac{p_L}{p_L+2}} \right]$$

$$\leqslant 24 \log_2(\bar{\Gamma}) \sum_{\ell=1}^{L} \left[ \frac{\mathcal{M}_\ell\, b\rho_{0\to\ell-1}}{\epsilon_\ell} \right]^{\frac{2p_\ell}{p_\ell+2}} \bar{w}_\ell^{\frac{2}{p_\ell+2}} \tilde{w}_\ell^{\frac{p_\ell}{p_\ell+2}}, \tag{E.140}$$

where $\Gamma_{\mathcal{F}_r^P,\ell} := \left[ \left( \frac{16[\mathcal{M}_\ell^{p_\ell}+1][b\prod_{i=1}^{\ell}\rho_i s_i+1][1+s_\ell][w_\ell+w_{\ell-1}]^2}{\epsilon_\ell^{\frac{p_\ell}{p_\ell+2}}} + 7 \right) w_\ell N \right]$, $\Gamma_{\mathcal{F}_r^P,L} :=$

$\left[ \left( \frac{16[\mathcal{M}_L^{p_L}+1][b\prod_{i=1}^{L}\rho_i s_i+1][1+\sqrt{w_L}\, s_L][w_L+w_{L-1}]}{\epsilon_L^{\frac{p_L}{p_L+2}}} + 7 \right) w_L N \right]$ (indeed, note that since $\|A_L^\top\|_{2,\infty} \leqslant s_L$, we certainly have $\|A_L\| \leqslant w_L s_L$). Next, we upper bound the logarithmic factors arising from $\Gamma_{\mathcal{F}_r^P,\ell}$ and $\Gamma_{\mathcal{F}_r^P,L}$ as follows:

$$\max_\ell \Gamma_{\mathcal{F}_r^P,\ell} \leqslant \left[ \left( \frac{64[\max_\ell \frac{\mathcal{M}_\ell^{p_\ell}}{s_\ell^{p_\ell}}+1][b+1]\prod_{i=1}^{L}[[\rho_i+1]s_i+1]^3 \overline{W}^{1.5}[L_l L+1]}{\min(1,\varepsilon)} + 7 \right) \overline{W} N \right]$$

$$\leqslant \left[ 128 \prod_{i=1}^{L}[[\rho_i+1]s_i+1]^3 [b+1][L_l L+1]\overline{W}^4 N^2 \right] := \bar{\Gamma} = 128\underline{\Gamma}^4, \tag{E.141}$$

where $\underline{\Gamma} := \left[ [b+1][L\,L_l+1]\overline{W} N \prod_{i=1}^{L}[[\rho_i+1]s_i+1] \right]$ and we have used the fact that $\max_\ell \frac{\mathcal{M}_\ell^{p_\ell}}{s_\ell^{p_\ell}} \leqslant \overline{W}\sqrt{w_L}$.

We now set $\epsilon_\ell := \frac{\alpha_\ell \varepsilon}{\rho_{\ell \to L}} = \frac{\alpha_\ell \varepsilon}{\prod_{i=\ell+1}^{L} s_i \rho_i}$ for some $\alpha_\ell$ such that $\sum_\ell \alpha_\ell = 1/\mathrm{L_l}$. Note that this ensures that the granularity of the cover $\mathcal{C}$ is

$$\epsilon = \mathrm{L_l} \sum_{\ell=1}^{L} \epsilon_\ell \rho_{\ell \to L} = \mathrm{L_l} \sum_{\ell=1}^{L} \frac{\alpha_\ell \varepsilon}{\rho_{\ell \to L}} \rho_{\ell \to L} = \varepsilon. \tag{E.142}$$

Then, we can plug in the value of $\varepsilon$ into equation (E.140) to obtain:

$$\log\left(|\mathcal{C}|\right) \leqslant 24 \log_2(\bar{\Gamma}) \sum_{\ell=1}^{L} \left[\frac{\mathcal{M}_\ell\, b\rho_{0 \to \ell-1}}{\epsilon_\ell}\right]^{\frac{2p_\ell}{p_\ell+2}} \bar{w}_\ell^{\frac{2}{p_\ell+2}} \tilde{w}_\ell^{\frac{p_\ell}{p_\ell+2}}$$

$$\leqslant 24 \log_2(\bar{\Gamma}) \sum_{\ell=1}^{L} \left[\frac{\mathcal{M}_\ell\, b \prod_{i \neq \ell} \rho_i s_i}{\varepsilon \alpha_\ell}\right]^{\frac{2p_\ell}{p_\ell+2}} \bar{w}_\ell^{\frac{2}{p_\ell+2}} \tilde{w}_\ell^{\frac{p_\ell}{p_\ell+2}}$$

$$\leqslant 24\, \mathrm{L_l} \log_2(\bar{\Gamma}) \sum_{\ell=1}^{L} \left[\frac{\mathcal{M}_\ell\, b \prod_i \rho_i s_i}{\varepsilon s_\ell \alpha_\ell}\right]^{\frac{2p_\ell}{p_\ell+2}} \bar{w}_\ell^{\frac{2}{p_\ell+2}} \tilde{w}_\ell^{\frac{p_\ell}{p_\ell+2}}$$

$$\leqslant 24 \log_2(\bar{\Gamma}) \left[\frac{(1 + L\,\mathrm{L_l})}{\min(\varepsilon, 1)}\right]^{\frac{2p}{2+p}} \sum_{\ell=1}^{L} \left[b \prod_i \rho_i s_i\right]^{\frac{2p_\ell}{p_\ell+2}} \left[\frac{\mathcal{M}_\ell^{p_\ell}}{s_\ell^{p_\ell}}\right]^{\frac{2}{p_\ell+2}} \bar{w}_\ell^{\frac{2}{p_\ell+2}} \tilde{w}_\ell^{\frac{p_\ell}{p_\ell+2}}, \tag{E.143}$$

where at the last line, we have set $\alpha_\ell = \frac{1}{\mathrm{L_l}\, L}$. $\qquad\qquad\square$

# F  Covering Numbers for Linear Maps with Schatten Quasi-norm Constraints (One Layer Case)

In this section, we prove covering number bounds for classes of linear maps with bounded Schatten $p$ quasi norms, which corresponds to the one layer case in our analysis. Thus, the results in this section form the basic ingredient used for the proofs in all other sections. To achieve this, we begin with the following parameter counting argument to bound the complexity of low rank matrices. Proposition F.1 is known, but we reproduce the proof for completeness. The other results in this section are original to the best of our knowledge.

**Proposition F.1** (Cf. [28] (Lemma D1 page 30), cf. also [90, 91]). *Let $\mathcal{E}_{\mathrm{r,s}}$ denote the set $\left\{A \in \mathbb{R}^{m \times d} : \mathrm{rank}(A) \leqslant r, \|A\|_\sigma \leqslant s\right\}$. There exists a cover $\mathcal{C} \subset \mathcal{E}_{\mathrm{r,s}}$ with respect to the spectral norm such that*

$$\log\left(|\mathcal{C}|\right) \leqslant [m+d]r \log\left[1 + \frac{6s}{\epsilon}\right] = \tilde{O}\left([m+d]r\right). \tag{F.1}$$

*Proof.* First, note that every matrix $A \in \mathcal{E}_{\mathrm{r,s}}$ can be written as $A = WV^\top$ for $W \in \mathbb{R}^{m \times r}$, $V \in \mathbb{R}^{d \times r}$ with $\|W\| \leqslant \sqrt{s}$ and $\|V\| \leqslant \sqrt{s}$. Then, by Lemma (I.4), there are $\epsilon/2\sqrt{s}$ covers $\mathcal{C}_\mathrm{l}$ and $\mathcal{C}_\mathrm{r}$ of the balls of radius $\sqrt{s}$ in $\mathbb{R}^{m \times r}$ and $\mathbb{R}^{d \times r}$ (w.r.t. the spectral norm) with cardinalities

$$\log\left(|\mathcal{C}_\mathrm{l}|\right) \leqslant [mr] \log\left[1 + \frac{6s}{\epsilon}\right]$$

$$\log\left(|\mathcal{C}_\mathrm{r}|\right) \leqslant [dr] \log\left[1 + \frac{6s}{\epsilon}\right]. \tag{F.2}$$

Let $A = WV^\top \in \mathcal{E}_{\mathrm{r,s}}$, we define $\bar{A} = \overline{W}\overline{V}^\top$ where $\overline{W}$ (resp. $\overline{V}$) is the cover element in $\mathcal{C}_\mathrm{l}$ (resp. $\mathcal{C}_\mathrm{r}$) associated to $W$ (resp. $V$). By definition of the covers via Lemma (I.4) we have

$$\|W - \overline{W}\|_\sigma, \|V - \overline{V}\|_\sigma \leqslant \frac{\epsilon}{2\sqrt{s}}. \tag{F.3}$$

Using this, we certainly have

$$\begin{aligned}
\|A - \bar{A}\|_\sigma &= \|WV^\top - \overline{W}\bar{V}^\top\|_\sigma \\
&\leqslant \|W(V^\top - \bar{V}^\top)\|_\sigma + \|(W - \overline{W})\bar{V}^\top\|_\sigma \\
&\leqslant \|W\|_\sigma \|(V^\top - \bar{V}^\top)\|_\sigma + \|(W - \overline{W})\|_\sigma \|\bar{V}^\top\|_\sigma \\
&< s\frac{\epsilon}{2\sqrt{s}} + \frac{\epsilon}{2\sqrt{s}}s = \epsilon.
\end{aligned} \tag{F.4}$$

Thus, $\mathcal{C} := \left\{ A \in \mathbb{R}^{m \times d} : A = WV^\top, W \in \mathcal{C}_\mathrm{l}, V \in \mathcal{C}_\mathrm{r} \right\}$ is a valid cover of $\mathcal{E}_{\mathrm{r,s}}$ w.r.t. the spectral norm. Furthermore, by equations (F.2) we certainly have

$$\log\left(|\mathcal{C}|\right) \leqslant \log\left(|\mathcal{C}_\mathrm{l}|\right) + \log\left(|\mathcal{C}_\mathrm{r}|\right) \tag{F.5}$$

$$\leqslant [m + d]r \log\left[1 + \frac{6s}{\epsilon}\right], \tag{F.6}$$

as expected. $\qquad\square$

**Proposition F.2** ($L^\infty$ cover of $\mathcal{F}_\mathrm{r}^\mathrm{p}$). *Consider the set of matrices with a bounded Schatten norm as follows: $\mathcal{F}_\mathrm{r}^\mathrm{p} := \left\{ Z \in \mathbb{R}^{m \times d} : \|Z - M\|_{\mathrm{sc},p}^p \leqslant \mathcal{M}^p = \mathrm{r}\,\mathrm{s}^p; \|Z\| \leqslant \mathrm{s} \right\}$ (where $M$ is a fixed reference matrix), viewed as linear maps from $\mathbb{R}^d$ to $\mathbb{R}^m$, and assume as usual that we have a training set $x_1, \ldots, x_N \in \mathbb{R}^d$ such that $\|x_i\| \leqslant b$ for all $i \leqslant N$.*

*There exists a cover $\mathcal{C} \subset \mathcal{F}_\mathrm{r}^\mathrm{p}$ such that for all $Z \in \mathcal{F}_\mathrm{r}^\mathrm{p}$, there exists a $\bar{Z} \in \mathcal{C}$ such that for all $i \leqslant N$, we have*

$$\|(Z - \bar{Z})x_i\|_\infty \leqslant \epsilon \tag{F.7}$$

*and*

$$\log\left(|\mathcal{C}|\right) \leqslant 24 \left[\frac{\mathcal{M}\,b}{\epsilon}\right]^{\frac{2p}{p+2}} [m + d]^{\frac{2}{p+2}} \min(m,d)^{\frac{p}{p+2}} \log_2\left(\Gamma_{\mathcal{F}_\mathrm{r}^\mathrm{p}}\right)$$

$$\leqslant 24 \left[\frac{b\,\mathrm{s}}{\epsilon}\right]^{\frac{2p}{p+2}} [m + d]\,\mathrm{r}^{1 - \frac{p}{p+2}} \log_2\left(\Gamma_{\mathcal{F}_\mathrm{r}^\mathrm{p}}\right),$$

*where $\Gamma_{\mathcal{F}_\mathrm{r}^\mathrm{p}} := \left(\frac{16[\mathcal{M}^p + 1][b+1][1+\mathrm{s}][m+d]}{\epsilon^{\frac{p}{p+2}}} + 7\right) mN$.*

*Remark* F.3. Note that in particular, the proposition holds for $p = 0$, replacing $\mathcal{M}^p$ with a constraint on the rank. In that case, the proposition follows immediately from Proposition F.1. Furthermore, it also follows from the limiting case as $p \to 0$. This is only true thanks to a careful management of the logarithmic factor $\Gamma_{\mathcal{F}_\mathrm{r}^\mathrm{p}}$: indeed, in addition to the constraint $\|Z\|_{\mathrm{sc},p}^p \leqslant \mathcal{M}^p$, we also independently require the constraint $\|Z\| \leqslant \mathrm{s}$. This second constraint is only necessary to maintain good scaling as $p \to 0$. Indeed, we could have bounded $\|Z\|$ via $\|Z\| \leqslant \|Z\|_{\mathrm{sc},p} \leqslant \mathcal{M}$. However, this would result in a factor of $\mathcal{M}$ (without an exponent of $p$) inside the logarithmic term, making the bound blow up at a rate of $1/p$ as $p \to 0$. To see this, note that as $p \to 0$,

$$\log(\|Z\|_{\mathrm{sc},p}) = \frac{\log(\|Z\|_{\mathrm{sc},p}^p)}{p} = O\left(\frac{\log(\mathrm{rank}(Z))}{p}\right) \to \infty. \tag{F.8}$$

*Proof.* The proof relies on the general "parametric interpolation" technique developed in [28]. After a simple translation argument, we can assume without loss of generality that $M = 0$. Let $\tau$ be a threshold to be determined later. We first prove the following claim:

**Claim 1:** *Any matrix $Z \in \mathcal{F}_\mathrm{r}^\mathrm{p}$ can be decomposed into a sum $Z_1 + Z_2$ where $Z_1 \in \mathcal{E}_{\mathrm{r}_1, \mathcal{M}}$ and $Z_2 \in \mathcal{F}_{\mathcal{M}_2}^2$ where $\mathrm{r}_1 = \frac{\mathcal{M}^p}{\tau^p}$ and $\mathcal{M}_2{}^2 = \tau^2 \min(m,d)$.*

**Proof of Claim 1:**

Let $\rho_1, \ldots, \rho_{\min(m,d)}$ denote the singular values of $Z$, so that so that the singular value decomposition of $Z$ is $Z = \sum_k \rho_k\, v_k w_k^\top$ for some unit vectors $v_k \in \mathbb{R}^m$ and $w_k \in \mathbb{R}^d$.

Let $\mathrm{T} = \max\left(k : \rho_k > \tau\right)$ and write $Z_1 = \sum_{k \leqslant \mathrm{T}} \rho_k\, v_k w_k^\top$ and $Z_2 = Z - Z_1 = \sum_{k > \mathrm{T}} \rho_k\, v_k w_k^\top$.

Note that by Markov's inequality, since $\|Z\|_{\text{sc},p}^p \leq \mathcal{M}^p$, we have that $\text{rank}(Z_1) \leq \text{T} \leq \frac{\mathcal{M}^p}{\tau^p}$. Furthermore, we clearly have $\|Z_1\| = \rho_1 = \|Z\| \leq \|Z\|_{\text{sc},p} \leq \mathcal{M}$. Thus, $Z_1 \in \mathcal{E}_{\text{r}_1,\mathcal{M}}$ as expected. Next, it is clear that

$$\|Z_2\|_{\text{Fr}}^2 = \sum_{k=\text{T}+1}^{\min(m,d)} \rho_k^2 \leq \tau^2 \min(m,d). \tag{F.9}$$

Thus, $Z_2 \in \mathcal{F}_{\mathcal{M}_2}^2$ as expected, which *concludes the proof of the claim.*

Thus, it follows that for any choice of $\tau$ and granularity $\epsilon$, we can apply Propositions F.1 and I.3 to obtain $\epsilon/2$ covers $\mathcal{C}_1 \subset \mathcal{E}_{\text{r}_1,\mathcal{M}}$ and $\mathcal{C}_2 \in \mathcal{F}_{\mathcal{M}_2}^2$ such that for any $Z_1 \in \mathcal{E}_{\text{r}_1,\mathcal{M}}$ and for any $Z_2 \in \mathcal{F}_{\mathcal{M}_2}^2$ there exist $\bar{Z}_1 \in \mathcal{C}_1$ and $\bar{Z}_2 \in \mathcal{C}_2$ such that for all $i \leq N$, $\|(Z_1 - \bar{Z}_1)x_i\|_\infty \leq \epsilon/2$ and $\|(Z_2 - \bar{Z}_2)x_i\|_\infty \leq \epsilon/2$. It is then clear that for any $Z \in \mathcal{F}_{\text{r}}^{\text{p}}$, the element $\bar{Z} = \bar{Z}_1 + \bar{Z}_2 \in \mathcal{E}_{\text{r}_1,\mathcal{M}} + \mathcal{F}_{\mathcal{M}_2}^2$ satisfies $\|(Z - \bar{Z})x_i\|_\infty \leq \epsilon$. Thus, $\mathcal{C} = \mathcal{C}_1 + \mathcal{C}_2$ is the required $\epsilon$ cover of $\mathcal{F}_{\text{r}}^{\text{p}}$ (which satisfies inequality F.7), and by Propositions F.1 (with $\epsilon \leftarrow \epsilon/(2b)$ since the required granularity is $\epsilon/2$ and Proposition F.1 gives a cover w.r.t. the spectral norm) and I.3 (with $\epsilon \leftarrow \epsilon/2$) its cardinality satisfies:

$$\log(|\mathcal{C}|) \leq \log(|\mathcal{C}_1|) + \log(|\mathcal{C}_2|)$$

$$\leq [m+d]\,\text{r}_1 \log\left[1 + \frac{12b\,\text{s}}{\epsilon}\right] + \frac{144\,\mathcal{M}_2^2\,b^2}{\epsilon^2} \log_2\left[\left(\frac{16\,\mathcal{M}_2\,b}{\epsilon} + 7\right)md\right]$$

$$= [m+d]\frac{\mathcal{M}^p}{\tau^p} \log\left[1 + \frac{12\,\text{s}\,b}{\epsilon}\right] + \frac{144\,\tau^2 \min(m,d)b^2}{\epsilon^2} \log_2\left[\left(\frac{16\,\tau \min(m,d)b}{\epsilon} + 7\right)mN\right], \tag{F.10}$$

where at Equation (F.10) we have substituted the values of $\text{r}_1$ and $\mathcal{M}_2$. This motivates the following choice of $\tau$:

$$\tau = 144^{-\frac{1}{p+2}}\,\mathcal{M}^{\frac{p}{p+2}}\,\epsilon^{\frac{2}{p+2}}b^{-\frac{2}{p+2}}\left[\frac{[m+d]}{\min(m,d)}\right]^{\frac{1}{p+2}}, \tag{F.11}$$

which leads to the following covering number bound:

$$\log(|\mathcal{C}|) \leq 144^{\frac{p}{p+2}}\left[\frac{\mathcal{M}\,b}{\epsilon}\right]^{\frac{2p}{p+2}} \frac{[m+d]^{\frac{2}{p+2}}}{\min(m,d)^{-\frac{p}{p+2}}}\left[\log\left[1 + \frac{12\,\text{s}\,b}{\epsilon}\right]\right.$$

$$\left. + \log_2\left[\left[\frac{16[\frac{\mathcal{M}^p\epsilon}{b}]^{\frac{2}{2+p}} \min(m,d)^{\frac{p+1}{p+2}}[m+d]^{\frac{1}{p+2}}b}{\epsilon} + 7\right]md\right]\right]$$

$$\leq 24\left[\frac{\mathcal{M}\,b}{\epsilon}\right]^{\frac{2p}{p+2}}[m+d]^{\frac{2}{p+2}} \min(m,d)^{\frac{p}{p+2}} \log_2\left[\left(\frac{16[\mathcal{M}^p+1][b+1][1+\text{s}][m+d]}{\epsilon^{\frac{p}{p+2}}} + 7\right)mN\right],$$

where at the last line, we have used the fact that $p \leq 2$ and therefore $144^{\frac{p}{p+2}} \leq 12$. The proof is complete. $\qquad\square$

We have the following very direct $L^2$ analogue, which follows directly by applying Proposition F.2 with $\epsilon \leftarrow \epsilon/\sqrt{m}$.

**Proposition F.4** ($L^2$ cover of $\mathcal{F}_{\text{r}}^{\text{p}}$). *Consider the set of matrices with a bounded Schatten norm as follows: $\mathcal{F}_{\text{r}}^{\text{p}} := \{Z \in \mathbb{R}^{m\times d} : \|Z - M\|_{\text{sc},p}^p \leq \mathcal{M}^p = \text{r}\,\text{s}^p; \|Z\| \leq \text{s}\}$ (where $M$ is a fixed referenece matrix) viewed as linear maps from $\mathbb{R}^d$ to $\mathbb{R}^m$, and assume as usual that we have a training set $x_1, \ldots, x_N \in \mathbb{R}^d$ such that $\|x_i\| \leq b$ for all $i \leq N$.*

*There exists a cover $\mathcal{C} \subset \mathcal{F}_{\text{r}}^{\text{p}}$ such that for all $Z \in \mathcal{F}_{\text{r}}^{\text{p}}$, there exists a $\bar{Z} \in \mathcal{C}$ such that for all $i \leq N$, we have*

$$\|(Z - \bar{Z})x_i\|_2 \leq \epsilon. \tag{F.12}$$

*and*

$$\log(|\mathcal{C}|) \leq 24\left[\frac{\mathcal{M}\,b}{\epsilon}\right]^{\frac{2p}{p+2}}[m+d]^{\frac{2}{p+2}} \min(m,d)^{\frac{p}{p+2}}[m]^{\frac{p}{p+2}} \log_2\left(\Gamma'_{\mathcal{F}_{\text{r}}^{\text{p}}}\right)$$

$$\leq 24\left[\frac{b\,\text{s}}{\epsilon}\right]^{\frac{2p}{p+2}}[m+d]^{1+\frac{p}{p+2}}\,\text{r}^{1-\frac{p}{p+2}} \log_2\left(\Gamma'_{\mathcal{F}_{\text{r}}^{\text{p}}}\right),$$

*where* $\Gamma'_{\mathcal{F}_\mathrm{r}^\mathrm{P}} := \left( \dfrac{16[\mathcal{M}^p + 1][b+1][1+s][m+d]^2}{\epsilon^{\frac{p}{p+2}}} + 7 \right) mN.$

# G   Extension to Convolutional Neural Networks (CNN)

In this section, we generalize our results to Convolutional Neural Networks. In particular, this section culminates in Theorem G.5, which provides a post hoc generalization bound for convolutional neural networks whose weight matrices have low Schatten $p$ quasi norm.

We first introduce some more detailed notation, which generally follows the existing literature on norm based bounds [16] and is reintroduced for completeness. One difference is that we assume that the pooling operation only operates on the spatial dimension (which is standard in real life CNNs applications), which results in the single notation $U_\ell$ for the channel dimension at layer $\ell$, which replaces both notations $m_\ell$ and $U_\ell$ from [16].

Let $x \in \mathbb{R}^{U \times w}$ denote an input 'image', $A \in \mathbb{R}^{U \times d}$ and $S^1, S^2, \ldots, S^O$ be $O$ ordered subsets of $(\{1, 2, \ldots, w\} \times \{1, 2, \ldots, U\})$ each of cardinality $d$ corresponding to the convolutional 'patches'. We will write $\Lambda_A(x) \in \mathbb{R}^{U \times O}$ for the image of $x$ under the convolutional operation associated with $A$: $\Lambda_A(x)_{j,o} = \sum_{i=1}^d X_{S_i^o} A_{j,i}$. The sets $S^1, S^2, \ldots, S^O$ represent the image patches where the convolutional filters are applied, and $\Lambda$ would be represented via the "tf.nn.conv2d" function in Tensorflow (or "torch.nn.functional.conv2d" in Pytorch). To represent the patches at a given layer, we add a layer index $\ell$. For instance, if the input is a $28 \times 28 \times 1$ image and the convolutional filters are of size $2 \times 2$ with stride 2, $S^{\ell-1=0, o=1} = S^{0,1} = \{(1,1), (1,2), (2,1), (2,2)\}$ represents the first convolutional patch, and $S^{0,2} = \{(3,1), (3,2), (4,1), (4,2)\}$ represents the second convolutional patch.

We will also write $\mathrm{op}(A^\ell)$ for the matrix in $\mathbb{R}^{U_\ell O_{\ell-1} \times U_{\ell-1} w_{\ell-1}}$ that represents the convolution operation $\Lambda_{A^\ell}$ (after unravelling all the dimensions), as we will rely on its sepctral norm $\|\mathrm{op}(A^\ell)\|$ in calculations. To represent the full architecture, we use the index $\ell$ to represent the $\ell$'th layer: for instance, the $\ell$th layer's convolutional operation is denoted by $\Lambda_{A^\ell} : \mathbb{R}^{U_{\ell-1} \times w_{\ell-1}} \to \mathbb{R}^{U_\ell \times O_{\ell-1}}$ and acts on the $\ell - 1$th layer's activations in the space. It is also assumed that the last layer is fully-connected.

The architecture above can help us represent a feedforward neural network involving possible (intra-layer) weight sharing as

$$F^{\mathfrak{C}}_{A^1, A^2, \ldots, A^L} : \mathbb{R}^{U_0 \times w_0} \to \mathbb{R}^{U_L \times w_l} : x \mapsto (\sigma_L \circ \Lambda_{A^L} \circ \sigma_{L-1} \circ \Lambda_{A^{L-1}} \circ \ldots \sigma_1 \circ \Lambda_{A^1})(x), \quad \text{(G.1)}$$

where for each $\ell \leqslant L$, the weight $A^\ell$ is a matrix in $\mathbb{R}^{w \times d_\ell}$ and for each $\ell$, $\sigma_\ell : \mathbb{R}^{U_\ell \times O_\ell} \to \mathbb{R}^{U_\ell \times w_\ell}$ is $L^\infty$ Lipschitz with constant $\rho_\ell$ and represents both the elementwise activation operation and any spatial pooling.

## G.1   One Layer Case

We now consider the covering number of the class of functions corresponding to a layer's convolutional operation with a matrix $A$ such that $\|A - M\|_{\mathrm{sc}, p} \leqslant \mathcal{M}$ for some number $\mathcal{M}$.

**Proposition G.1** ($L^\infty$ cover of $\mathfrak{C}_r^p$ for a convolutional layer)**.** *Assume we are given a training set* $x_1, \ldots, x_N \in \mathbb{R}^{U_0 \times w_0}$ *such that* $\sup_{i,o} \|(x_i)_{S^{0,o}}\| \leqslant b$. *Consider the set of matrices with a bounded Schatten norm as follows:* $\mathcal{F}_\mathrm{r}^\mathrm{P} := \left\{ Z \in \mathbb{R}^{U' \times d} : \|Z - M\|_{\mathrm{sc}, p}^p \leqslant \mathcal{M}^p = \mathrm{r}\, \mathrm{s}^p; \|\mathrm{op}(Z)\| \leqslant \mathrm{s} \right\}$ *(where* $M$ *is an arbitrary reference matrix) and the associated class* $\mathfrak{C}_r^p := \{\Lambda_A : A \in \mathcal{F}_\mathrm{r}^\mathrm{P}\}$ *of linear maps representing the assocated convolution operations from* $\mathbb{R}^{U \times w}$ *to* $\mathbb{R}^{U' \times w'}$.

*There exists a cover* $\mathcal{C} \subset \mathfrak{C}_r^p$ *such that for all* $Z \in \mathfrak{C}_r^p$, *there exists a* $\bar{Z} \in \mathcal{C}$ *such that for all* $i \leqslant N$, *we have*

$$\|(\Lambda_Z - \Lambda_{\bar{Z}})(x_i)\|_\infty \leqslant \epsilon \quad \text{(G.2)}$$

*and*

$$\log\left(|\mathcal{C}|\right) \leqslant 24\left[\frac{\mathcal{M}\,b}{\epsilon}\right]^{\frac{2p}{p+2}}[U'+d]^{\frac{2}{p+2}}\min(U',d)^{\frac{p}{p+2}}\log_2\left(\Gamma_{\mathfrak{C}_r^p}\right)$$

$$\leqslant 24\left[\frac{b\,\mathrm{s}}{\epsilon}\right]^{\frac{2p}{p+2}}[U'+d]\min(U',d)^{\frac{p}{p+2}}\,\mathrm{r}^{1-\frac{p}{p+2}}\log_2\left(\Gamma_{\mathfrak{C}_r^p}\right),$$

*where* $\Gamma_{\mathfrak{C}_r^p} := \left(\frac{16[\mathcal{M}^p+1][b+1][1+\mathrm{s}][U'+d]}{\epsilon^{\frac{p}{p+2}}} + 7\right)U'NO.$

*Proof.* Note that $\|Z\| \leqslant \|\operatorname{op}(Z)\| \leqslant \mathrm{s}$. Thus, the proposition follows immediately from Proposition F.2 applied to the auxiliary dataset collecting each input convolutional patch as a sample and replacing $N \leftarrow NO$ and $m \leftarrow U'$.

$\square$

We also have the following immediate $L^2$ version, which follows from the previous proposition with $\epsilon \leftarrow \frac{\epsilon}{U'w'}$ and utilizing the $L^\infty$ Lipschitzness of the nonlinearity $\sigma$.

**Proposition G.2** ($L^2$ *cover of* $\mathfrak{C}_r^p$ *for a convolutional layer*)**.** *Instate the notation of Proposition G.1 and consider a $\rho$-Lipschitz activation function $\sigma : \mathbb{R}^{U'\times O} \to \mathbb{R}^{U'\times w'}$. There exists a cover $\mathcal{C}$ such that for all $Z \in \mathfrak{C}_r^p$, there exists a $\bar{Z} \in \mathcal{C}$ such that for all $i \leqslant N$, we have*

$$\|(\Lambda_Z - \Lambda_{\bar{Z}})(x_i)\|_{\mathrm{Fr}} \leqslant \epsilon\rho \tag{G.3}$$

*and*

$$\log\left(|\mathcal{C}|\right) \leqslant 24\left[\frac{\mathcal{M}\,b}{\epsilon}\right]^{\frac{2p}{p+2}}[U'+d]^{\frac{2}{p+2}}\min(U',d)^{\frac{p}{p+2}}[U'w']^{\frac{p}{p+2}}\log_2\left(\Gamma'_{\mathfrak{C}_r^p}\right)$$

$$\leqslant 24\left[\frac{b\,\mathrm{s}}{\epsilon}\right]^{\frac{2p}{p+2}}[U'+d]\,\mathrm{r}^{1-\frac{p}{p+2}}[U'w']^{\frac{p}{p+2}}\log_2\left(\Gamma'_{\mathfrak{C}_r^p}\right),$$

*where* $\Gamma'_{\mathfrak{C}_r^p} := \left(\frac{16[\mathcal{M}^p+1][b+1][1+\mathrm{s}][U'+d]U'w'}{\epsilon^{\frac{p}{p+2}}} + 7\right)U'NO.$

We now move on to our covering number bound for the class of CNNs with fixed norm constraints: we write

$$\mathcal{F}_{\mathcal{B}}^{\mathfrak{C}} := \left\{F_A = F_{(A_1,\dots,A_L)}^{\mathfrak{C}} : \|A_\ell - M_\ell\|_{\mathrm{sc}} \leqslant \mathcal{M}_\ell \quad \|\operatorname{op}(A_\ell)\| \leqslant s_\ell \quad \forall\ell < L \quad \|A_L^\top\|_{2,\infty} \leqslant s_L\right\}$$

for the class of all neural networks whose matrices satisfy the norm constraints with fixed constraints $\mathcal{M}_\ell, s_\ell$ etc.

## G.2 Covering Number For the Class $\mathcal{F}_{\mathcal{B}}^{\mathfrak{C}}$

We have the following analogue of Proposition E.11 for the convolutional case:

**Proposition G.3.** *Let* $x_1,\dots,x_N$ *be a given training set satisfying* $\|x_i\| \leqslant b$ *for all* $i$. *Consider the class* $\mathcal{F}_{\mathcal{B}}^{\mathfrak{C}} := \left\{F_A = F_{(A_1,\dots,A_L)}^{\mathfrak{C}} : \|A_\ell - M_\ell\|_{\mathrm{sc}} \leqslant \mathcal{M}_\ell \quad \|\operatorname{op}(A_\ell)\| \leqslant s_\ell \quad \forall\ell < L \quad \|A_L^\top\|_{2,\infty} \leqslant s_L\right\}$ *of all CNNs satisfying a set of fixed norm constraints for a given set of* $p_\ell, \mathcal{M}_\ell, s_\ell$. *We have the following covering number bound for the loss class* $\mathrm{l}\circ\mathcal{F}_{\mathcal{B}}^{\mathfrak{C}}$ *for any* $\epsilon \geqslant \frac{1}{N}$:

$$\log(\mathcal{N}_\infty(\mathrm{l}\circ\mathcal{F}_{\mathcal{B}}^{\mathfrak{C}},\epsilon)) \leqslant 72\log_2(4\underline{\Gamma}^{\mathfrak{C}})\left[\frac{(1+L\,\mathrm{L_l})}{\min(\varepsilon,1)}\right]^{\frac{2p}{2+p}} \times \tag{G.4}$$

$$\sum_{\ell=1}^{L}\left[b\prod_i\rho_i s_i\right]^{\frac{2p_\ell}{p_\ell+2}}\left[\frac{\mathcal{M}_\ell^{p_\ell}}{s_\ell^{p_\ell}}\right]^{\frac{2}{p_\ell+2}}[U_\ell+d_{\ell-1}]^{\frac{2}{p_\ell+2}}\min(U_\ell,d_{\ell-1})^{\frac{p_\ell}{p_\ell+2}}W_\ell^{\frac{p_\ell}{p_\ell+2}},$$

*where* $\underline{\Gamma}^{\mathfrak{C}} := [b+1][L\,\mathrm{L_l}+1]\mathcal{W}\mathcal{A}N\prod_{i=1}^{L}[[\rho_i+1]s_i+1].$

*Proof.* The proof is similar to that of Proposition E.11: most differences are in the detailed calculations.

As before, for any choice of $\epsilon_1, \ldots, \epsilon_L$, by Lemma E.12, we know that $l \circ \mathcal{F}_{\mathcal{B}}^{\mathfrak{C}}$ admits an $L^\infty$ cover $\mathcal{C}$ with granularity $\epsilon = \sum_{\ell=1}^{L} \epsilon_\ell \rho_{\ell \to L}$ and cardinality

$$\log\left(|\mathcal{C}|\right) \leqslant \sum_{\ell=1}^{L} \log(\mathcal{N}_{\infty,\ell}(\mathcal{B}_\ell^{\mathfrak{C}}, \epsilon_\ell, N, b\rho_{0 \to \ell-1})), \tag{G.5}$$

where $\mathcal{B}_\ell^{\mathfrak{C}} := \left\{ A_\ell \in \mathbb{R}^{U_\ell \times d_{\ell-1}} : \|A_\ell - M_\ell\|_{\mathrm{sc},p_\ell} \leqslant \mathcal{M}_\ell; \, \|\mathrm{op}(A_\ell)\| \leqslant s_\ell \right\}$ (for $\ell \neq L$) and $\mathcal{B}_L^{\mathfrak{C}} := \left\{ A_\ell \in \mathbb{R}^{U_\ell \times d_{\ell-1}} : \|A_L - M_L\|_{\mathrm{sc},p_L} \leqslant \mathcal{M}_L; \, \|\mathrm{op}(A_L)\| \leqslant s_L \right\}$.

We now use the following notation: we write $U_\ell$ for the number of channels at layer $\ell$ and $d_\ell$ for the number of pixels in each convolutional patch at layer $\ell$, so that $\min(U_\ell, d_{\ell-1})$ is the minimum dimension of the filter applied at layer $\ell$. we also write $W_\ell := U_\ell \times w_\ell$ with the convention that $W_L = 1$. We also write $\mathcal{W} := \sum_{\ell=1}^{L} d_{\ell-1} \times U_\ell$ for the total number of parameters in the network and $\mathcal{A} := \sum_{\ell=0}^{L} U_\ell \times O_{\ell-1}$ for the total number of preactivations/input dimensions including both the input and the final layer output. Note that by Propositions G.1 and G.2, we can continue from equation (G.5) as follows:

$$\log\left(|\mathcal{C}|\right) \leqslant 24 \sum_{\ell=1}^{L} \left[ \frac{\mathcal{M}_\ell \, b\rho_{0 \to \ell-1}}{\epsilon_\ell} \right]^{\frac{2p_\ell}{p_\ell+2}} [U_\ell + d_{\ell-1}]^{\frac{2}{p_\ell+2}} \min(U_\ell, d_{\ell-1})^{\frac{p_\ell}{p_\ell+2}} W_\ell^{\frac{p_\ell}{p_\ell+2}} \log_2(\Gamma_{\mathcal{F}_\mathrm{r}^{\mathrm{P}},\ell}^{\mathfrak{C}}), \tag{G.6}$$

where for all $\ell \leqslant L$, $\Gamma_{\mathcal{F}_\mathrm{r}^{\mathrm{P}},\ell}^{\mathfrak{C}} := \left\lceil \left( \frac{16[\mathcal{M}_\ell{}^{p_\ell}+1][b\prod_{i=1}^{\ell}\rho_i s_i+1][1+\sqrt{\mathcal{A}}s_\ell]\mathcal{W}\mathcal{A}}{\epsilon_\ell^{\frac{p_\ell}{p_\ell+2}}} + 7 \right) N\mathcal{A} \right\rceil$ (indeed, since $\|A_L^\top\|_{2,\infty} \leqslant s_L$, we certainly have $\|A_L\| \leqslant s_L\mathcal{A}$).

We now set $\epsilon_\ell = \frac{\epsilon}{(L\,\mathrm{L_l}+1)\rho_{\ell \to L}}$ as before and bound all the logarithmic factors as follows:

$$\max_\ell \Gamma_{\mathcal{F}_\mathrm{r}^{\mathrm{P}},\ell}^{\mathfrak{C}} \leqslant \left\lceil \left( \frac{16[\max_\ell \frac{\mathcal{M}_\ell{}^{p_\ell}}{s_\ell{}^{p_\ell}}+1]\prod_{i=1}^{L}[[\rho_i+1]s_i+1]^3[b+1][L\,\mathrm{L_l}+1]\mathcal{W}\mathcal{A}^{1.5}}{\min(\epsilon,1)} + 7 \right) N\mathcal{A} \right\rceil$$

$$\leqslant \left\lceil 64 \prod_{i=1}^{L}[[\rho_i+1]s_i+1]^3[b+1][\mathrm{L_l}+1]\mathcal{W}\mathcal{A}^3 N^2 \right\rceil \leqslant [4\underline{\Gamma}^{\mathfrak{C}}]^3 \tag{G.7}$$

where $\underline{\Gamma}^{\mathfrak{C}} := [b+1][L\,\mathrm{L_l}+1]\mathcal{W}\mathcal{A}N\prod_{i=1}^{L}[[\rho_i+1]s_i+1]$. Plugging the value of epsilon into equation (G.6) we can continue with a calculation similar to that of equations (E.143):

$$\log\left(|\mathcal{C}|\right)$$
$$\leqslant 72\log_2(4\underline{\Gamma}^{\mathfrak{C}})\sum_{\ell=1}^{L}\left[\frac{\mathcal{M}_\ell \, b\rho_{0\to\ell-1}}{\epsilon_\ell}\right]^{\frac{2p_\ell}{p_\ell+2}}[U_\ell+d_{\ell-1}]^{\frac{2}{p_\ell+2}}\min(U_\ell,d_{\ell-1})^{\frac{p_\ell}{p_\ell+2}}W_\ell^{\frac{p_\ell}{p_\ell+2}}$$
$$\leqslant 72\log_2(4\underline{\Gamma}^{\mathfrak{C}})\left[\frac{(1+L\,\mathrm{L_l})}{\min(\varepsilon,1)}\right]^{\frac{2p}{2+p}} \times$$
$$\sum_{\ell=1}^{L}\left[b\prod_i\rho_i s_i\right]^{\frac{2p_\ell}{p_\ell+2}}\left[\frac{\mathcal{M}_\ell{}^{p_\ell}}{s_\ell{}^{p_\ell}}\right]^{\frac{2}{p_\ell+2}}[U_\ell+d_{\ell-1}]^{\frac{2}{p_\ell+2}}\min(U_\ell,d_{\ell-1})^{\frac{p_\ell}{p_\ell+2}}W_\ell^{\frac{p_\ell}{p_\ell+2}},$$

as expected. $\qquad\square$

### G.3 Generalization Bound with Fixed Constraints (CNNs)

Based on the above, we obtain the following analogue of Proposition E.1:

**Proposition G.4.** *Fix some reference/initialization matrices $M^1 \in \mathbb{R}^{w_1 \times w_0} = \mathbb{R}^{w_1 \times d}, \ldots, M^L \in \mathbb{R}^{w_L \times w_{L-1}} = \mathbb{R}^{C \times w_{L-1}}$. Assume also that the inputs $x$ satisfy $\|x\| \leqslant b$ w.p. 1. Fix a set of constraint parameters $0 \leqslant p_\ell \leqslant 2, s_\ell, \mathcal{M}_\ell$ (for $\ell = 1, \ldots, L$).*

*With probability greater than $1 - \delta$ over the draw of an i.i.d. training set $x_1, \ldots, x_N$, every neural network $F_A^{\mathfrak{C}}(x)$ defined with equation (G.1) satisfying the following conditions:*

$$
\begin{aligned}
\|A^\ell - M^\ell\|_{\mathrm{sc}, p_\ell} &\leqslant \mathcal{M}_\ell && \forall 1 \leqslant \ell \leqslant L \\
\|\operatorname{op}(A)_\ell\| &\leqslant s_\ell && \forall 1 \leqslant \ell \leqslant L - 1 \\
\|A_L^\top\|_{2,\infty} &\leqslant s_L,
\end{aligned}
\tag{G.8}
$$

*also satisfies the following generalization bound:*

$$
\mathbb{E}\left[\mathrm{l}(F_A(x), y)\right] - \frac{1}{N}\sum_{i=1}^N \mathrm{l}(F_A(x_i), y_i)
$$

$$
\leqslant 6[\mathcal{B}+1]\sqrt{\frac{\log(\frac{4}{\delta})}{N}} + \frac{208\sqrt{\log_2(4\underline{\Gamma}^{\mathfrak{C}})}\log(N)[\mathcal{B}+1]}{\sqrt{N}}(1 + L\, \mathrm{L_l})^{\frac{p}{2+p}}\mathcal{R}_{\mathcal{M},\mathrm{s},p,b}^{\mathfrak{C}},
\tag{G.9}
$$

*where $\mathcal{R}_{\mathcal{M},\mathrm{s},p,b}^{\mathfrak{C}} :=$*

$$
\left[\sum_{\ell=1}^L \left[b\prod_i \rho_i s_i\right]^{\frac{2p_\ell}{p_\ell+2}} \left[\frac{\mathcal{M}_\ell^{p_\ell}}{s_\ell^{p_\ell}}\right]^{\frac{2}{p_\ell+2}} \mathrm{r}_{\mathfrak{C},\ell}^{\frac{2}{p_\ell+2}}[U_\ell + d_{\ell-1}]^{\frac{2}{p_\ell+2}}\min(U_\ell, d_{\ell-1})^{\frac{p_\ell}{p_\ell+2}}W_\ell^{\frac{p_\ell}{p_\ell+2}}\right]^{\frac{1}{2}}
\tag{G.10}
$$

*with $\mathrm{r}_{\mathfrak{C},\ell} := \frac{\mathcal{M}_\ell^{p_\ell}}{s_\ell^{p_\ell}}$ and $\underline{\Gamma}^{\mathfrak{C}} := [b+1][L\, \mathrm{L_l}+1]\mathcal{WAN}\prod_{i=1}^L[[\rho_i + 1]s_i + 1]$.*

*Proof.* The proof is exactly the same as that of Proposition E.11 with $\mathrm{r}_\ell^{\frac{2}{p_\ell+2}}\bar{w}_\ell^{\frac{2}{p_\ell+2}}\tilde{w}_\ell^{\frac{p_\ell}{p_\ell+2}}$ replaced by $\mathrm{r}_{\mathfrak{C},\ell}^{\frac{2}{p_\ell+2}}[U_\ell + d_{\ell-1}]^{\frac{2}{p_\ell+2}}\min(U_\ell, d_{\ell-1})^{\frac{p_\ell}{p_\ell+2}}W_\ell^{\frac{p_\ell}{p_\ell+2}}$ and the initial constant of 98 replaced by 72. $\square$

### G.4 Post Hoc Result for CNNs

Based on the above, we are finally in a position to present the following analogue of Theorem E.2 for the convolutional case:

**Theorem G.5.** *With probability greater than $1 - \delta$, for any $b \in \mathbb{R}^+$, every trained neural network $F_A$ and sampling distribution with $\|x\| \leqslant b$ w.p. 1 satisfy the following generalization gap for all possible values of the $p_\ell s$:*

$$
\mathbb{E}\left[\mathrm{l}(F_A(x), y)\right] - \frac{1}{N}\sum_{i=1}^N \mathrm{l}(F_A(x_i), y_i)
\tag{G.11}
$$

$$
\leqslant 6[\mathcal{B}+1]\sqrt{\frac{\log(1/\delta)}{N}} + 6\,\mathcal{B}\sqrt{\frac{2 + \Theta_{\log}^{\mathfrak{C}}}{N}} + 416[\mathcal{B}+1](1 + L\, \mathrm{L_l})^{\frac{p}{2+p}}\frac{\sqrt{\log(\gamma_{\mathfrak{C}})}\log(N)}{\sqrt{N}}\mathcal{R}_{F_A}^{\mathfrak{C}},
$$

*where*

$$
\Theta_{\log}^{\mathfrak{C}} :=
$$

$$
2\log\left[4\mathcal{WA} + \sup_{i=1}^N \|x_i\|\prod_i \rho_i\|\operatorname{op}(A_i)\|\right]\left[|\log(\sup_i \|x_i\|)| + L\left[\sum_{\ell=1}^L |\log(\|\operatorname{op}(A_\ell)\|)| + 2\log(4\mathcal{AW})\right]\right]
$$

$$
\gamma_{\mathfrak{C}} :=
$$

$$
12\left[[\mathfrak{B}+1][L\, \mathrm{L_l}+1]\mathcal{WAN}\prod_{i=1}^L[\rho_i\|\operatorname{op}(A_i)\| + 1]\right]\left[4\overline{W} + \mathfrak{B}\prod_i \rho_i\|\operatorname{op}(A_i)\|\right].
$$

*and where*

$$\mathcal{R}_{F_A}^{\mathfrak{C}} := \left[ \sum_{\ell=1}^{L-1} \left[ \sup_i \|x_i\| \rho_L \|A_L^\top\|_{2,\infty} \prod_{i=1}^{L-1} \rho_i \| \operatorname{op}(A_i)\| \right]^{\frac{2p_\ell}{p_\ell+2}} \left[ \frac{\|A_\ell - M_\ell\|_{\text{sc},p_\ell}^{p_\ell}}{\|\operatorname{op}(A_\ell)\|^{p_\ell}} \right]^{\frac{2}{p_\ell+2}} \right.$$

$$\left. + \left[ b\rho_L \|A_L^\top\|_{2,\infty} \prod_{i=1}^{L-1} \rho_i \|A_i\| \right]^{\frac{2p_L}{p_L+2}} \left[ \frac{\|A_L - M_L\|_{\text{sc},p_L}^{p_L}}{\|A_L^\top\|_{2,\infty}^{p_L}} \right]^{\frac{2}{p_L+2}} \bar{w}_L^{\frac{2}{p_L+2}} \underline{w}_L^{\frac{p_L}{p_L+2}} \right]^{\frac{1}{2}}.$$

*Proof.* The proof is the same as that of Theorem E.2 with the constant 240 replaced by 208, with $\|A_\ell\|$ replaced by $\|\operatorname{op}(A_\ell)\|$, and $\overline{W}$ replaced by $\mathcal{A}\mathcal{W}$.

$\square$

### G.5  The case $p_\ell = 0$ (CNN)

For $p_\ell = 0$ we obtain the following immediate corollary from Proposition G.4.

**Corollary G.6.** *Instate the notation and assumptions of Proposition G.4 , and assume additionally that $p_\ell = 0$ for all $\ell$. We have*

$$\mathbb{E}\left[\mathrm{l}(F_A(x), y)\right] - \frac{1}{N} \sum_{i=1}^N \mathrm{l}(F_A(x_i), y_i)$$

$$\leqslant 6[\mathcal{B}+1]\sqrt{\frac{\log(\frac{4}{\delta})}{N}} + \frac{208\sqrt{\log_2(4\underline{\Gamma}^{\mathfrak{C}})}\log(N)[\mathcal{B}+1]}{\sqrt{N}} \left[ \sum_{\ell=1}^L [U_\ell + d_{\ell-1}]\tilde{r}_\ell \right], \quad (\text{G.12})$$

*where $\tilde{r}_\ell$ is an a priori upper bound on $\operatorname{rank}(A_\ell)$ and as usual, $\underline{\Gamma}^{\mathfrak{C}} = [b + 1][L\,L_\mathrm{l} +1]\mathcal{W}\mathcal{A}N \prod_{i=1}^L [[\rho_i + 1]s_i + 1]$.*

With a union bound, we achieve the following post hoc version:

**Theorem G.7.** *W.p. $\geqslant 1 - \delta$ over the draw of the training set, every neural network satisfies:*

$$\mathbb{E}\left[\mathrm{l}(F_A(x), y)\right] - \frac{1}{N} \sum_{i=1}^N \mathrm{l}(F_A(x_i), y_i) \leqslant 6[\mathcal{B}+1]\sqrt{\frac{\log(\frac{4}{\delta})}{N}} + \frac{208\sqrt{\log_2(4\Gamma_R^{\mathfrak{C}})}\log(N)[\mathcal{B}+1]}{\sqrt{N}}$$

$$(\text{G.13})$$

*where $\Gamma_R^{\mathfrak{C}} = \left[ [\mathfrak{B} + 2][L\,L_\mathrm{l} +1]\mathcal{W}\mathcal{A}N \prod_{i=1}^L [[\rho_i + 1][\|A_i\| + 2] + 1] \right].$*

*Proof.* The proof is nearly exactly the same as that of Theorem E.8. $\square$

### G.6  Results for CNNs with Loss Function Augmentation

**Proposition G.8.** *For any granularity $\epsilon \geqslant \frac{1}{N}$ and any values of $1 \leqslant b_{\ell_1}, \ldots b_{L-1}, s_\ell, \rho_\ell$, there exists a cover $\mathcal{C}$ of the augmented loss class $\mathcal{L}^{\mathfrak{C},\lambda_{aug}} := \{\lambda_{aug}(F_A, x, y) : F_A \in \mathcal{F}_\mathcal{B}^{\mathfrak{C}}\}$ with cardinality bounded as follows:*

$$\log\left(|\mathcal{C}|\right) \leqslant 216 \log(4\underline{\Gamma}^{\mathfrak{C},\lambda_{aug}}) \times \quad (\text{G.14})$$

$$\left[ \frac{L(1 + L_\mathrm{l})}{\min(\epsilon, 1)} \right]^{\frac{2p}{2+p}} \sum_{\ell=1}^L \left[ b_{\ell-1} \prod_{i=\ell}^L s_i \rho_i \right]^{\frac{2p_\ell}{p_\ell+2}} [U_\ell + d_{\ell-1}]^{\frac{2}{p_\ell+2}} \min(U_\ell, d_{\ell-1})^{\frac{p_\ell}{p_\ell+2}} W_\ell^{\frac{p_\ell}{p_\ell+2}}.$$

*Proof.* The proof is similar to that of Proposition G.3 and E.4 with the main difference only in the calculation of the log terms. By Lemma E.3 and Propositions G.1 and G.2, there exists a cover $\mathcal{C}$ of

the loss class $\lambda_{aug}(F_A)$ with cardinality satisfying

$$\log(|\mathcal{C}|) \leqslant \sum_{\ell=1}^{L} \log(\mathcal{N}_{\infty,\ell}(\mathcal{B}_\ell^{\mathfrak{C}}, \epsilon_\ell, N, 3b_{\ell-1})) \tag{G.15}$$

$$\leqslant 72 \sum_{\ell=1}^{L} \left[ \frac{\mathcal{M}_\ell b_{\ell-1}}{\epsilon_\ell} \right]^{\frac{2p_\ell}{p_\ell+2}} [U_\ell + d_{\ell-1}]^{\frac{2}{p_\ell+2}} \min(U_\ell, d_{\ell-1})^{\frac{p_\ell}{p_\ell+2}} W_\ell^{\frac{p_\ell}{p_\ell+2}} \log_2(\Gamma_{\mathcal{F}_r^P,\ell}^{\mathfrak{C},\lambda_{aug}}).$$

Next, recall that

$$\epsilon_\ell = \frac{b_\ell \epsilon}{L \hat{\rho}_\ell} \tag{G.16}$$

and $\hat{\rho}_{\ell_1} = \max_\ell \rho_{\ell_1 \to \ell}/b_\ell \leqslant \max_\ell \rho_{\ell_1 \to \ell}/(\min(1, L_l^{-1})) \leqslant \rho_{\ell_1 \to \ell} \rho_i s_i [1 + L_l] \leqslant \rho_{\ell_1 \to L} \rho_i s_i [1 + L_l] = \rho_i \prod_{i=\ell_1+1}^{L} L \rho_i s_i [1 + L_l]$, thus

$$\epsilon_\ell \geqslant \frac{\epsilon}{L \rho_{\ell \to L}[1 + L_l]}. \tag{G.17}$$

Thus, by a nearly identical calculation to that in equation (G.7) we obtain

$$\max_\ell \Gamma_{\mathcal{F}_r^P,\ell}^{\mathfrak{C},\lambda_{aug}} \leqslant \left[ \left( \frac{16[\max_\ell \frac{\mathcal{M}_\ell{}^{p_\ell}}{s_\ell{}^{p_\ell}} + 1] \prod_{i=1}^{L}[[\rho_i + 1]s_i + 1]^3 [b+1][L\,L_l+1]\mathcal{W}\mathcal{A}^{1.5}}{\min(\epsilon, 1)} + 7 \right) N\mathcal{A} \right]$$

$$\leqslant \left[ 64 \prod_{i=1}^{L}[[\rho_i + 1]s_i + 1]^3 [b+1][L_l+1]\mathcal{W}\mathcal{A}^3 N^2 \right] \leqslant [4\underline{\Gamma}^{\mathfrak{C},\lambda_{aug}}]^3 \tag{G.18}$$

where $\underline{\Gamma}^{\mathfrak{C},\lambda_{aug}} := [b+1][L(L_l+1))\mathcal{W}\mathcal{A}N \prod_{i=1}^{L}[[\rho_i + 1]s_i + 1]$. Thus, we can continue from equation (G.15): $\log(|\mathcal{C}|) \leqslant$

$$216 \log(4\underline{\Gamma}^{\mathfrak{C},\lambda_{aug}}) \sum_{\ell=1}^{L} \left[ \frac{\mathcal{M}_\ell b_{\ell-1}}{\epsilon_\ell} \right]^{\frac{2p_\ell}{p_\ell+2}} [U_\ell + d_{\ell-1}]^{\frac{2}{p_\ell+2}} \min(U_\ell, d_{\ell-1})^{\frac{p_\ell}{p_\ell+2}} W_\ell^{\frac{p_\ell}{p_\ell+2}}$$

$$\leqslant 216 \log(4\underline{\Gamma}^{\mathfrak{C},\lambda_{aug}}) \sum_{\ell=1}^{L} \left[ \frac{\mathcal{M}_\ell b_{\ell-1} \rho_\ell L[1 + L_l] \prod_{i=\ell+1}^{L} s_i \rho_i}{\epsilon} \right]^{\frac{2p_\ell}{p_\ell+2}}$$

$$\times \qquad\qquad\qquad [U_\ell + d_{\ell-1}]^{\frac{2}{p_\ell+2}} \min(U_\ell, d_{\ell-1})^{\frac{p_\ell}{p_\ell+2}} W_\ell^{\frac{p_\ell}{p_\ell+2}}$$

$$\leqslant 216 \log(4\underline{\Gamma}^{\mathfrak{C},\lambda_{aug}}) \left[ \frac{L(1 + L_l)}{\min(\epsilon, 1)} \right]^{\frac{2p}{2+p}} \sum_{\ell=1}^{L} \left[ b_{\ell-1} \prod_{i=\ell}^{L} s_i \rho_i \right]^{\frac{2p_\ell}{p_\ell+2}}$$

$$\times \qquad\qquad\qquad [U_\ell + d_{\ell-1}]^{\frac{2}{p_\ell+2}} \min(U_\ell, d_{\ell-1})^{\frac{p_\ell}{p_\ell+2}} W_\ell^{\frac{p_\ell}{p_\ell+2}},$$

as expected.

$\square$

Next, similarly to the proof of Proposition G.4, we can obtain:

**Proposition G.9.** *Fix some reference/initialization matrices $M^1 \in \mathbb{R}^{w_1 \times w_0} = \mathbb{R}^{w_1 \times d}, \ldots, M^L \in \mathbb{R}^{w_L \times w_{L-1}} = \mathbb{R}^{C \times w_{L-1}}$. Assume also that the inputs $x$ satisfy $\|x\| \leqslant b$ w.p. 1. Fix a set of constraint parameters $0 \leqslant p_\ell \leqslant 2, s_\ell, \mathcal{M}_\ell$ (for $\ell = 1, \ldots, L$).*

*With probability greater than $1 - \delta$ over the draw of an i.i.d. training set $x_1, \ldots, x_N$, every neural network $F_A^{\mathfrak{C}}(x)$ defined with equation (G.1) satisfying the following conditions:*

$$\|A^\ell - M^\ell\|_{\mathrm{sc},p_\ell} \leqslant \mathcal{M}_\ell \qquad \forall 1 \leqslant \ell \leqslant L$$
$$\|\mathrm{op}(A)_\ell\| \leqslant s_\ell \qquad \forall 1 \leqslant \ell \leqslant L-1$$
$$\|A_L^\top\|_{2,\infty} \leqslant s_L, \tag{G.19}$$

*also satisfies the following generalization bound:*

$$\mathbb{E}\left[\mathrm{l}(F_A(x), y)\right] - \frac{1}{N}\sum_{i=1}^{N}\mathrm{l}(F_A(x_i), y_i) \tag{G.20}$$

$$\leqslant 6[\mathcal{B}+1]\sqrt{\frac{\log(\frac{4}{\delta})}{N}} + \frac{361\sqrt{\log_2(4\underline{\Gamma}^{\mathfrak{C},\lambda_{aug}})}\log(N)[\mathcal{B}+1]}{\sqrt{N}}(1 + L\,\mathrm{L_l})^{\frac{p}{2+p}}\mathcal{R}_{\mathcal{M},s,p,b}^{\mathfrak{C},\lambda_{aug}}$$

*where* $\mathcal{R}_{\mathcal{M},s,p,b}^{\mathfrak{C},\lambda_{aug}} :=$

$$\left[\sum_{\ell=1}^{L}\left[b_{\ell-1}\prod_{i=\ell}^{}\rho_i s_i\right]^{\frac{2p_\ell}{p_\ell+2}}\left[\frac{\mathcal{M}_\ell{}^{p_\ell}}{s_\ell{}^{p_\ell}}\right]^{\frac{2}{p_\ell+2}}\mathrm{r}_{\mathfrak{C},\ell}^{\frac{2}{p_\ell+2}}[U_\ell + d_{\ell-1}]^{\frac{2}{p_\ell+2}}\min(U_\ell, d_{\ell-1})^{\frac{p_\ell}{p_\ell+2}}W_\ell^{\frac{p_\ell}{p_\ell+2}}\right]^{\frac{1}{2}}$$

*with* $\mathrm{r}_{\mathfrak{C},\ell} := \frac{\mathcal{M}_\ell{}^{p_\ell}}{s_\ell{}^{p_\ell}}$ *and* $\underline{\Gamma}^{\mathfrak{C},\lambda_{aug}} := [b+1][L(\mathrm{L_l}+1))\mathcal{W}\mathcal{A}\mathcal{N}\prod_{i=1}^{L}[[\rho_i+1]s_i+1].$

*Proof.* The proof is the same as that of Proposition G.4 with an additional factor of $\sqrt{3}$, relying on Proposition G.8 instead of Proposition G.3. □

Finally, we can obtain the following post hoc version from the above.

**Theorem G.10.** *With probability greater than* $1 - \delta$*, for any* $b \in \mathbb{R}^+$*, every trained neural network* $F_A$ *satisfies the following generalization gap for all possible values of the* $p_\ell s$*:*

$$\mathbb{E}\left[\mathrm{l}(F_A(x), y)\right] - \frac{1}{N}\sum_{i=1}^{N}\mathrm{l}(F_A(x_i), y_i) \leqslant 6[\mathcal{B}+1]\sqrt{\frac{\log(1/\delta)}{N}} + \tag{G.21}$$

$$6\mathcal{B}\sqrt{\frac{2 + \Theta_{\log}^{\mathfrak{C},\lambda_{aug}}}{N}} + 722[\mathcal{B}+1][L(1 + \mathrm{L_l})]^{\frac{p}{2+p}}\frac{\sqrt{\log(\gamma_{\mathfrak{C},\lambda_{aug}})}\log(N)}{\sqrt{N}}\mathcal{R}_{F_A}^{\mathfrak{C},\lambda_{aug}},$$

*where*

$$\Theta_{\log}^{\mathfrak{C},\lambda_{aug}} := 2\log\left[4\mathcal{W}\mathcal{A} + \sup_{i=1}^{N}\|x_i\|\prod_i\rho_i\|\mathrm{op}(A_i)\|\right] \times$$

$$\left[|\log(\sup_i\|x_i\|)| + L\left[\sum_{\ell=1}^{L}|\log(\|\mathrm{op}(A_\ell)\|)| + \sum_{\ell=1}^{L-1}\log(|\mathfrak{B}_{\ell-1,A}|) + 2\log(4\mathcal{A}\mathcal{W})\right]\right],$$

$$\gamma_{\mathfrak{C},\lambda_{aug}} := 12\left[\left[[\mathfrak{B}+1][L\,\mathrm{L_l}+1]\mathcal{W}\mathcal{A}\mathcal{N}\prod_{i=1}^{L}[\rho_i\|\mathrm{op}(A_i)\|+1]\right]\right]\left[4\overline{W} + \mathfrak{B}\prod_i\rho_i\|\mathrm{op}(A_i)\|\right],$$

*and where*

$$\mathcal{R}_{F_A}^{\mathfrak{C},\lambda_{aug}} := \left[\sum_{\ell=1}^{L}\left[\sup_i\|x_i\|\|A_L^\top\|_{2,\infty}\mathfrak{B}_{\ell-1,A}\rho_\ell\prod_{i=\ell+1}^{L}\rho_i s_i\right]^{\frac{2p_\ell}{p_\ell+2}}\times \tag{G.22}$$

$$\left[\frac{\|A_\ell - M_\ell\|_{\mathrm{sc},p_\ell}^{p_\ell}}{\|\mathrm{op}(A_\ell)\|^{p_\ell}}\right]^{\frac{2}{p_\ell+2}}[U_\ell + d_{\ell-1}]^{\frac{2}{p_\ell+2}}\min(U_\ell, d_{\ell-1})^{\frac{p_\ell}{p_\ell+2}}W_\ell^{\frac{p_\ell}{p_\ell+2}}\right]^{\frac{1}{2}}.$$

*Proof.* The proof is the same as that of Theorem E.6 with the original constant of 416 replaced by 361 and $\|A_\ell\|$ replaced by $\|\mathrm{op}(A_\ell)\|$. □

# H  Generalization Bound for Linear Networks with Schatten Constraints

In this section, we briefly mention that it is possible to prove alternative bounds for linear networks assuming the presence of Schatten quasi norm constraints on the factor matrices (instead of $L^2$ constraints). Indeed, a generalization of Theorem 3.1 (Theorem I.2) allows also one to translate such constraints into a single Schatten quasi norm constraint on the linear map $A$.

**Theorem H.1.** *For any $0 \leqslant p_\ell \leqslant 1$, write $p$ for the conjugate such that $\frac{1}{p} = \sum_{\ell=1}^{L} \frac{1}{p_\ell}$. W.p. $\geqslant 1 - \delta$ over the draw of the training set, every linear network as defined in equation (3.5) satisfies the following generalization bound:*

$$\text{GAP} - O\left(\mathcal{B}\sqrt{\frac{\log(1/\delta)}{N}}\right) \leqslant \tilde{O}\left(\sqrt{[\text{L}_l\,\mathfrak{B}]^{\frac{2p}{2+p}}\frac{\left[\sum_{\ell=1}^{L}\frac{p\|B_\ell\|_{\text{sc},p_\ell}^{p_\ell}}{p_\ell}\right]^{\frac{2}{2+p}}[\mathcal{C}+d]}{N}}\right) \tag{H.1}$$

*where $\mathfrak{B} = \sup_{i=1}^{N} \|x_i\|$, the $\tilde{O}$ notation hides polylogarithmic factors.*

# I  Known Results

This section collects some well-known results which are key to our proof, and which we include for the benefit of self-completeness.

**Theorem I.1** ([24], Theorem 1). *Let $w \geqslant \min(\mathcal{C}, d)$ and let $B_1, \ldots, B_L$ be matrices such that $B_1 \in \mathbb{R}^{w \times d}, B_L \in \mathbb{R}^{\mathcal{C} \times w}, B_2 \ldots, B_{L-1} \in \mathbb{R}^{w \times w}$. For any matrix $A \in \mathbb{R}^{w \times d}$ we have*

$$L\|A\|_{\text{sc},\frac{2}{L}}^{\frac{2}{L}} = \min \sum_{\ell=1}^{L} \|B_\ell\|^2 \quad \text{subject to } B_L B_{L-1} \ldots B_1 = A. \tag{I.1}$$

**Theorem I.2** ([92, 93, 94]). *Let $w \geqslant \min(\mathcal{C}, d)$ and let $B_1, \ldots, B_L$ be matrices such that $B_1 \in \mathbb{R}^{w \times d}, B_L \in \mathbb{R}^{\mathcal{C} \times w}, B_2 \ldots, B_{L-1} \in \mathbb{R}^{w \times w}$. Fix $p_1, \ldots, p_L$ such that $\frac{1}{p} = \sum \frac{1}{p_i}$.*

*For any matrix $A \in \mathbb{R}^{w \times d}$ we have*

$$\|A\|_{\text{sc},p}^{p} = \min \sum_{\ell=1}^{L}\left[\frac{p\|B_\ell\|_{\text{sc},p_\ell}^{p_\ell}}{p_\ell}\right] \quad \text{subject to } B_L B_{L-1} \ldots B_1 = A. \tag{I.2}$$

**Proposition I.3** (Cf. Proposition 5 in [16] (with $U = 1$), see also [65]). *Let $x_1, \ldots, x_N \in \mathbb{R}^d$ be such that $\|x_i\| \leqslant b$ for all $i \leqslant N$. For any fixed choice of reference matrix $M \in \mathbb{R}^{m \times d}$, consider the set $\text{F}_\text{M}$ of matrices $A \in \mathbb{R}^{m \times d}$ such that $\|A - M\|_{\text{Fr}} \leqslant a$. For any $\epsilon > 0$ there exists a cover $\mathcal{C} \subset S_M$ such that for all $A \in \text{F}_\text{M}$, there exists $\bar{A} \in \mathcal{C}$ such that for all $i \leqslant N$, $\|(A - M)x_i\|_\infty \leqslant \epsilon$ and*

$$\log(|\mathcal{C}|) \leqslant \frac{36a^2b^2}{\epsilon^2}\log_2\left[\left(\frac{8ab}{\epsilon} + 7\right)mN\right]. \tag{I.3}$$

**Lemma I.4** (Lemma 8 in [4]). *The (internal) covering number $\mathcal{N}$ of the ball of radius $\kappa$ in dimension $d$ (with respect to any norm $\|\cdot\|$) can be bounded by:*

$$\mathcal{N} \leqslant \left\lceil\frac{3\kappa}{\epsilon}\right\rceil^d \leqslant \left(\frac{3\kappa}{\epsilon} + 1\right)^d \tag{I.4}$$

Recall also the following lemma from [83]:

**Proposition I.5.** *Let $n, d \in \mathbb{N}$, $a, b > 0$. Suppose we are given $n$ data points collected as the rows of a matrix $X \in \mathbb{R}^{n \times d}$, with $\|X_{i,\cdot}\|_2 \leqslant b, \forall i = 1, \ldots, n$. For $U_{a,b}(X) = \{X\alpha : \|\alpha\|_2 \leqslant a, \alpha \in \mathbb{R}^d\}$, we have*

$$\log \mathcal{N}(U_{a,b}(X), \epsilon, \|\cdot\|_\infty) \leqslant \frac{36a^2b^2}{\epsilon^2}\log_2\left(\frac{8abn}{\epsilon} + 6n + 1\right).$$

**Theorem I.6** (Generalization bound from Rademacher complexity, cf. e.g., [95], [96], [97] etc.)**.** *Let $Z, Z_1, \ldots, Z_N$ be i.i.d. random variables taking values in a set $\mathcal{Z}$. Consider a set of functions $\mathcal{F} \in [0, 1]^{\mathcal{Z}}$. $\forall \delta > 0$, we have with probability $\geqslant 1 - \delta$ over the draw of the sample $S$ that*

$$\forall f \in \mathcal{F}, \quad \mathbb{E}(f(Z)) \leqslant \frac{1}{N} \sum_{i=1}^{N} f(z_i) + 2\,\mathfrak{R}_S(\mathcal{F}) + 3\sqrt{\frac{\log(4/\delta)}{2N}},$$

*where $\mathfrak{R}_S(\mathcal{F})$ can be either the empirical or expected Rademacher complexity. In particular, if $f^* \in \arg\min_{f \in \mathcal{F}} \mathbb{E}(f(Z))$ and $\hat{f} \in \arg\min_{f \in \mathcal{F}} \frac{1}{N} \sum_{i=1}^{N} f(z_i)$, then*

$$\mathbb{E}(\hat{f}(Z)) \leqslant \mathbb{E}(f^*(Z)) + 4\,\mathfrak{R}_S(\mathcal{F}) + 6\sqrt{\frac{\log(2/\delta)}{2N}}.$$

**Lemma I.7** (Dudley's entropy theorem [98], this version from [16, 99], see also [1, 74, 4])**.** *Let $\mathcal{F}$ be a real-valued function class taking values in $[0, \mathcal{B}]$. Let $S$ be a finite sample of size $N$. We have the following relationship between the Rademacher complexity $\mathfrak{R}(\mathcal{F}|_S)$ and the $L^2$ covering number $\mathcal{N}(\mathcal{F}|S, \epsilon, \|\cdot\|_2)$.*

$$\mathfrak{R}(\mathcal{F}|_S) \leqslant \inf_{\alpha > 0} \left( 4\alpha + \frac{12}{\sqrt{N}} \int_{\alpha}^{\mathcal{B}} \sqrt{\log \mathcal{N}(\mathcal{F}|S, \epsilon, \|\cdot\|_2)} d\epsilon \right),$$

*where the norm $\|\cdot\|_2$ on $\mathbb{R}^N$ is defined by $\|x\|_2^2 = \frac{1}{N}(\sum_{i=1}^{N} |x_i|^2)$.*

