# OpenReview forum: "Generalization Bounds for Rank-sparse Neural Networks"
_NeurIPS.cc/2025/Conference — NeurIPS 2025 poster_

### Official Review · Reviewer_wqbT · 2025-06-24

**Clarity:** 4
**Significance:** 3
**Originality:** 3
**Rating:** 4
**Confidence:** 4

**Summary:**

The paper proves a novel generalization bound for fully-connected and convolutional neural networks. This bound features a free parameter $p_l$ for each layer $l$ which could be chosen a-posteriori (that is, the bound is uniform wrt these parameters). By choosing these parameters, one could interpolate between a norm based bound similar to [1] to a classical parameter-counting bound, individually for each layer. To be specific, the bound is based on $p$-Schatten norms of layer weights, which interpolate between a Frobenius norm ($p=2$) and a usual matrix rank ($p=0$). Being able to interpolate gives flexibility that allows to cover the case of approximately low-rank layer weights conveniently.

The paper provides a thorough overview on existing norm-based and parameter-counting generalization bounds, as well as on a low-rank bias of neural nets. There are also experiments conducted for simple multi-layer fully-connected and convolutional networks trained on MNIST and CIFAR dataset.

[1] Behnam Neyshabur, Srinadh Bhojanapalli, and Nathan Srebro. A PAC-bayesian approach to spectrally-normalized margin bounds for neural networks. In International Conference on Learning Representations

**Questions:**

1. Am I right that Line 110 does not coincide with (2) when all $p_l=2$? In other words, does your Theorem 3.2 recover the main result of [Neyshabur et al. 2018] when all $p_l=2$? If no, why?
2. Lines 90 — 94: which "deep results" specifically are you referring to?

**Ethical Concerns:**

["NO or VERY MINOR ethics concerns only"]

**Final Justification:**

I thank the authors for responding thoroughly to my questions. I am convinced that I understood the paper fairly during the initial evaluation and therefore, I keep my score.

**Limitations:**

If the obtained bounds are numerically vacuous, I would mention it as a limitation.

**Quality:**

3

**Strengths And Weaknesses:**

**Strengths**

1. The approach taken in the paper generalizes two large classes of generalization bounds (norm-based and parameter counting) in a natural and neat way.
2. The literature overview is rather thorough.
3. The paper spends a lot of time comparing its main result and approach to relevant works, thus showing clearly where this work stands among other works.
4. The main theorems are stated in a clean and easy-to-absorb way.

**Weaknesses**

1. Rank sparsity was specifically enforced in experiments for fully-connected networks.  To me, this is not really fair because this looks like forging a setup that favors the bound of the paper but not the bounds the paper compares to.
2. The architecture used for experiments on convolutional nets looks strange: 3 conv layers followed by 5 fully-connected layers. What was the reason to use this structure? This looks neither like a standard setup to compare with other works, nor like a simple well-performing ConvNet.
3. The comparison ignores logarithmic factors, and, probably, constants; are the exact numbers provided in Appendix D informative then? Maybe, an asymptotic for large depth or width could be more informative.
4. The paper does not disclose any exact values of the bounds they prove. This makes me suspect these bounds to be vacuous in more-or-less realistic setups when evaluated up to a number. At the same time, non-vacuous bounds do exist (see e.g. [2,3]), but the paper never mentions them.
5. One relevant citation seems to be missing: [1] studies low rank biases for linear networks induced by $l_2$-regularization.

**Minor issues**

1. Not clear from this point what is L_1 in (1) (it is introduced later). If it is a Lipschitz constant, why "we assume that the nonlinearities are 1-Lipschitz when discussing related works"?
2. Line 111: there is no $p$ in the above formula.
3. Line 110: is the factor $L^{p/(2+p)}$ forgotten here?
4. Not clear from this point what is $p_l$ in the formula on line 110 (before the main result is formally introduced).
5.  Line 78: "expreimntally"
6. Line 285: the argument $x_i$ seems to be at the wrong place.

[1] Jacot, A., Ged, F., Gabriel, F., Simsek, B., and Hongler, C. (2021). Deep linear networks dynamics: Low-rank biases induced by initialization scale and l2 regularization. https://arxiv.org/abs/2106.15933v1

[2] Zhou, Wenda, et al. "Non-vacuous generalization bounds at the imagenet scale: a PAC-bayesian compression approach."

[3] Biggs, Felix, and Benjamin Guedj. "Non-vacuous generalisation bounds for shallow neural networks." International Conference on Machine Learning. PMLR, 2022.

---

> ### Author Rebuttal · Authors · 2025-07-31
>
> Many thanks for your positive review and your statement that our paper **clearly where this work stands among other works**, and that our theorems are **clean and easy to absorb**. We try our best to allay all your concerns below and stand ready to provide further clarifications upon request.
>
> Re weaknesses:
>
> 1.  Many thanks for your comment. We agree that this additional regularization in the fully-connected experiments favors our bound and somewhat departs from our general approach of minimizing optimization tricks. We believe that both our bounds would still outperform the baselines if we replaced this trick by spectral constraints, and that our sharper result (including loss augmentation) would still outperform the baselines with neither trick involved. We promise to incorporate a discussion of some additional such experiments in the camera ready version if accepted.
>
> 2. The architecture was chosen to be large enough to involve the rank-sparsity inducing effects of depth which we study, but also small enough to allow us to fun experiments for a wide range of widths without undue computational overload.
>
> 3. Yes, our numerical comparison ignores logarithmic factors and constants. This is in line with much of the recent literature. Therefore, the main message of our experimental section is that our bounds are **less sensitive to increases** in width, since they can **capture the spontaneous reduction** in function class capacity arising from low-rank representations.
> We would also like to point out that it is very difficult, if not impossible, to compare all bounds in the literature without ignoring logarithmic factors and constants. Indeed, many works (e.g. [2]) skip constants entirely, and **many works (e.g. [2,3]) do not explicitly write down the proof of the post hoc step,**  leaving it as an exercise for the reader. This makes a comparison with our theorem E.2. unfair if one includes the logarithmic factors arising from the numerous post hoc steps over $s\_{\ell}, p\_{\ell}, \mathcal{M}\_{\ell},$ etc.
>
>
> 4. Yes, the bounds are vacuous at realistic scales if one incorporates all the logarithmic factors (though as explained above, this is a potentially questionable point). Our aim was to limit ourselves to uniform convergence bounds, excluding many of the tricks used to optimize bounds. For instance, to the best of our understanding,  [9] rely on a **compression**of the learned network, similarly, [10] assumes that the weights of the first and second layer are optimized on distinct subsets of the training set. We interpret these facts as warranting the classification of those results as belonging to a slightly different branch of the literature. However, we really **enjoyed reading both papers [9,10]** suggested  (especially [10]) and **promise to incorporate a further discussion of them** in the camera ready version if accepted.
>
> 5. Many thanks for the suggestion. Although we have already included other references by the same authors which study the same phenomenon [13,14,15], we have missed this one, which we promise to discuss in the camera ready revision.
>
>
> Re **question 1**:
>
> Yes, you are right, and this is a very sharp observation. This is due to the Markov inequality argument in the key proof of Proposition F2, which we also recommend you check in the proof sketch we added in the response to reviewer 9AFC. The key reason is that the parameter counting component of the function class involves a factor of $[m+d]r$ and even the norm-based component involves a factor of $\min(m,d)\tau^2$, resulting in an unavoidable dependence on $[m+d]$ (or at least $\min(m,d)$).
>
> To the best of our knowledge, it is difficult to smoothly remove this limitation except by trivially  imposing a minimum between both bounds. This underscores the substantial differences between our application of parametric interpolation in the neural networks context as opposed to its original introduction in the context of matrix completion in [1]. However, we want to emphasize that this doesn't make our results uniformly inferior to either norm-based or parameter counting bounds.
>  Indeed, counting parameters directly when estimating the capacity of the first layer  of linear maps $A^1\in\mathbb{R}^{m\times d}$ would yield a sample complexity of at least $md$ instead of $[m+d]$, and the norm-based bound would be very large if the norms of the weights are large, though our own results involve norm based quantities only at a reduced and tunable power of $\frac{2p}{p+2}$.  Furthermore, if all the weights are binary (making norm based and parameter counting approaches equal), then our result will coincide with both, indicating that there is **no "pure loss"**: it can merely be said that although our bound interpolates between $p=0$ and $p=2$, a casualty of the interpolation is that **even setting $p=2$** (which corresponds to the norm-based bound), the bound **still leans slightly towards parameter counting despite incorporating strong norm-based aspects**.
> Indeed, consider the one layer case in n idealised situation where each weight belongs to the set $\{-1,1\}$. In this case, the norm based bound gives a sample complexity of $\|A\|\_{Fr}^2=md$, which recovers the same bound as a pure parameter counting bound. **In this case**, our \textit{sample complexity} bound for $p=2$ and $B=1$ scales as (cf Theorem 3.1)  $\mathcal{M}^{\frac{2p}{p+2}} \min(m,d)^{\frac{p}{p+2}} [m+d]^{\frac{2}{p+2}}=\mathcal{M} [m+d]=O(\|A\|\_{Fr}\sqrt{\min(m,d) \max(m,d)} )=O(md)$, **which also coincides with both norm based and parameter counting bounds**. However, as explained in the main paper, our bound is also sensitive to the rank, approximate rank, etc.
>
> Re Question 2: we meant to refer to the results in [12,13,14,15,16], and promise to clarify this further in the camera ready revision.
>
> Re Limitations: many thanks, we will incorporate a full discussion as above in the revision.
> Re minor issues: many thanks for your help in improving the clarity of our manuscript. Regarding points 2 and 3, the confusion arises from the fact that we treat the Lipschitz constants as absolute constants in the informal discussion in the introduction, removing the need for the factor in question.
>
> Once again, thank you for your thorough comments. We hope that our rebuttal has helped further convince you of the value of our work and that you will continue to engage with us in the discussion phase.
>
>
>
> =================================**References**===========================
>
>
> [1] Ledent, A.; Alves, R. Generalization Analysis of Deep Non-linear Matrix Completion. ICML 2024.
>
> [2] Long, P. M.; Sedghi, H. Generalization Bounds for Deep Convolutional Neural Networks. ICLR 2020.
>
> [3] Graf, F.; Zeng, S.; Rieck, B.; Niethammer, M.; Kwitt, R. On Measuring Excess Capacity in Neural Networks. NeurIPS 2022.
>
> [4] Zhang, T. Covering Number Bounds of Certain Regularized Linear Function Classes. JMLR 2002.
>
> [5] Bartlett, P. L.; Foster, D. J.; Telgarsky, M. Spectrally-Normalized Margin Bounds for Neural Networks. NeurIPS 2017.
>
> [6] Suzanna Parkinson, Greg Ongie, Rebecca Willett. ReLU Neural Networks with Linear Layers are Biased Towards Single- and Multi-Index Models
>
>
> [7] Gintare Karolina Dziugaite, Kyle Hsu, Waseem Gharbieh, Gabriel Arpino, Daniel M. Roy. On the role of data in PAC-Bayes bounds
>
> [8] Arthur Jacot, François Ged, Berfin Şimşek, Clément Hongler, Franck Gabriel. Saddle-to-Saddle Dynamics in Deep Linear Networks: Small Initialization Training, Symmetry, and Sparsity
>
> Earlier version: Jacot, A., Ged, F., Gabriel, F., Simsek, B., and Hongler, C. (2021). Deep linear networks dynamics: Low-rank biases induced by initialization scale and l2 regularization.
>
> [9] Zhou, Wenda, et al. "Non-vacuous generalization bounds at the imagenet scale: a PAC-bayesian compression approach."
>
> [10] Biggs, Felix, and Benjamin Guedj. "Non-vacuous generalisation bounds for shallow neural networks." International Conference on Machine Learning. PMLR, 2022.
>
> [11]  Neyshabur, Bhojanapalli, Srebro.  A PAC-Bayesian Approach to Spectrally-Normalized Margin Bounds for Neural Networks. ICLR 2018.
>
> [12] Arthur Jacot, François Ged, Berfin Şimşek, Clément Hongler, Franck Gabriel. Saddle-to-Saddle Dynamics in Deep Linear Networks: Small Initialization Training, Symmetry, and Sparsity
>
> Earlier version: Jacot, A., Ged, F., Gabriel, F., Simsek, B., and Hongler, C. (2021). Deep linear networks dynamics: Low-rank biases induced by initialization scale and l2 regularization.
>
>
> [13]  Bottleneck structure in learned features: Low-dimension vs regularity tradeoff. NeurIPS 2023.
>
> [14] Implicit bias of large depth networks: a notion of rank for nonlinear functions.
>
>
> [15] Zihan Wang and Arthur Jacot. Implicit bias of SGD in $l_2$-regularized linear DNNs: One-way jumps from high to low rank. ICLR 2024.
>
> [16] Laura Balzano, Tianjiao Ding, Benjamin D. Haeffele, Soo Min Kwon, Qing Qu, Peng Wang, Zhangyang Wang, Can Yaras. An Overview of Low-Rank Structures in the Training and Adaptation of Large Models. ArXiv 2025

---

> > ### Comment · Reviewer_wqbT · 2025-08-04
> >
> > Many thanks to the authors for their thorough clarifications!

---

### Official Review · Reviewer_9AFC · 2025-07-02

**Clarity:** 2
**Significance:** 3
**Originality:** 3
**Rating:** 4
**Confidence:** 3

**Summary:**

This paper provides generalization bounds via covering numbers for neural networks with low-rank (or approximately low-rank) weight matrices.

**Questions:**

Can you provide a proof sketch of a simplified form of Theorems E.2? I am not comfortable accepting this paper without a readable proof sketch to fascilitate verification of the validity of the proof, and I think a good proof sketch will make this paper much more useful to the community. Aim for as much simplicity in the sketch as possible; e.g. assume that $L_1 = B = \rho_i = 1, L=2$, etc., and use $\lesssim$ liberally instead of explicitly showing constants, etc.. If the sketch is clear and verifiable, I will increase my score to a 5.

You mention that you believe that the dependency on input dimension is likely removable. Do you believe that the dependency on the output dimension and/or hidden width is also removable?

**Ethical Concerns:**

["NO or VERY MINOR ethics concerns only"]

**Final Justification:**

I believe this work represents a significant result, and including a proof sketch and addressing other presentation issues we have discussed will improve the paper. While the proof sketch provided during the discussion period was morally correct and the authors have since corrected several typos, the presence of typos in the proof sketch gives me some trepidation that there may be other unidentified issues in the full proofs. However, I believe reasons to accept outweigh reasons to reject, so I raise my score to a 4.

**Limitations:**

Yes

**Quality:**

3

**Strengths And Weaknesses:**

Generalization bounds based on low-rank weight matrices have the potential to be much tighter than existing bounds. It has been observed empirically and theoretically that depth induces an inductive bias towards low-rank weight matrices, this result is a great first step towards tightening generalization bounds in this setting.

However, the paper is currently poorly presented. Technical notation is used in the introduction long before it is defined in section 3. I was unable to verify technical details because of the very, very long proof without a sketch. A readable sketch that gives more insight into the techniques would be helpful.

Additionally, the final bound has poor dependency on the input space dimension. At least this work is *upfront* about the fact that dependency on input dimension is a major limitation.

Other minor comments/suggestions to improve clarity:
- Line 26: "DNNs *on* natural data"
- Line 38: the parenthetical (for simplicity...) is better as a footnote
- Eq 1,2,3 and Lines 104, 110, 114: *Lots* of undefined notation here, including $p_\ell, L_1$, etc. Perhaps simplify the presentation by assuming some terms are constants here-- $L_1, B, \rho$, etc.
- Line 56: make the norm *small*
- Line 75: Should also cite Parkinson et al. 2023 (arXiv:2305.15598)
- Line 83: Typo: $A \in \mathbb R^{C \times d}$ not $A \in \mathbb R^{w \times d}$
- Line 91: These results are more about the effect of adding a $L_2$ regularization term to the loss than about optimization itself.
- Line 115: While later you specify that it's best to choose $M_\ell$ to be zero in which case talking about the low rank structure in the term makes sense, it's confusing that $M_\ell$ is dropped here. If $M_\ell = 0$ is the best choice, why include $M_\ell$ in line 110 at all?
- Line 237: The first sentence seems to have a word missing?
- Line 250-251: This doesn't make sense as written. I think there's a typo in equation it's comparing to?
- Line 285: is this supposed to be the notation from Line 266?

---

> ### Author Rebuttal · Authors · 2025-07-31
>
> Many thanks for your detailed comments and interest in our work and promise to increase your score to 5 pending a satisfactory proof sketch of our main theorem. Due to the character limit, we include only the sketch here. **Please note the answers to your other questions are provided in the response to reviewer Udna**.
>
> **Simplified Prop F2 (covering number for one layer)**: Consider the following function class of linear maps from $\mathbb{R}^d$ to $ \mathbb{R}^m$: $\mathcal{F}^p:=\{M \in\mathbb{R}^{m\times d}, \|M\|\_p\leq \mathcal{M}; \|M\|\leq s\}$. For any $\epsilon>0$ and for any dataset $x_1,\ldots,x_N\in\mathbb{R}^d$ such that $\|x_i\|\leq B$ for all $i$, we have the following bound on the $L^\infty$ covering number (cf Prop F2 for formal defn) of the class:
>
> $$\log(\mathcal{N}_{\infty}(\mathcal{F}_r^p,\epsilon))\lesssim [m+d]\Bigg[\frac{\mathcal{M}B}{\epsilon}\Bigg]^{\frac{2p}{p+2}}\log(\frac{mdN\mathcal{M}}{\epsilon}).$$
>
> **Proof sketch (for simplicity we assume $B=1$ in the sketch):**
>
> For any $M\in\mathcal{F}^p$, write $\sum_{i=1}^{\min(m,d)} \rho_i u_iv_i^\top$ for its singular value decomposition. For a threshold $\tau>0$ to be determined later, we decompose every $M\in\mathcal{F}^p$ into $M=M_1+M_2$ where $M_1=\sum_\{i\leq T\} \rho_i u_iv_i^\top$ and $M_2=\sum_{i\geq T+1}\rho_i u_iv_i^\top$ where $T$ is the last index such that $\rho_{T}> \tau$.
>
> By Markov's inequality, since $\|M\|\_p^p=\sum_i \rho_i^p \leq \mathcal{M}^p $, we have $\text{rank}(M_1)\leq \tau\leq \frac{\mathcal{M}^p}{\tau^p}$. In addition, the spectral norm of $M_1$ is bounded as: $\|M\_1\|=\rho_1=\|M\|\leq \|M\|\_{p}\leq \mathcal{M}$. Thus, $M_1$ belongs to the set ${\mathcal{F}^p}\_1:=\Big\{ Z\in\mathcal{F}^p:\text{rank}(Z)\leq  \frac{\mathcal{M}^p}{\tau^p}, \quad \|Z\|\leq \mathcal{M}\Big\} $ (a function class with few parameters its members are **low-rank**)
> 	Furthermore, it is clear that $$
> 		\|M_2\|\_{FR}^2=\sum_{k=T+1}^{\min(m,d)} \rho_k^{2} \leq \tau^2 {\min(m,d)}.$$
> Therefore $M_2$ belongs to the class $\mathcal{F}^p\_2:=\{Z\in\mathcal{F}^p: \|Z\|\_{\text{FR}}^2\leq \tau^2 {\min(m,d)}\} $ (a function class whose members have **small norms**).
>
> By classic *parameter counting* arguments (cf. [2,3] etc., the specific result being Lemma D1 page 30 in [1]), we have the following bound on the $L^\infty$ covering number of $\mathcal{F}^p_1$:
> $$\mathcal{N}_\infty(\mathcal{F}^p_1,\epsilon/2)\lesssim [m+d][\text{rank}] \log(\frac{\mathcal{M}}{\epsilon})\lesssim [m+d]\frac{\mathcal{M}^p}{\tau^p}\log(\frac{\mathcal{M}}{\epsilon}).$$
>
> By classic *norm-based arguments* (Theorem 4 in [4]), we can bound the covering number of the class $\mathcal{F}^p_2$ as follows:
> $$\log(\mathcal{N}(\mathcal{F}^p_2,\epsilon/2))\lesssim [\text{max Frob norm}] \log(\frac{\tau^2 mdN}{\epsilon}) =\frac{\tau^2 {\min(m,d)}}{\epsilon^2} \log(\frac{\tau^2 mdN}{\epsilon}).$$
>
> Combining both covering number bounds gives
> $$\log(\mathcal{N}_{\infty}(\mathcal{F}_r^p))\leq \log(\mathcal{N}(\mathcal{F}^p_1,\epsilon/2))+\log(\mathcal{N}(\mathcal{F}^p_2,\epsilon/2))\lesssim     \frac{\tau^2 {\min(m,d)}}{\epsilon^2} \log(\frac{\tau^2 mdN}{\epsilon})+[m+d]\frac{\mathcal{M}^p}{\tau^p}\log(\frac{\mathcal{M}}{\epsilon})$$
>
> Setting $\tau=\mathcal{M}^{\frac{p}{p+2}} \epsilon^{\frac{2}{p+2}}$ gives
> $\log(\mathcal{N}_{\infty}(\mathcal{F}_r^p))\lesssim $ $[m+d]\left[\frac{\mathcal{M}}{\epsilon}\right]^{\frac{2p}{p+2}}\log(\frac{mdN\mathcal{M}}{\epsilon})$
> as expected.
>
>
>
> The above result forms the basis of the one-layer case. As requested, we now consider the two-layer case:
>
> **Simplified form of Proposition E.1.:** Consider neural networks of the form $F_{A}:x\rightarrow A^2 \sigma(A^1 x)$ where $\sigma$ is an elementwise 1-Lipschitz activation function (e.g. ReLU) and $A_1\in \mathbb{R}^{m\times d}, A^2\in\mathbb{R}^{C\times m}$ are weight matrices. For fixed $p_1=p_2=p\in[0,2]$ and $s_1,s_2,\mathcal{M}_1,\mathcal{M}_2>0$, let $\mathcal{F}$ denote the class of such networks which further satisfy for all $i\in\{1,2\}$: $\|A^i\|\leq s_i; \> \|A^i\|_p\leq \mathcal{M}_i$.  By abuse of notation, we also use $\mathcal{F}$ to refer to the set of matrices $(A^1,A^2)$ satisfying the above requirements.
>
> Let $\ell: \mathbb{R}^C\times [C]\rightarrow  \mathbb{R}^+$  be a 1-Lipschitz loss function w.r.t. the $L^\infty$ norm (e.g. a margin-based loss function as in [2,3,5] with margin 1) and assume we are given an i.i.d. training set $(x_1,y_1),\ldots, (x_N,y_N)$ from a joint distribution over $\mathbb{R}^d\times [C]$ such that $\|x_i\|\leq B$ w.p. 1.
>
> For any $\delta>0$, w.p.  $\geq 1-\delta$, any network $F_A\in\mathcal{F}$ satisfies the following generalization bound:
> $$\mathbb{E}\ell(F(x,y))-\hat{\mathbb{E}}(\ell(F(x,y)))$$ $$\leq  \tilde{O}\bigg([s_1s_2]^{\frac{p}{2+p}} \bigg[\big[\frac{\mathcal{M}_1^{p}}{s_1^{p}}\big]^{\frac{2}{p+2}} [m+d]^{1+\frac{p}{p+2}} + \big[\frac{\mathcal{M}_2^{p}}{s_2^{p}}\big]^{\frac{2}{p+2}} [m+C]\bigg]^{\frac{1}{2}} \frac{1}{\sqrt{N}} +\sqrt{\frac{\log(1/\delta)}{N}}\bigg).$$ $$ < \tilde{O}\Bigg([s_1s_2]^{\frac{p}{2+p}} \bigg[\big[r_1[m+d]^{1+\frac{p}{p+2}} + r_2 [m+C]\bigg]^{\frac{1}{2}} \frac{1}{\sqrt{N}} +\sqrt{\frac{\log(1/\delta)}{N}} \Bigg),$$
>
> where for $i=1,2$, $r_i:=\big[\frac{\mathcal{M}_i^{p}}{s_i^{p}}\big]^{\frac{2}{p+2}}  $ is a **soft analogue of the rank**.
>
>
>
> **Remarks:** We assume the margin parameter is 1 because you specifically requested that the lipschitz constant be ignored. However, our full results include a fine analysis of the dependency on the margin parameter $\gamma\simeq 1/L_1$. The $\tilde{O}$ notation incorporates logarithmic factors of $N,m,d,C,1/\epsilon, s_1,s_2,\mathcal{M}_1,\mathcal{M}_2$.
>
> **Proof sketch:**
>
> **Step 1 (key)**: bounding the $L^{\infty}$ covering number of $\mathcal{F}$. Since the loss function is 1 Lipschitz, we only need to find an $L^\infty$ cover of $\mathcal{F}$, i.e., a set $\mathcal{C}\subset \mathcal{F}$ such that for any $A=(A^1,A^2)\in\mathcal{F}$, there exists a cover element $(\bar{A}^1,\bar{A}^2)\in \mathcal{C}$ such that $$\forall i, \|F_{A}(x_i)-F_{\bar{A}}(x_i)\|\_\infty\leq \epsilon$$ (here, recall $F_{A}(x_i)\in\mathbb{R}^C$ is a vector of scores for all classes). We achieve this by adapting standard chaining techniques (see [3,5]), with the caveat that we need to change the norm of the cover at the intermediary layer, incurring a factor of $\sqrt{m}$. Let $\mathcal{F}^1,\mathcal{F}^2$ denote the set of matrices $A^1\in\mathcal{R}^{m\times d}$ and $A^2\in\mathcal{R}^{C\times m}$ satisfying the relevant constraints above. First, we can apply Proposition F2 above, with $\epsilon$ set to $\epsilon_1:=\epsilon/[2\sqrt{m}s_2]$, to achieve a $\mathcal{C}_1\subset\mathcal{F}_1$ such that for all $A\in\mathcal{F}$ there exists a $\bar{A}\in\mathcal{C}$ such that $\forall i$,
> $\|\sigma(A^1(x_i))-\sigma(\bar{A}^1(x_i))\|\leq \|A^1x_i-\bar{A}^1x_i\|\leq \sqrt{m} \|\leq \|A^1x_i-\bar{A}^1x_i\| \leq \sqrt{m}\epsilon_1 =\frac{\epsilon}{2s_2}.$ and $\log(|\mathcal{C}|)\in \tilde{O}\Bigg( [m+d] \Big[\frac{\mathcal{M}_1 B }{\epsilon_1}\Big]^{\frac{2p}{p+2}} \Bigg) =\tilde{O}\Bigg( [m+d]  m^{\frac{p}{p+2}} [s_1 s_2B]^{\frac{2p}{p+2}}\Big[\frac{\mathcal{M}_1   }{s_2\epsilon}\Big]^{\frac{2p}{p+2}}\Bigg)$.
>
> Similarly, for any $\bar{A}^1\in\mathcal{C}_1$, every $i\leq N$ satisfies $\|\sigma(\bar{A}^1x_1)\|\leq s_1 B $, and therefore another application of Proposition F2 above with $B$ replaced by $\tilde{B}:=s_1 B$ and $\epsilon$ replaced by $\epsilon_2:=\epsilon/2$ yields a cover $\mathcal{C}_2(\bar{A}^1)$ with size $\log(|\mathcal{C}_2(\bar{A}^1)|)\in \tilde{O}\Bigg( \Big[\frac{\mathcal{M}_2 [Bs_1] }{\epsilon_2}\Big]^{\frac{2p}{p+2}} \Bigg)=\tilde{O}\Bigg([Bs_1s_2]^{\frac{2p}{p+2}} \Big[\frac{\mathcal{M}_2  }{\epsilon s_2}\Big]^{\frac{2p}{p+2}}   \Bigg)$.
>
> The cover $\mathcal{C}:=U\_{\bar{A}^1\in\mathcal{C}_1} \mathcal{C}_2(\bar{A}^1)$ is now an $\epsilon$ cover of $\mathcal{F}$.
>
> Indeed, for any $(A^1,A^2)\in\mathcal{F}$, we can define $\bar{A}^1$ to be the cover element associated to $A^1$ in $\mathcal{C}_1$ and subsequently $\bar{A}^2$ to be the cover element in $\mathcal{C}\_1(\bar{A}^1)$ associated to $A^2$, which yields for any $i\leq N$: $$\|A^2\sigma(A^1x_i)-\bar{A}^2\sigma(\bar{A}^1x_i)\|\_{\infty}\leq \|A^2\sigma(A^1x_i)-A^2\sigma(\bar{A}^1x_i)\|\_{\infty}+\|A^2\sigma(\bar{A}^1x_i)-\bar{A}^2\sigma(\bar{A}^1x_i)\|\_{\infty}$$ $$ \leq \epsilon_2+\|A^2\|  \|\sigma(A^1(x_i))-\sigma(\bar{A}^1(x_i))\|\leq \epsilon/2+s_2\epsilon/[2s_2] =\epsilon.$$
> Furthermore, the resulting cover has cardinality bounded as $$\log(|\mathcal{C}|)\in \tilde{O}\Bigg( \Big[\frac{Bs_1s_2}{\epsilon}\Big]^{\frac{2p}{p+2}} \Big[[m+d]^{1+\frac{p}{p+2}} \Big[\frac{\mathcal{M}_1  }{s_1}\Big]^{\frac{2p}{p+2}} +[m+C] \Big[\frac{\mathcal{M}_2  }{s_2}\Big]^{\frac{2p}{p+2}}  \Big]\Bigg).$$
>
> **Step 2:** Apply Dudley's entropy integral to obtain Rademacher complexity and apply classic results. This step  (cf. Proof of Prop E1) involves more straightforward calculations.
>
>
> In prop E2, the result holds **uniformly over all $p$s** and all $\mathcal{M}_1,\mathcal{M}_2,s_1,s_2$. The idea of the proof is to apply Prop E1 to a each element of grid $\{0,2/K, 4/K,\ldots 2\}$ of $p$s with the failure probability replaced by $\delta/[K+1]$ and to tune the granularity $K$ together with a tedious continuity argument w.r.t $p$. Cf. pp 39 to 42 for details.
>
>
>
>
>
>
> [1] Ledent, A.; Alves, R. Generalization Analysis of Deep Non-linear Matrix Completion. ICML 2024.
>
> [2] Long, P. M.; Sedghi, H. Generalization Bounds for Deep Convolutional Neural Networks. ICLR 2020.
>
> [3] Graf, F.; Zeng, S.; Rieck, B.; Niethammer, M.; Kwitt, R. On Measuring Excess Capacity in Neural Networks. NeurIPS 2022.
>
> [4] Zhang, T. Covering Number Bounds of Certain Regularized Linear Function Classes. JMLR 2002.
>
> [5] Bartlett, P. L.; Foster, D. J.; Telgarsky, M. Spectrally-Normalized Margin Bounds for Neural Networks. NeurIPS 2017.

---

> ### Comment · Reviewer_9AFC · 2025-08-04
>
> Thank you for the very helpful proof sketch. While I believe there are a few minor typos in the proof sketch, the arguments are morally correct. Can you address the following?
>
> - By Markov's inequality... $rank(M_1) \le T \le M^p/\tau^p$.
> That is, the middle $\tau$ in this line in proof sketch should be replaced with $T$, correct?
> - Markov's inequality: Why do you call this inequality Markov's inequality? The fact that $rank(M_1) \le T \le M^p/\tau^p$ is straightforward to verify, but it's odd to call it Markov's inequality since there is no obvious probability distribution on the singular values. Unless you refer to a different inequality due to Markov than $P(X \ge a) \le \mathbb E[X]/a$?
> - What is happening here?
>
> > $\le \sqrt{m} | \le |A^1 x_i - \bar A_i x_i| $
>
> - I believe the bound on $\log(|C_1|)$ should have an $s_1$ instead of $s_2$ in the denominator, correct?
> - The term $(m+C)$ appears to be missing from the bound on $\log(|C_2(\bar A^1)|)$
> - It is suggested to state that the final bound on the metric entropy comes from
> the fact that $|C| \le |C_1| \sup_{\bar A^1 \in C_1} |C_2(\bar A^1)|$ to clarify how the metric entropies are combined.

---

> ### Author Response · Authors · 2025-08-05
>
> Dear Reviewer,
>
>
> Thank you again for your efforts in assessing our paper and for your valuable feedback and suggestions.
>
>
> As the time for discussion is beginning to run out, we would like to reach out and hear about your thoughts on our rebuttal. Please let us know if any further clarification is needed, or if you would like us to elaborate on any particular aspect of the proof sketch, or provide a sketch for any of the other theorems.
>
> Please let us know if our rebuttal has improved your opinion of our work.
>
> Best regards,
>
> Authors

---

> > ### Comment · Reviewer_9AFC · 2025-08-05
> >
> > I apologize! I submitted this feedback yesterday, but did not mark the correct visibility settings. I hope you still have sufficient time to respond.

---

> > > ### Author Response · Authors · 2025-08-06
> > >
> > > Dear Reviewer,
> > >
> > >
> > > **Many thanks** for your thorough reading and your acceptance that our sketch is at least 'morally correct’.
> > >
> > > We apologize for the typos, which we will fix in the revision. We have already started thoroughly working on the camera-ready version.
> > >
> > > We address your points one by one:
> > >
> > > 1. 'The middle $\tau$ should be $T$'. Yes! **Many thanks** for your keen observation. Note that this equation is **already correct in our original submission** (cf. line 597 page 57). We **promise to fix it in the sketch** to be included in the revision.
> > > 2.  'Why do you call this Markov’s inequality...' Many thanks for your keen observation. We did mean to refer to the same inequality by Markov, i.e., $\mathbb{P}(X\geq a)\leq \frac{\mathbb{E}(X)}{a}$: here, the implied 'distribution over singular values’ you mention is a **uniform distribution**. In other words, we are using the following simple version of 'Markov’s Inequality’: let $\lambda_1,\ldots,\lambda_{K}$ be nonnegative numbers and let $T>0$, we have $\\#(i:\lambda_i\geq T)\leq  \frac{\sum_i \lambda_i}{T}.$ Dividing both sides by $K$, this corresponds to Markov’s inequality with a uniform distribution over $\\{1,2\ldots,K\\}$. In our argument, as you are aware, this result is applied to $\lambda_i=\rho_i^p$ and $K=\min(m,d)$. **We agree with you that this is also 'easy to check’.** However, in case some readers may find it harder to immediately verify, we propose to include this detailed justification in the final version to maximise clarity. However, please let us know if you would prefer a different solution.
> > > 3. Apologies for the typo (which doesn’t affect the final conclusion). To write things more precisely, the sequence of inequalities can read: $\|\sigma(A^1x_i)-\sigma(\bar{A}^1x_i)\|\_{2}\leq \|A^1x_i-\bar{A}^1x_i\|\_{2}\leq \sqrt{m}\|A^1x_i-\bar{A}^1x_i\|\_{\infty}\leq \sqrt{m} \epsilon_1 =\frac{\epsilon}{2s_2}$. (Note that in our proof sketch in the rebuttal, norms are assumed to be $L^2$ by default to simplify markdown compiling. ) Note that we need the cover to be with respect to the $L^2$ norm at the intermediary layer to be able to propagate the error forward with the spectral norm of $A^2$, but the cover only needs to be $L^\infty$ at the last layer, consistent with the assumed $L^\infty$-continuity of the loss function. This explains the slight difference in the exponents of the dimensional quantities $m+d$ and $m+C$ in the terms corresponding to the first and second layers respectively.
> > > 4. Yes, the bound on $\log(|\mathcal{C}_1|)$ in the body of the sketch should have $s_1$ instead of $s_2$ in the denominator, consistent with the final formula at the end of the proof and in the Proposition statement. Thank you for your keen reading!
> > > 5. Yes, there should be an additional factor of $[m+C]$ in the bound on $\log(|\mathcal{C}_2|)$ in body of the sketch, consistent with the end of the proof and the Proposition statement. Thank you for your keen reading.
> > >
> > >
> > > Once again, **many thanks** for your careful reading and your interest in our work. We hope you will **consider updating your score** as promised.  However, regardless of the outcome, we are sincerely **happy and grateful to see you take so many steps to understand and appreciate the main ideas of our proof**.
> > > If accepted, we will make sure to include a fully polished version of this proof sketch in the camera-ready version and work on the polishing of the rest of the paper.
> > >
> > > In the meantime, we note that the discussion period has been extended by two days. Therefore, we are **happy to continue answering any questions as needed**. Although we admit the original submission has some minor typos which may make it less reader-friendly, we maintain our confidence in the correctness of our final results, consistent with the fact that **no reviewer has identified any issue which impacts the correctness of any of the formal theorem statements**.
> > >
> > > Best regards,
> > >
> > > Authors

---

> > > > ### Comment · Reviewer_9AFC · 2025-08-06
> > > >
> > > > Thank you for addressing my concerns. I believe this work represents a significant result, and including the proof sketch and addressing other issues we have discussed will improve the presentation. While the proof sketch was morally correct and you have corrected the identified typos, the presence of typos in the proof sketch gives me some trepidation that there may be other unidentified issues in the full proofs. However, I believe reasons to accept outweigh reasons to reject, so I raise my score to a 4.

---

> ### Author Response · Authors · 2025-08-06
>
> Dear Reviewer,
>
> **Many thanks** for increasing your score and for your help improving our work throughout the reviewing process. We promise take all your valuable comments into account and thoroughly polish our whole work for the camera-ready version if accepted.
>
> Best regards,
>
> Authors

---

### Official Review · Reviewer_a1zf · 2025-07-03

**Clarity:** 2
**Significance:** 2
**Originality:** 3
**Rating:** 4
**Confidence:** 3

**Summary:**

The paper presents a generalization bound that utilizes the low-rankness of the weights in deep learning. Choice of the matrix Schatten $p$-norm to characterize network layers result in a particular effect in the generalization bound: as $p\to0$, the remaining dominant term in the bound makes the bound act like a parameter-counting bound, while as $p\to2$, the bound acts like a product-of-norms bound, while still incorporating the approximate low-rankness of the matrices. The authors extend their results to CNNs as well.

**Questions:**

- L8: Schatten norms are introduced as quasi-norms in the abstract, and described as such in L84. Then in the later section results with $0\leq p \leq 2$ are presented. This conflict persists elsewhere in the paper as well, please fix.
- L42: Using a more specific term instead of "architectural parameters" would be helpful
- L56: Cut-off sentence?
- L60: Difficult to understand summary
- L75: The abstract and introduction are unclear regarding whether the focus of the paper will be low-rankness of weights vs. representations. A stark example is L94 discussing representation rank vs. L95 declaring that the paper will focus on weight rank.
- L78: "experimentally" typo
- L83: Dimensions of $A$ wrong?
- L89: "with more tolerance for small non-zero singular values". This is unclear. What does tolerance mean in this case?
- L93: "rank of the activation layer"? Rank of the post-activation representations?
- L94: The term “bottleneck rank” is used multiple times by now without a clear or formal definition, two of these in the abstract. Its relevance to the paper’s main results is also not well motivated. Unless it's central to the core theorems - which currently does not appear to be the case - I suggest removing all references to it. An alternative is to present this within a cogent discussion as in L365 (and doing so *once*).
- L107: "over the classes"?
- L115: Omitting the reference matrix before the clarification in L223 is confusing
- L123: I appreciate that the authors are not burying the lead by introducing baseline vs. their own proposed bounds in Section 1, but without a proper introduction to the notation this gets real confusing real fast
- L139: "spacial" typo
- L145: I cannot understand the main point of this paragraph
- L175: I wonder if this detailed a discussion between previous and current work should be left for after the main results.
- L208: "common joint distribution"
- L208: $\mathcal{D}$ undefined
- L226: "extreme"?
- L228: I don't think this paragraph clearly introduces the significance of the linear network results
- L276: "indices"?
- L277: max. input norm is already defined
- L281: Is the replaced term realistically/often smaller than the maximum input norm?
- L317: Unclear, use the standard term "matricization" perhaps?
- L323: "spacial"

**Ethical Concerns:**

["NO or VERY MINOR ethics concerns only"]

**Final Justification:**

I think that this is a paper that has merit, but is hindered by its disorganized presentation. The authors have committed to resolving these issues, thus I maintain my recommendation for acceptance.

**Limitations:**

The authors are transparent about the limitations of their work.

**Paper Formatting Concerns:**

No outstanding concerns.

**Quality:**

3

**Strengths And Weaknesses:**

## Strengths
- The paper is clearly situated and motivated within the context of previous literature.
- The authors present generalization bounds that exploit the (approximate) low-rankness of the weight matrices. Although their bounds are not necessarily strictly tighter than previous bounds (e.g. with full-rank matrices), that they can be tighter utilizing the approximate low-rankness of the matrices is important.
- That their bounds behave like a parameter-counting bound vs. product-of-norms bound based on the $p$ value is independently interesting
- The authors expand their results to CNNs.

## Weaknesses
- The paper presents a strong narrative and addresses a meaningful problem, but it has significant issues with clarity of the presentation, particularly in the introduction. The reader is presented with technical claims before being properly introduced to the notation and setup; the introduction repeatedly emphasizes the low-rankness of representations in deep learning, only to later clarify that their results are based on the low-rankness of *weights*; the abstract refers to Schatten quasi-norm, but only later does it become apparent that the results apply across the full range $0 \leq p \leq 2$. None of the issues are individually disqualifying but their cumulative effect significantly undermines the readability of the paper. See the section below for more details.

---

> ### Author Rebuttal · Authors · 2025-07-31
>
> **Many thanks** for your careful reading and for your appreciation of our work. We are happy that you deem our work to **address meaningful problems** and ** is clearly situated and motivated within the context of previous literature.**
>
> Re **clarity **(especially the  introduction): many thanks for your comment and for the typos identified.  We promise to restructure the introduction to further improve clarity by defining the relevant quantities earlier and introducing further simplifications in the introduction.
>
> **Re L 281 "realistically smaller"**: yes, the improvement provided in this theorem is frequently significant, since it removes a factor of the product of spectral norms from layer 1 to layer $\ell-1$ (replacing it by an empirical estimate of the norms of the intermediary activations.
>
> **Re "full range $0\leq p\leq 2$"/ L8, conflict with abstract**: **for $p<1$**, the resulting object is **not a norm** but only a quasi norm, though we agree **it is a norm for $p\geq 1$**.  All norms are also quasi norms, but the opposite is not true, which makes it difficult to choose a single term since (as you observe) we cover the full range $p\in[0,2]$. Therefore, we use the term "quasi-norm" in all cases to avoid the more redundant term "quasi norm/norm". We **promise to explain this** more clearly earlier in the introduction and are willing to adapt to any further suggested change of terminology.
>
>
> **Re "low-rank weights, not low-rank activations"**: many thanks for the insightful comment. We do mention the distinction in various places including the conclusion/future directions. However, we agree to discuss it more clearly earlier in the introduction. As mentioned in the paper, the two are very closely related $\star$ and we believe that a significant extension of our techniques would allow us to *directly capture* the low-rank structure in the activations as well (without truncating the rank of the weights after training). However, this requires a much more careful treatment involving advanced uses of loss function augmentation, especially when the activations are only approximately low-rank. From our initial calculations, the singular value thresholds must be tuned simultaneously over all layers, which is very tedious.
>
>
> Re typos: many thanks, we will fix all of them.
>
>
>
>
> Many thanks again for your comments, we promise to discuss all the above issues in more detail in the camera ready revision if accepted and look forward to continuing our discussion during the author-reviewer discussion phase.
>
>
>
>
> Footnote:
> $\star$ Indeed, treating only the exactly low-rank case for simplicity, if the inputs $x_1,\ldots,x_N$ lie in a low-rank space of dimension $r$, then for any high rank weight matrix $W\in\mathbb{R}^{m\times d}$, there exists a low rank weight matrix $\bar{W}$ of rank less than $r$ which has exactly the same effect on these inputs: $Wx_i=\bar{W} x_i$ for all $i$. Indeed, if $U\in \mathbb{R}^{d\times r}$ is an orthonormal basis of that subspace, then $\bar{W}=[WU ]U^\top$ will work because $UU^\top x_i =x_i$ for all $i$.   Thus, **if the activations are low-rank, then the same network can be represented with low-rank weights**.

---

> > ### Comment · Reviewer_a1zf · 2025-08-04
> >
> > I thank the authors for their comments. I am happy to hear that they are committed to addressing the issues I've raised. My most serious concerns were mostly related to the presentation of the work: Though I cannot increase my overall score further without reading an edited version of the paper, I increase my score for clarity in response to authors' feedback.

---

### Official Review · Reviewer_jSQv · 2025-07-03

**Clarity:** 2
**Significance:** 3
**Originality:** 3
**Rating:** 4
**Confidence:** 4

**Summary:**

This paper studies why neural networks often form approximately low-rank representations in their layers during training, a property sometimes called the bottleneck rank. The authors analyze this phenomenon and develop new theoretical generalization bounds that explicitly account for the rank-sparse structure of neural network weights. Using the framework of Schatten p quasi norms, they derive sample complexity results that interpolate between traditional norm-based bounds and parameter-counting bounds, capturing the benefit of low-rank weight matrices. The paper extends these results to linear networks, fully-connected deep neural networks, and convolutional neural networks (CNNs), providing a new perspective on how low-rank properties induced by depth can improve generalization. Overall, the work offers novel theoretical insights on the role of implicit low-rank regularization in deep learning.

**Questions:**

1. The paper introduces a parameter 𝑝 in the Schatten 𝑝-quasi norm, but does not give any practical guidance for choosing or interpreting this parameter. Could the authors discuss how to select 𝑝 or its influence on the bound?
2. Could the authors clarify the approximate magnitude of the bound, or at least discuss whether they are likely to be meaningful for realistic network sizes?
2. The paper’s interpolation between norm-based and rank-based bounds is interesting. Could the authors clarify more explicitly how their framework compares to, or might be combined with, margin-based generalization analyses (e.g., in kernel methods or neural tangent kernel literature)?
2. The theoretical guarantees appear to rest on the existence of low-rank (or approximate low-rank) weight matrices. Can the authors comment on how realistic this assumption is across a wider range of architectures, for example transformers or large language models?

**Ethical Concerns:**

["NO or VERY MINOR ethics concerns only"]

**Final Justification:**

Thanks for Authors' detailed clarification, I have increased my rating. However, the authors should add additional content mentioned in the rebuttal to the final paper version.

**Limitations:**

yes

**Paper Formatting Concerns:**

1. The overall organization of the paper could be significantly improved. Currently, the manuscript jumps rather quickly to presenting the main results without providing a sufficiently detailed problem definition, formal setup, or clear description of the model architecture.

2. Most figures have no detailed captions, which make it diffuicult to inteterpret the key results. There are no definitions on the math symbols shown in the figures.

3. Consistently refer to equations by their numbers for clarity (e.g., “the above bound.”) .

**Quality:**

2

**Strengths And Weaknesses:**

Strengths:

The paper is rigorously grounded in mathematical theory, presenting novel generalization bounds for neural networks by leveraging the low-rank (“bottleneck rank”) structure of their weight matrices.

1. Extending low-rank sample complexity arguments beyond linear networks to FNNs and deep CNNs, while explicitly considering the Schatten p quasi-norm, is innovative.
2. Overall the paper is clearly written, especially in its statement of the main results and their significance. Morever, the related works are concrete including key literatures.
3. The analysis is thorough, covering linear networks, deep neural networks, and CNNs, with detailed treatment of covering numbers and complexity measures. Clear notations, definitions, and assumptions are well documented.

Weaknesses:

1. It presents main theoretical results in section 3, while there is no any problem set up, no empirical results and detailed experimental settings in the main text, though some results presented in the appendix. However, they are not well referred to the theorems presented in the main text.
2. The proofs, while thorough, are highly technical and may be difficult to follow for a broader ML audience not specialized in generalization theory.
3. There is no experimental verification of the tightness of these bounds beyond theoretical discussion (though they mention an appendix with MNIST/CIFAR10 references, it is not sufficiently emphasized and lacks of comparisons with recent approaches).
4. Some transitions between theoretical claims and related works are abrupt, making it hard to track their originality relative to prior results without repeatedly cross-referencing.
5. There is limited demonstration of the practical impact for network design or training, which might weaken its appeal to practitioners.
6. The assumptions of low-rank weight matrices may not hold in all architectures, especially in smaller models or architectures with strong regularization.

---

> ### Author Rebuttal · Authors · 2025-07-30
>
> Many thanks for your efforts reading and reviewing our paper. We try our best to allay most of your concerns in this response and stand ready to answer any further questions you may have during the author discussion period.
>
> Re weakness 1 "there is no any problem set up": The problem set up is described in the "main results" section, and is a **standard supervised learning set up** with neural networks where we study the excess risk in terms of an arbitrary $L^\infty$ loss function. As explained in that section, this setting can cover both classification [1,4] and regression [2] settings, and is a similar generic characterisation as in [3] (mentioned by reviewer wqbT).
>
> **Re weakness 1** "no empirical results and detailed experimental settings in the main text" and **weakness 3** "*There is no experimental verification of the tightness of these bounds beyond theoretical discussion (though they mention an appendix with MNIST/CIFAR10 references, it is not sufficiently emphasized and lacks of comparisons with recent approaches)*":
>
> Our paper contains a large amount of content including novel generalization bounds for linear classifiers as well as bounds for deep neural networks with a low rank structure, it is difficult to include every result in the main paper. However, the appendix does include substantial **experimental verification on pages 31 to 36** where we demonstrate our bounds' competitiveness and ability to capture some of the phenomenon of unused function class capacity. Indeed, we demonstrate:
> 1. That our bounds are competitive with other uniform convergence bounds
> and 2. that our bounds are less sensitive to architecture changes than other bounds in the overparametrization setting. In particular, this demonstrates our bounds' ability to capture unused function class capacity via an alternative notion of effective capacity than the well-established alternative such as the distance to initialization and the margins.
>
> Re weakness 1 "the experiments lack comparison to recent approaches":  whilst our paper is mostly theoretical, we compare our work experimentally to **11 baselines from 7 papers** including one from ICML 2024. The choice was based on a focus on bounds in the same category as ours: **uniform convergence** bounds which do not incorporate the optimization procedure into the bound and do **not overly optimize the bound** with data dependent priors  or by splitting the dataset and learning different parameters on each subset (such as e.g. [3]). To the best of our knowledge, the compared approaches are state of the art in this category. **Please let us know if you have another specific "recent approach" in mind for us to compare to.** This would help us further improve the contextualisation of our work.
>
> Re weakness 5 "no practical impact in terms of model design" and "assumption of low-rank matrices may not hold in practice": the main message in our submission is that the implicit low-rank condition which occurs with depth and overparametrization implies a reduction in effective function class capacity and therefore, estimation error. The convergence to low-rank representations is a broadly studied phenomenon which many recent works have demonstrated occurs at realistic scales. We believe that demonstrating the effects of this phenomenon on learning is worthy in itself, even if it may not yield completely new models (though it does provide some evidence for why depth can reduce rather than increase function capacity in some regimes.
>
>
> On your questions:
>
> 1. As we tried to explain in lines 86-94, the parameters $p$ control the strictness of the low rank constraints and the bounds position on the spectrum between parameter counting bounds and norm-based bounds. Indeed,  when $p\rightarrow 0$, the bound behaves as a parameter counting bound taking into account the rank of the weight matrices. For larger $p$s, the bound behaves more like a norm-based bound based on the *approximate rank* of the weight matrix, with a larger tolerance for moderately small singular values.
>
> Next, note that as explained in line 289, the bounds **hold simultaneously over all values** of $p_l$. Therefore, the $p_\ell$s can be be optimized after training via **grid search**.
>
>
> 2. As can be seen from the (logarithmic) scale of the axes of Figures D1 and D4, the bounds are generally vacuous at realistic scales, even ignoring logarithmic factors.  However, so are all comparable bounds in the literature when used directly. We acknowledge it is possible to obtain tighter bounds through a more aggressive optimization of the bound or by modifying the training procedure to validate certain parameters using the training data. However, such bounds are no longer directly comparable.
>
> 3. **Our bounds** include **margin-based** generalization analyses as particular cases of our results. In particular, our setting for the classification case is exactly the same as that in [1]. In that case, our result is generally tighter but nether uniformly superior nor inferior to most existing approaches. A combination of our results with bounds based on the **Neural Tangent Kernel** would be more subtle since such results  ** rely heavily on the optimization procedure**. However, if one assumes that the neural network training is purely at the neural tangent kernel scale, then the problem becomes analogous to the study of linear predictors, which could in principle be combined with our results. However, to the best of our knowledge, the convergence of neural network learning to a neural tangent kernel limit has not been proved in a reasonable classification setting (all NTK results rely on the square loss).
>
> We promise to incorporate a discussion of these concerns in the camera ready version.
>
>
> 4. As explained in lines 76 to 80, the emergence of the low-rank condition is well-documented in many influential recent papers [5,6,7,8], including large language models [8]. We also verified it experimentally, as can be seen in figures D2 and D4.
>
>
> Re formatting concern 3" in line 248, "the above bound" refers to equation (11) in the theorem just above the remark inside which line 248 is found. We will replace the statement by a concrete reference to the equation as requested, many thanks for helping us improve the presentation of our work.
>
> Re figure captions:  in figures D2 and D4, the plots illustrate the spectral decay of the weight matrices: each given plot represents the singular values of a weight matrix, ordered from the largest one (on the left) to the smallest one (on the right). Hence, the x axis is referred to as "index" in the caption and the y axis similarly refers to the "magnitude" of the singular value. Thus, the graphs are always decreasing by definition, and a graph which hits 0 (i.e. the x axis) at index r would correspond to a matrix of rank exactly less than r. We will  make sure to explain this in further details in the revision.
>
>
>
>
> [1] Peter Bartlett, Dylan J. Foster, Matus Telgarsky. Spectrally-normalized margin bounds for neural networks. NeurIPS 2017
>
> [2] Sanjeev Arora, Simon S. Du, Wei Hu, Zhiyuan Li, Ruosong Wang. Fine-Grained Analysis of Optimization and Generalization for Overparameterized Two-Layer Neural Networks. ICML 2019
> [3] Biggs, Felix, and Benjamin Guedj. "Non-vacuous generalisation bounds for shallow neural networks." ICML 2022.
>
> [4] Florian Graf, Sebastian Zeng, Bastian Rieck, Marc Niethammer, Roland Kwitt. NeurIPS 2022On Measuring Excess Capacity in Neural Networks.
>
> [5] Arthur Jacot. Bottleneck Structure in Learned Features: Low-Dimension vs Regularity Tradeoff. NeurIPS 2023
>
> [6] Zihan Wang, Arthur Jacot. Implicit bias of SGD in L2-regularized linear DNNs: One-way jumps from high to low rank. ICLR 2024
>
> [7]  Jacot, A., Ged, F., Gabriel, F., Simsek, B., and Hongler, C. (2021). Saddle-to-Saddle Dynamics in Deep Linear Networks: Small Initialization Training, Symmetry, and Sparsity. Arxiv 2021
>
> [8]  Laura Balzano, Tianjiao Ding, Benjamin D. Haeffele, Soo Min Kwon, Qing Qu, Peng Wang, Zhangyang Wang, Can Yaras. An Overview of Low-Rank Structures in the Training and Adaptation of Large Models.  ArXiv 2025

---

### Official Review · Reviewer_Udna · 2025-07-08

**Clarity:** 4
**Significance:** 3
**Originality:** 3
**Rating:** 5
**Confidence:** 4

**Summary:**

This paper obtains generalization bounds for neural networks with low complexity weight matrices - both in terms of low rank as well as low Schatten $p$-quasi norm for $0<p<1$. The bound obtained by the authors scales as $O \left ( \sqrt{\frac{L^2 r W}{n}} \right) $ and does not scale exponentially with the depth or width of the network. The authors also evaluate their bounds on trained deep networks and show that their results compare favorably to other generalization bounds.

**Questions:**

1. Are your bounds still larger than the test error? It seems like the are in Fig D.6 In which case, are they still technically vacuous?

2. Is there a route to obtaining non-vacuous generalization bounds through the low-rank bias of deep networks? Possibly applying your techniques with PAC-Bayesian bounds.

**Ethical Concerns:**

["NO or VERY MINOR ethics concerns only"]

**Final Justification:**

I believe this paper makes a meaningful contribution to the literature on generalization bounds. The questions I raised were exploratory in nature and not threshold issues for the paper's validity. I recommend acceptance.

**Limitations:**

yes

**Paper Formatting Concerns:**

Some minor typos:

line 56 - incomplete sentence - should be finished.
line 137 - spatial instead of spacial
line 213 - should the definition of $I_{\gamma, X}$ be the fraction of misclassified samples rather than correctly classified samples? Please clarify.

Appendix D2/D3 - redundant subsection
Equation E.142 - missing square root in the second term?
Equations F.12, F.15 - $mN$ instead of $md$

**Quality:**

4

**Strengths And Weaknesses:**

Strengths: The paper is very well written and comprehensively discusses a lot of related papers on generalization bounds. Their appendix sections serve as a good review of the literature on generalization bounds for deep networks. The application of covering number based bounds allows the authors to avoid exponential dependence on the depth. The authors derive their bound for fully connected networks as well as convolutional networks. They also study how their techniques interact with others such as loss function augmentation to obtain data dependent bounds. The results seem quite favorable, especially when looking at the comparisons on trained networks.

Weaknesses: While this is a strong paper with a lot of technical results, it mostly applies techniques developed by prior work in the context of deep network generalization bounds. Rademacher complexity bounds through metric entropy, parametric interpolation for bounding the covering numbers of approximately low rank matrices, and complexity bounds for convolutional networks that depend on the weights of the filters rather than the number of patches are all known results. Putting them together is a valuable contribution, but that also means that the mathematical novelty in this paper is limited. In case this is mistaken, the authors should do a more precise job of outlining their novel mathematical contributions.

---

> ### Author Rebuttal · Authors · 2025-07-31
>
> Many thanks for your positive review and your interest in our work, as well as the efforts invested in reading our paper. We also appreciate your statements that our paper is **very well written** and **comprehensively discusses** a lot of **related papers**.
>
>
> **Re main weakness** "Putting them together is a valuable contribution, but that also means that the mathematical novelty in this paper is limited. In case this is mistaken, the authors should do a more precise job of outlining their novel mathematical contributions":
>
> Many thanks for your comment. We want to clarify that our bounds are the first to incorporate the technique of **parametric interpolation** into the study of neural networks. Thus, we are the first to provide generalization bounds in this context which are neither fully "parameter counting bounds" nor fully "norm-based" bounds. This is achieved by interpolating between the two regimes.  More precisely, we the class $\mathcal{F}_p:\{A\in\mathbb{R}^{m\times d}: \|A\|\leq s, \|A\|_p\leq \mathcal{M}\}$ into two components: (1) **a component** with (exactly) **low rank**, to be treated with *parameter counting* techniques [2,3], and (2) a component with **small norms**, to be treated with norm based techniques [4,5]. For linear maps $A\in\mathbb{R}^{m\times d}: x\rightarrow Ax$, parameter counting bounds on the covering number of the class $\{A\in \mathbb{R}^{m\times d}}: \|w\|\_{FR}}\leq a\}$ scale as $O\Big( D\text{Polylog}(\epsilon,a,b) \Big)$ where $b$ is a bound on the input samples' norms and $D=md$ is the number of parameters. On the other hand, norm-based bounds scale as $O\Big(\frac{a^2b^2}{\epsilon^2}\text{Polylog}(N,a,b,d)\Big)$. Combining them in an approximately low-rank context allows us to replace $D$ by $[m+d]r$ where $r$ is a threshold rank depending on the Schatten quasi norms, gives rise to a completely different dependency on $\epsilon, a,b$ and even the Lipschitz constant of the loss (i.e. the margin parameter in a classification context). The combination is not trivial and involves decomposing the function class by thresholding the singular values with a carefully tuned threshold (cf. eq F13).  In addition, there are many additional technicalities which are new to this setting. For instance, the continuity argument to make the bounds post hoc with respect to all the Schatten indices $p\_{\ell}$ is relatively tedious.  To further improve your assessment of the novelty of our method, we invite you to read the **proof sketch** of a minimum working product version of our theorem which is **provided in our answer to reviewer 9AFC**.
>
> We promise to further emphasise the novelty of our techniques and provide the proof sketch in the camera ready version if accepted.
>
>
> Re typos:
>
> Line 56 (also pointed out by reviewer 9AFC): we will add the word "small" at the end of the sentence, many thanks. Line 137: thank you for the correction. Line 213: yes, you are right. Many thanks for helping us improve the quality of the manuscript.
>
> E. 142: yes, you are right, we will fix this, which will harmonize the result with the version provided in the main paper. Idem for F12/F15. Once again, we **thank you8* warmly for your efforts in reading our main paper and appendix carefully. We stand **ready to answer any further questions** as required.
>
> Re question 1 "is the bound larger than the test error": we appreciate your observation based on the axes of the graph. Yes, like most comparable works, the bounds are vacuous at realistic scales if we do not apply further tricks (e.g. compression, data dependent priors, aggressive optimization of the bound through additional spectral regularization). We wanted to compare how all bounds vary with architectural parameters in a reasonable simple pure uniform convergence setting which excludes applying too many post-processing tricks to make the bounds tighter. Our reasoning is that the numerical values of generalization bounds are not as important as how they behave when the architecture changes. Indeed, numerically tighter estimates of the test error can be obtained with a validation set. Our bounds demonstrate the validity of  the approximate rank as a complement to margins [3,5] as a measure of **unused function class capacity**.
>
> Re question 2: "route to non-vacuous bounds": yes, we believe it is possible to achieve non-vacuous bounds with more aggressive tricks. For instance, the bounds for the fully connected case are not far from being non vacuous if one ignores logarithmic factors. They may become non-vacuous if we were to tweak the architecture, explicitly **compress** the rank by removing higher order singular values, and apply spectral norm constraints. In addition, incorporating PAC-Bayesian techniques, including data-dependent priors [7,9,10] into our analysis is a **tantalizing direction for future research** with the potential for non-vacuous results. However, we wanted to limit our setup to uniform convergence as closely as possible to provide a fair proof of concept. We hope and believe our contributions are a sufficient component to warrant one publication, without having to combine it to even more other techniques to make the bounds numerically tighter, as we believe doing so would make the paper even less approachable to readers and reviewers.
>
> On a side note, we would also like to point out that it is very difficult, if not impossible, to compare all bounds in the literature without ignoring logarithmic factors and constants. Indeed, many works (e.g. [2]) skip constants entirely, and **many works (e.g. [2,3]) do not explicitly write down the proof of the post hoc step,**  leaving it as an exercise for the reader. This makes a comparison with our theorem E.2. unfair if one includes the logarithmic factors arising from the numerous post hoc steps over $s\_{\ell}, p\_{\ell}, \mathcal{M}\_{\ell},$ etc.
>
>
> Once again, we thank you for your appreciation of our work and look forward to our further discussion with you.
>
>
>
>
> ============================**Continuation of Answer to Reviewer 9AFC**======================
>
>
>
> Many thanks for your detailed comments. We **appreciate the constructive suggestion** of a proof sketch and the corrections of the typos found in the appendix. We promise to fix all of these typos and suggestions, including introducing a footnote at line 38.  We also promise to cite and discuss [6] as requested. Re Line 115, we will remove the initialized matrices $M_\ell$ from the introduction to improve clarity. Many thanks. Re 250, both equation (2) and Neyshabur et al. refer to the same result, we will clarify this and fix the sentence. Re 285/266: many thanks, we will harmonize both notations by removing the initialized matrices $M\_{\ell}$ from the introduction.
>
> **Re dependency on the input space dimension:**/**possibility to remove the dependence on the input dimension or the hidden layer**.
>
>
> Many thanks for your comment, which is closely related to reviewer wqbT's insightful question 1. For the one layer case, we agree that our results translate to a sample complexity of at least $[m+d]$ where $d$ is the input dimension. This is a weakness compared to the purely norm-based bounds in [4,5], which only involve the Frobenius norm of the weights. In particular, even for $p=2$, the bound doesn't coincide with [4,5]. To the best of our knowledge, this limitation is extremely challenging to remove (we invite the reviewers to check the argument involving Markov's inequality in the proof sketch of Prop F2 to satisfy themselves that no simple modification of the technique can improve this). However, this limitation *doesn't make our results uniformly inferior* to either norm-based or parameter counting bounds.
>
>
> We want to clarify that when we claim it could be possible to improve the dependence on the input dimension, we mean to achieve this under the assumption that the input samples approximately lie in a low-rank subspace. Similarly, taking into account the fact that activations approximately lie in a low-rank subspace may also allow us to further improve the dependency. However, these as-yet uncertain improvements require very significant technical modifications to the proofs which are better left to future work.
>
> Cf answer to rev. wqbT for further details on these last questions.
>
>
> We thank you again for your interest in our work and are looking forward to our further discussion.
>
>
>
>
>
> =================================**References**===========================
>
>
> [1] Ledent, A.; Alves, R. Generalization Analysis of Deep Non-linear Matrix Completion. ICML 2024.
>
> [2] Long, P. M.; Sedghi, H. Generalization Bounds for Deep Convolutional Neural Networks. ICLR 2020.
>
> [3] Graf, F.; Zeng, S.; Rieck, B.; Niethammer, M.; Kwitt, R. On Measuring Excess Capacity in Neural Networks. NeurIPS 2022.
>
> [4] Zhang, T. Covering Number Bounds of Certain Regularized Linear Function Classes. JMLR 2002.
>
> [5] Bartlett, P. L.; Foster, D. J.; Telgarsky, M. Spectrally-Normalized Margin Bounds for Neural Networks. NeurIPS 2017.
>
> [6] Suzanna Parkinson, Greg Ongie, Rebecca Willett. ReLU Neural Networks with Linear Layers are Biased Towards Single- and Multi-Index Models
>
>
> [7] Gintare Karolina Dziugaite, Kyle Hsu, Waseem Gharbieh, Gabriel Arpino, Daniel M. Roy. On the role of data in PAC-Bayes bounds
>
> [8] Jacot, A., Ged, F., Gabriel, F., Simsek, B., and Hongler, C. (2021). Deep linear networks dynamics: Low-rank biases induced by initialization scale and l2 regularization. https://arxiv.org/abs/2106.15933v1
>
> [9] Zhou, Wenda, et al. "Non-vacuous generalization bounds at the imagenet scale: a PAC-bayesian compression approach."
>
> [10] Biggs, Felix, and Benjamin Guedj. "Non-vacuous generalisation bounds for shallow neural networks." International Conference on Machine Learning. PMLR, 2022.
>
> [11]  Neyshabur, Bhojanapalli, Srebro.  A PAC-Bayesian Approach to Spectrally-Normalized Margin Bounds for Neural Networks. ICLR 2018.

---

### Decision · Program_Chairs · 2025-09-17

**Decision:**

Accept (poster)

**Comment:**

The authors develop generalization bounds for linear networks, fully-connected and convolutional neural networks.  Using the notion of Schatten quasi norms, they have obtained sample complexities that interpolate between classical  norm-based bounds and parameter-counting bounds.

The reviewers had originally concerns about the novelty of the results, tightness of the bounds, and appropriateness of the experiments to test the implications of theory. All reviewers are already happy with the paper and noted the rebuttal addressed their major concerns. Considering the rebuttal and reviewers’ final justification, I would like to recommend acceptance and suggest that authors address all revisions that are promised within the camera-ready version.